# Time-frequency super-resolution with superlets

Vasile V. Moca[1,3], Harald Bârzan[1,2,3], Adriana Nagy-Dăbâcan[1] & Raul C. Mureșan [1✉]

Due to the Heisenberg–Gabor uncertainty principle, finite oscillation transients are difficult to localize simultaneously in both time and frequency. Classical estimators, like the short-time Fourier transform or the continuous-wavelet transform optimize either temporal or frequency resolution, or find a suboptimal tradeoff. Here, we introduce a spectral estimator enabling time-frequency super-resolution, called superlet, that uses sets of wavelets with increasingly constrained bandwidth. These are combined geometrically in order to maintain the good temporal resolution of single wavelets and gain frequency resolution in upper bands. The normalization of wavelets in the set facilitates exploration of data with scale-free, fractal nature, containing oscillation packets that are self-similar across frequencies. Superlets perform well on synthetic data and brain signals recorded in humans and rodents, resolving high frequency bursts with excellent precision. Importantly, they can reveal fast transient oscillation events in single trials that may be hidden in the averaged time-frequency spectrum by other methods.

[1] Department of Experimental and Theoretical Neuroscience, Transylvanian Institute of Neuroscience, Pta. Timotei Cipariu 9/20, 400191 Cluj-Napoca, Romania. [2] Basis of Electronics Department, Technical University of Cluj-Napoca, Str. G. Baritiu 26-28, 400027 Cluj-Napoca, Romania. [3] These authors contributed equally: Vasile V. Moca, Harald Bârzan. ✉email: muresan@tins.ro

Time-series describing natural phenomena, such as sounds, earth movement, or brain activity, often express oscillation bursts, or "packets," at various frequencies and with finite duration. In brain signals, these packets span a wide range of frequencies (e.g., 0.1–600 Hz) and temporal extents ($10^{-2}$–$10^2$ s)[1,2], and these signals were proposed to have a fractal, scale-free nature[3–7], whereby their properties are self-similar across different timescales/ frequencies. Identifying the frequency, temporal location, duration, and amplitude of finite oscillation packets with high precision is a significant challenge.

Time-frequency (TF) analysis of digitized signals is traditionally performed using the short-time Fourier transform (STFT)[8], which computes Fourier spectra on successive sliding windows. Long windows provide good frequency resolution but poor temporal resolution, whereas short windows increase temporal resolution at the expense of frequency resolution. This is the effect of the Heisenberg–Gabor uncertainty principle[9] or the Gabor limit[10], i.e., one cannot simultaneously localize precisely a signal in both time and frequency[11]. Frequency resolution is proportional to window size, as defined by the Rayleigh frequency[12,13]. Therefore, shortening the window to gain temporal resolution leads to a degradation of frequency resolution (Fig. 1a, left)[14].

For a given window size, the STFT has fixed frequency resolution but its temporal precision relative to period decreases with increasing frequency (Fig. 1a, right), i.e., as frequency increases, the size of an oscillation packet spanning a finite number of cycles is decreasing relative to the fixed analysis window size. This is especially problematic in the analysis of scale-free, fractal-like signals that contain oscillation bursts which are self-similar across frequencies. To overcome this limitation multiscale, also called multiresolution[15,16] techniques have been introduced, like the continuous-wavelet transform (CWT). The CWT provides good relative temporal localization by compression/dilation of a mother wavelet as a function of frequency[11,14]. A popular wavelet for TF analysis is the Morlet[17,18], defined as a plane wave multiplied by a Gaussian envelope (see Supplementary Fig. 1). The original Morlet wavelet contains two terms, the second being a normalization constant to render the wavelet admissible (i.e., to remove its mean)[18]. In practice, when the wavelet is wide enough, this constant becomes negligible, and one can define the modified Morlet (also called Gabor) wavelet, as:

$$\psi_{f,c}(t) = \frac{1}{B_c \sqrt{2\pi}} e^{-\frac{t^2}{2B_c^2}} e^{j2\pi f t} \tag{1}$$

$$B_c = \frac{c}{k_{sd} f} \tag{2}$$

where, $f$ is the central frequency, $c$ is the number of cycles of the wavelet, $B_c$ is the time spread parameter (in $\mathrm{Hz}^{-1} = \mathrm{s}$), controlling the time variance of the wavelet[19]. The time spread parameter is inversely proportional to the variance in frequency, i.e., a smaller $B_c$ spreads the energy in a wider frequency band, and vice versa[20]. In Eq. (2), we set $B_c$ such that the plane wave spans $c$ full cycles within $k_{sd}$ SDs of the Gaussian envelope. Throughout the rest of the study, we set $k_{sd} = 5$. In practice, $k_{sd}$ is a design choice and is almost never changed.

The Morlet wavelet does not have compact support but it has other advantages, such as optimal joint TF concentration[19]. In practice, the Gaussian decays dramatically outside a range of ±3 SDs, such that in numerical implementation one considers a Morlet window spanning 6 SDs (see Supplementary Fig. 1). A Morlet wavelet with higher time spread parameter contains more cycles, is wider in time, but has a narrower frequency response (narrow frequency bandwidth). Here we will use Morlet wavelets for TF analysis, but other choices are also possible.

The normalization of the wavelet is an important aspect to consider because it determines the ability of the time-scale (TS) representation to express different properties of the data[21]. When the wavelet is normalized to unit energy, the representation spreads the energy captured by the wavelet over its temporal extent and its frequency bandwidth[15]. As the frequency bandwidth increases as the wavelet compresses, it follows that the representation of an oscillation burst which is progressively compressed in time (increase in frequency) becomes progressively spread out in frequency, i.e., it is diluted out. This is detrimental if one desires a scale-free representation, whereby an oscillation burst with constant peak amplitude is represented with the same peak magnitude, independent of its frequency. Scale-free representations are especially useful for data that have a fractal, scale-free nature, such as brain signals, where oscillation bursts can be self-similar across frequencies.

Here we use a wavelet in Eq. (1) that is normalized to the unit integral of the modulus rather than to unit energy. This normalization enables the estimation of the instantaneous power at scale (or power captured by wavelet)[21] in a way that is independent of frequency, being widely used in wavelet ridge detection[11,22,23]. The main advantage of this normalization is that, rather than focusing on conservation of energy, wavelets become better suited to detecting events that are self-similar across scales, i.e., such events receive the same "intensity" in the representation if they have the same shape and same peak amplitude but are simply scaled (compressed/dilated). For an in-depth discussion, please refer to Supplementary Information—"II. Fundaments of spectrograms and scalograms."

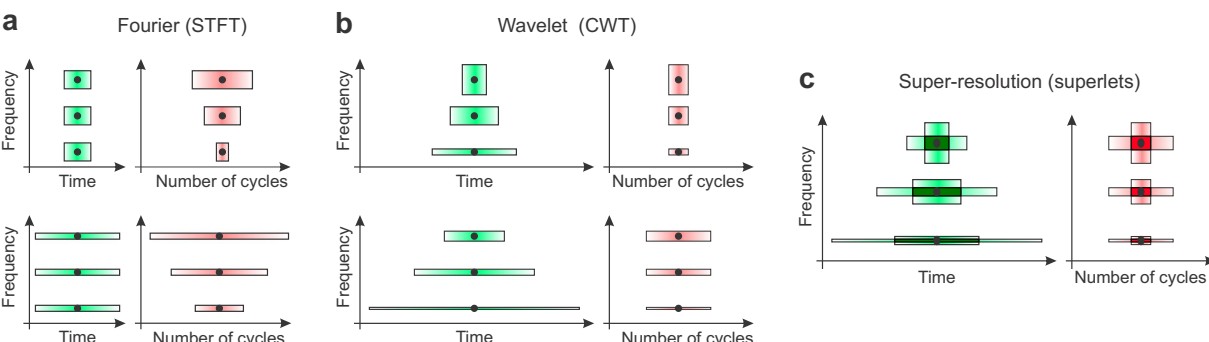

**Fig. 1 Sketch of time-frequency uncertainty in Fourier (STFT), wavelet (CWT), and superlet (SL) analysis. a** Time and frequency resolution of the STFT for a short (top) and wide (bottom) window at three different frequencies. Temporal resolution is expressed in time (left) or in oscillation cycles at the target frequency (right). **b** Same as in **a** but for wavelets (CWT). Here, the number of cycles is fixed across the spectrum but the spanned temporal window decreases with frequency increase. **c** Superlets of order 2 (SLT). Time-frequency super-resolution is achieved by combining short, large-bandwidth wavelets, with longer, narrow-bandwidth wavelets.

In practice, the STFT and CWT are used to generate TF and TS representations, respectively, by computing the spectrogram (squared modulus STFT) and the scalogram (squared modulus CWT). For simplicity of the notation, throughout the rest of the paper we will use STFT-spectrogram and CWT-scalogram interchangeably. Also, we will convert scale to frequency to facilitate comparisons.

The CWT localizes well the oscillation packets in time, but trades in frequency resolution as frequency increases[24,25] (Fig. 1b). Neighboring high-frequency components cannot be distinguished, i.e. the representation is redundant across wavelets with close central frequencies in the high range. For this reason, analyses are often performed using a dyadic representation, like in the discrete wavelet transform (DWT), where frequencies are represented as powers of 2[24,26]. This representation however resolves the high frequencies very poorly.

Both the STFT and CWT (or DWT) have significant limitations. The STFT provides good frequency resolution but poor relative temporal resolution at high frequencies, whereas the CWT maintains a good relative temporal resolution throughout the spectrum but degrades in frequency resolution and becomes redundant with increasing frequency. This TF uncertainty hampers analysis of neuronal signals, which have rich TF content[27,28].

To overcome the limitations of the STFT, it has been proposed to combine Fourier-based spectrograms obtained with a short and a long window[29], or with a set windows with varying sizes[30]. This technique was termed super-resolution[31,32], because it can localize oscillation packets simultaneously in both time and frequency better than it is possible with any single spectrogram. A similar idea, using multiple measurements or estimates, is applied in super-resolution methods used in imaging[33,34].

To increase resolution, multiple spectrograms can be combined by computing their geometric mean (GM)[29], which is equivalent to the minimum mean cross-entropy (MMCE)[30,35]. The latter is optimal with respect to an entropic criterion[30]. While any such combination of representations can be called MMCE, for historical reasons we will use the term MMCE to refer to the geometric combination of spectrograms throughout the rest of the paper. Here we use the minimum mean cross-entropy technique in combination with wavelets to introduce an approach that reveals a sharper localization of oscillation packets than can be achieved with STFT and CWT.

Other high-resolution techniques are based on directionally smoothed Wigner–Ville distributions (WVD)[15]. The WVD provides the highest possible TF resolution, but in multi-component signals it suffers from cross-terms (TF artifacts) that render it unusable for some practical applications[32]. The WVD can be smoothed with a variety of kernels and an infinite number of TF representations can be obtained, including the spectrogram. When the kernel is a unit modulus "directional" filter, the smoothed WVDs are called Cohen class[36], which has many nice properties, including preservation of marginals. Among the directionally smoothed WVDs, those based on the Choi–Williams (CW)[37] and Born–Jordan (BJ)[38] kernels are among the most popular.

Here we introduce a high-resolution technique based on wavelet sets and compare its performance to that of the classical STFT (spectrogram) and CWT (scalogram), and of other high-resolution techniques, such as MMCE, CW, and BJ. The comparison will be mainly performed on brain signals and conclusions should be interpreted within this context.

## Results

### Superlets

In structured illumination microscopy, one uses a set of known illumination patterns[34] to obtain multiple measurements that are combined to achieve super-resolution. Similarly, in signal analysis one can combine multiple estimates by computing spectrograms with multiple windows[29,30]. The technique proposed here employs multiple wavelets to detect localized TF packets better than a single wavelet does.

The method can be formalized as follows. A base wavelet, e.g., Morlet with a fixed number of cycles, provides multiscale in the standard sense, with constant relative temporal resolution but degrading frequency resolution (increased redundancy) as the central frequency of the wavelet increases. By increasing the time spread parameter of the wavelet (more cycles), one increases frequency resolution (Fig. 1b) but loses temporal resolution. To increase resolution, we propose to combine short wavelets having high temporal resolution (small number of cycles, low time spread parameter) with longer wavelets, having high-frequency resolution (larger number of cycles, lower temporal resolution) (Fig. 1c), in the same manner this has been done with spectrograms[29,30].

A "superlet" (SL) is defined as a set of wavelets with a fixed central frequency, $f$, and spanning a range of different cycles (progressively constraining the bandwidth):

$$\mathrm{SL}_{f,o} = \left\{ \psi_{f,c} \,|\, c = c_1, c_2, \ldots, c_o \right\} \quad (3)$$

where, $o$ is the "order" of the SL, and $c_1, c_2, \ldots, c_o$ are the number of cycles for each wavelet in the set. A SL of order 1 is a single (base) wavelet with $c_1$ cycles. In other words, a SL is a finite set of $o$ wavelets spanning multiple bandwidths at the same central frequency, $f$. The order of the SL represents the number of wavelets in the set. The number of cycles defining the wavelets in the SL set can be chosen multiplicatively or additively. In a multiplicative SL, $c_i = i \cdot c_1$, whereas in an additive SL $c_i = c_1 + i - 1$, for $i = 2, \ldots, o$. Unless specified otherwise, here we will always operate with multiplicative SLs.

We define the response of a SL to a signal, $x$, as the GM of the responses of individual wavelets in the set:

$$R\left[\mathrm{SL}_{f,o}\right] = \sqrt[o]{\prod_{i=1}^{o} R\left[\psi_{f,c_i}\right]} \quad (4)$$

where, $R[\psi_{f,ci}]$ is the response of wavelet $i$ to the signal, i.e., the complex convolution (for complex wavelets, such as Morlet):

$$R\left[\psi_{f,c_i}\right] = \sqrt{2} \cdot x * \psi_{f,c_i} \quad (5)$$

where, $*$ is the complex convolution operator and $x$ the signal. The SL is an estimator of the oscillation packets present in the signal at the central frequency, $f$, of the SL. To estimate magnitude, one computes the GM of response magnitudes on individual wavelets. To compute a scalogram, the magnitude of the SL is simply squared. The $\sqrt{2}$ term in Eq. (5) is used only for analytic wavelets, e.g., the complex Morlet or Gabor. The reason is that the analytic wavelet recovers only half the power of a real signal. For a proof, see Supplementary Information—"II. Fundaments of spectrograms and scalograms".

The SL transform (SLT) of a signal is computed analogously to the CWT, except that one uses SLs instead of wavelets. A SLT with SLs of order 1 is the CWT. As will be shown next, the SLT with orders >1 is a less redundant, sharper representation of the signal than the corresponding CWT. For a rigorous analytical treatment of SLs, please refer to Supplementary Information —"III. Superlets and redundancy suppression: towards multiscale TFRs".

### Adaptive SLs

At low central frequencies, single wavelets (i.e., SLs of order 1) may provide sufficient TF resolution. Indeed, the CWT is less redundant at low than at high frequencies[24].

Adaptive SLs (ASLs) adjust their order to the central frequency to compensate the increasing wavelet bandwidth with increasing frequency. In an adaptive SLT (ASLT), one starts with a low order for estimating low frequencies and increases the order as a function of frequency to achieve an enhanced representation in both time and frequency across the entire frequency domain, as follows:

$$\mathrm{ASL}_f = \mathrm{SL}_{f,o}|o = a(f) \qquad (6)$$

where, $a(f)$ is a monotonically increasing function of the central frequency, having integer values. A simple choice is to vary the order linearly:

$$a(f) = o_{\min} + \left[ (o_{\max} - o_{\min}) \cdot \frac{f - f_{\min}}{f_{\max} - f_{\min}} \right] \qquad (7)$$

where, $o_{\min}$ is the order corresponding to the smallest central frequency, $f_{\min}$, $o_{\max}$ is the order corresponding to the largest central frequency, $f_{\max}$, in the TF representation, and [] is the nearest integer (round) operator. We recommend using the ASLT when a wide frequency range needs to be resolved and the SLT for narrower bands. The parameters $o_{\min}$ and $o_{\max}$ can be computed analytically, for certain design parameters. It can be shown that the ASLT can achieve a constant absolute bandwidth. For more information, see Supplementary Information—"III. Superlets and redundancy suppression: towards multiscale TFRs"). In a constant absolute bandwidth configuration, the ASLT provides TF representations that are more similar to the MMCE.

The ASLT may introduce "banding" in the representation due to the discrete jumps of the order of the SL as the frequency increases. To overcome this issue, a flavor of SLs may be used, called fractional SLs, where one uses the weighted GM in the SL formula, such that the order can be a fractional number[39]. This enables the continuous and smooth variation of the order. The fractional ASLT (FASLT) provides sharp representations across the entire frequency domain[39].

**Operating principle of SLs**. We will first illustrate the basic principle behind SLs by considering a known set of packets composed of seven sinusoidal cycles. A target oscillation packet, T, is composed of a finite number of cycles at a target central frequency. We define two additional oscillation packets: a temporal neighbor $N_T$, having the same frequency, but shifted in time with a temporal offset $\Delta t$, and a frequency neighbor $N_F$, at the same location in time, but shifted with a frequency offset $\Delta f$ (Fig. 2a, top). For convenience, all packets have an amplitude of 1.

An example instantiation of this scenario is shown in Fig. 2a, bottom, for a target frequency of 50 Hz in a signal sampled at 1 kHz. We next evaluated how the presence of $N_F$ or that of $N_T$ influences the estimation at the location of T. In other words, without T being present, we systematically moved $N_F$ in frequency or $N_T$ in time and computed their contribution (leakage) to the estimate at the TF location of T (Fig. 2b). As estimators, we initially considered a wavelet with $c = 3$ cycles and a multiplicative SL with $c_1 = 3$ and $o = 5$. The bandwidth of the wavelet was large, with a broad frequency response around the target frequency of T, indicating that T was hard to distinguish from $N_F$ over a large frequency domain (Fig. 2b, top). By contrast, the SL significantly sharpened the frequency response, reducing frequency cross-talk between $N_F$ and T. Along the temporal dimension, when $N_T$ was shifted in time away from the target's location (time offset 0), the response of both the wavelet and the SL dropped sharply after half the size of the target packet (3.5 cycles) (Fig. 2b, bottom). This indicates that, while increasing frequency resolution, the SL did not induce a significant loss of temporal resolution.

To investigate how the SL achieves this, we evaluated the full width at half maximum of the frequency and temporal responses measured at T and induced by $N_F$ and $N_T$, respectively (Fig. 2c). We varied the order of the SL and compared its response to the one of the longest wavelet in its corresponding set [$c_o$, see Eq. (3)]. As the order was increased, both the largest wavelet (with highest time spread parameter) and the SL approached the frequency resolution limit (Rayleigh frequency corresponding to the Gaussian-windowed oscillation packet) (Fig. 2c, top). By contrast, while the single wavelet's temporal resolution decreased rapidly by increasing its number of cycles, the temporal resolution of the SL degraded considerably slower (Fig. 2c, bottom). These results indicate that, as its order is increased, a SL nears the theoretical frequency resolution possible for a limited duration oscillation packet (Rayleigh frequency), while maintaining a significantly better time resolution than a single, long wavelet.

The CWT provides a representation of the signal that is increasingly redundant for higher frequencies[24,25], because the frequency response of wavelets becomes broader as the number of samples per oscillation cycle decreases. The SLT (and ASLT) decreases the redundancy of the representation with increasing order of the SLs. Figure 2d depicts the average power measured over a long signal composed of three frequency components (20, 50, and 100 Hz) with unitary amplitude. The average power in a perfect energy-conserving transform should be 1.5 (Fig. 2d, green). We used two types of SLs with base cycles $c_1 = 3$ and 5, and progressively increased their order while computing the SLT and collapsing it in time (Welch-like). Order 1 corresponded to the CWT and, as the order was increased, the redundancy in the representation of high frequencies was reduced (Fig. 2d, insets) and the average power across the spectrum approached that of an energy-conserving transform (e.g., Fourier). Importantly, SLs with larger base cycles provide a less redundant representation than those with smaller number of base cycles (compare Fig. 2d red with Fig. 2d blue), albeit at the expense of decreased temporal resolution (see below).

The SL acts as a sharp narrow-bandwidth bandpass filter, concentrating frequency resolution as the order is increased and suppressing the redundancy of the representation. We have analytically derived the frequency response of the SL as a function of its order (see Supplementary Information—"III. Superlets and redundancy suppression: towards multiscale TFRs/Redundancy suppression by superlets–analytical derivation"). Results indicate that the frequency resolution of the SL is very close to that of the largest wavelet in its set.

**Comparison to classical techniques**. In another test, we generated a signal as a sum of multiple TF packets (Fig. 3a), as follows. Three target packets of 11 cycles were generated at target frequencies of 20, 40, and 60 Hz. For each target, a neighbor in frequency (+10 Hz) and a neighbor in time (+12 cycles) were added to the signal. Due to constructive-destructive summation, a clear modulation of magnitude is visible where the target was summed with its frequency neighbor—this is equivalent to amplitude modulation (AM), whereby two sideband frequencies sum up to give rise to an amplitude modulated central frequency. Locally, the correct TF representation of this phenomenon should reveal corresponding bursts of magnitude (or power) at the two summed frequencies (i.e., locally the signal looks like it is modulated in amplitude and it is composed of two AM sideband components; thus, locally, both interpretations are correct at the same time). We computed the TF power representation of the signal using Blackman-windowed Fourier (STFT), wavelets (CWT), and adaptive additive SLs ($o = 1:30$, order varied

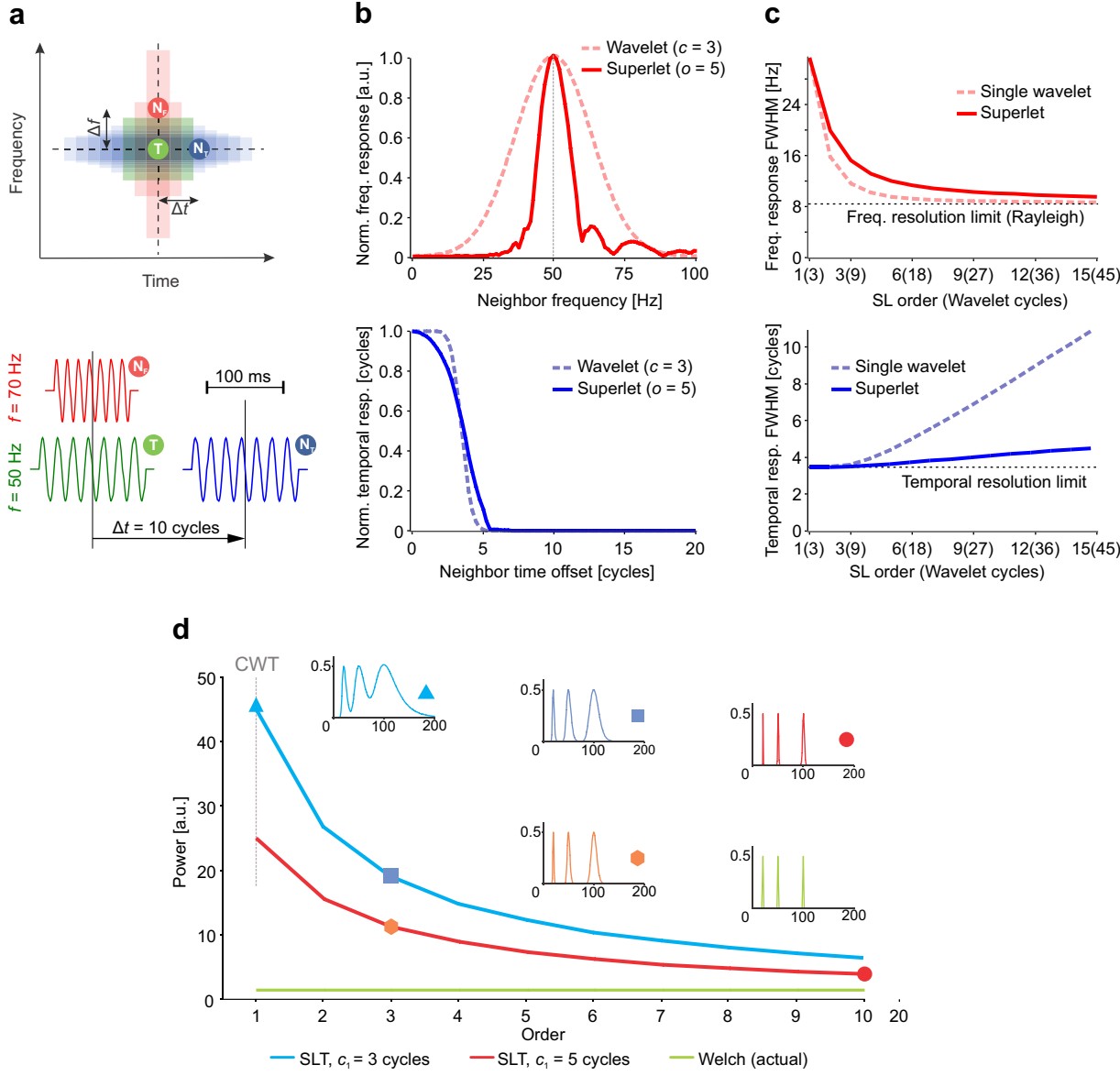

**Fig. 2 The principle behind superlets and redundancy of spectral representation. a** Test setup where a target oscillation packet, T, is contaminated by frequency and time neighbors ($N_F$ and $N_T$). Top: using a wavelet with small number of cycles enables good time separation but has poor frequency resolution (red), whereas wavelets with many cycles enable good frequency separation but suffer from temporal contamination (blue). Bottom: a particular instantiation with packets of seven cycles having: target frequency 50 Hz, neighbor frequency 70 Hz, neighbor time offset ten cycles. **b** Target contamination in frequency by $N_F$ (top) and in time by $N_T$ (bottom). Contamination is measured as the normalized response (magnitude) of a single wavelet ($c = 3$) or a multiplicative superlet ($c_1 = 3$; $o = 5$) at the time-frequency location of the target (without the target being present) to $N_F$ with various frequencies (top) or $N_T$ with various time offsets (bottom). **c** Frequency (top) and time (bottom) superlet resolution measured as the half-width of the frequency and time peak in **b**, respectively, as a function of the order of a multiplicative superlet (line). The same is shown for the longest wavelet in the superlet set (dotted line). The frequency resolution limit is the Rayleigh frequency of T with Gaussian windowing. The temporal resolution limit is half the size of T (3.5 cycles). **d** A long signal composed of 3 summed unitary amplitude sine waves has an average power of 1.5 (green). Two superlet transforms (SLT) using multiplicative superlets with $c_1 = 3$ (blue) and 5 (red) give an increasingly sharper representation of the higher frequencies, as their order is increased. Sharper representations signify less redundancy. Insets show the time-collapsed power spectra computed using the CWT (SLT of order 1), SLT, and Welch for the corresponding marked points on the average power traces.

linearly from 1@10 Hz to 30@80 Hz, at $c_1 = 3$ and 5 cycles; and $o = 5 : 40$ at $c_1 = 1$ cycle) (see Fig. 3).

At the location of frequency neighbors, the STFT with various window sizes revealed either the temporal modulation (Fig. 3b, left) or the two frequencies (Fig. 3b, right), but it was unable to fully segregate time and frequency, in spite of an "optimized" intermediate window size (Fig. 3b, center). A similar conclusion was reached with a CWT using increasing number of wavelet cycles (increasing time spread parameter; Fig. 3c), with the

difference that the CWT provided better frequency resolution in the low frequency range. By contrast, ASLs (ASLT) provided a faithful local representation, with high resolution in both time and frequency, across the entire spectrum (Fig. 3d). Increasing the number of base cycles ($c_1$) had the effect of further increasing frequency precision, albeit at the cost of losing some temporal resolution at the low frequencies. On the other hand, one could decrease $c_1$ to achieve higher time resolution and compensate the larger overall frequency bandwidth with higher orders in the SLs

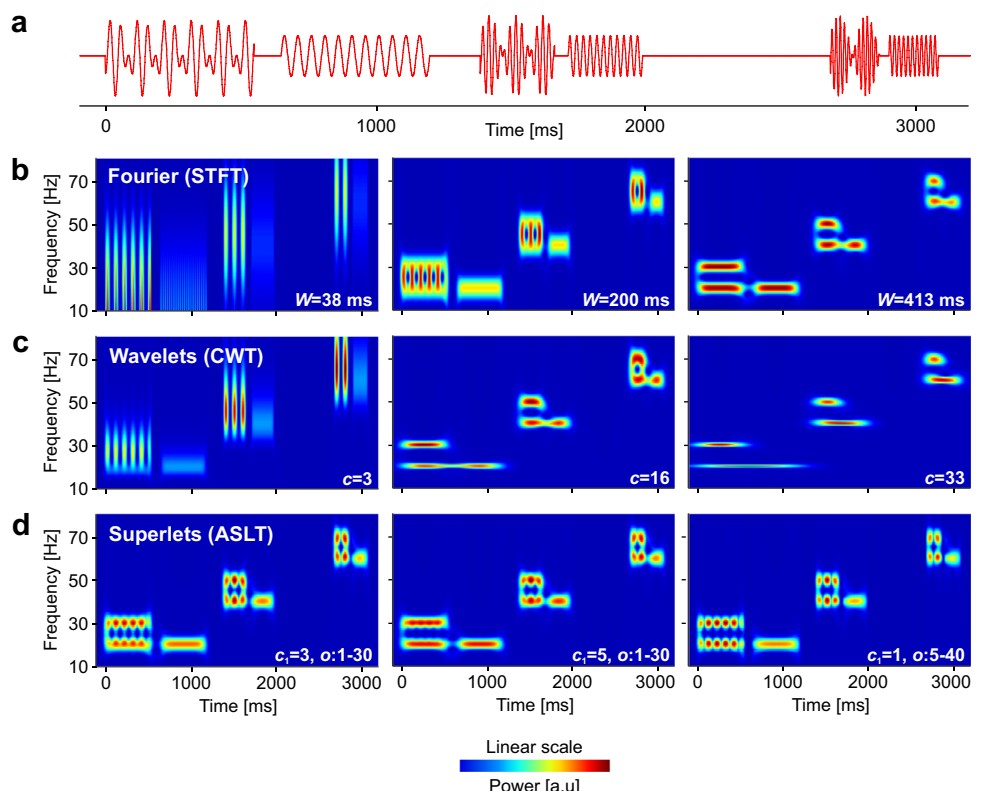

**Fig. 3 Evaluation of time-frequency resolution on a known signal structure. a** Test signal containing 3 target packets with 11 cycles at 20, 40, and 60 Hz, each accompanied by a neighbor in frequency (+10 Hz) and a neighbor in time (+12 cycles). The signal sampling rate is 1024 samples/s. **b** Time-frequency power representation of the signal using STFT (Blackman window) and window size 38, 200, and 413 ms (roughly matching the size of a single wavelet with 3, 16, 33 cycles at the largest target frequency). **c** Same as in **b** but using Morlet wavelets with increasing number of cycles. **d** Same as **b** and **c** but using adaptive additive superlets with linearly varying order from: $o = 1$, for 10 Hz, to $o = 30$, for 80 Hz, and with different number of base cycles ($c_1 = 3$, left; $c_1 = 5$, middle), and $o = 5:40$, $c_1 = 1$ (right). Power is given in arbitrary units (a.u).

(Fig. 3d, right). This latter strategy led to a representation that surpassed all the others.

We next used SLs to analyze brain signals (electroencephalography, EEG) recorded from humans in response to visual stimuli representing objects (deformable dot lattices)[40] (see "Methods"). As EEG signals are strongly affected by the filtering properties of the skull and scalp, having a pronounced $1/f$ characteristic[41,42] that masks power in the high-frequency range, we have baselined (z-score) spectra to the pre-stimulus period[43]. The TF power spectrum of the occipital signal over the Oz electrode was estimated using STFT, CWT, and ASLT (Fig. 4a). The STFT window was chosen to optimize the representation in the γ-range (>30 Hz), whereas the number of cycles for the CWT was chosen to maximize temporal resolution. The STFT provided a poor resolution in the low frequency range (Fig. 4a, top), whereas the CWT showed good temporal resolution but poor frequency resolution for higher frequencies (Fig. 4a, middle). On the same data, the ASLT provided the sharpest TF resolution across the whole frequency range (Fig. 4a, bottom).

We next zoomed in on the γ-frequency range, which poses particular challenges for TF analysis[44–47]. The Fourier window (Fig. 4b, top row) and the number of wavelet cycles (Fig. 4b, middle row) were varied to optimize the temporal (left) or frequency (right) precision, or a tradeoff between the two (middle). SLs (Fig. 4b, bottom) shared the major features with the other representations but provided TF details that could not be simultaneously resolved by any of the latter.

In vivo electrophysiology signals are recorded at much higher sampling rates than EEG (32 kHz compared to 1 kHz), offering

the opportunity to observe oscillation bursts with higher temporal precision in local field potentials (LFPs) than in EEG. We next focused on LFPs recorded from mouse visual cortex during presentation of drifting sinusoidal gratings (see "Methods"). LFPs suffer from the $1/f$ issue significantly less than EEG and therefore baselining is typically not necessary for their analysis. We computed the TF representation of an LFP signal using the STFT (Fig. 4c, top), CWT (Fig. 4c, bottom), and ASLT (Fig. 4c, middle) around the presentation of the visual stimulus (drifting grating at 45°) and averaged it across ten presentations (trials). As was the case for EEG data, ASLs provided a sharp TF representation across the entire analyzed spectrum. They revealed 45 Hz γ-bursts induced by the passage of the grating through the receptive fields of cortical neurons[48] and resolved many details in both the low and high frequency range.

The capability of SLs was, however, revealed when we zoomed in on a compact γ-burst induced by the passage of the grating (see Fig. 4d). SLs were computed with a base cycle $c_1 = 2$, to increase temporal resolution, and we used a fixed multiplicative order of 7 (SLT). The SLT provided fine temporal and frequency details, whose presence in the signal was validated by computing the local CWT optimized for time ($c = 2$), frequency ($c = 11$), or a tradeoff between time and frequency ($c = 6$). The components seen in the SL representation could be inferred from these multiple wavelet representations but none of the latter was able to simultaneously reveal all the TF details (Fig. 4d).

We further explored a TF detail revealed by SLs (Fig. 4d left bottom and Fig. 4e), composed of a lower ongoing rhythm (LOR) at ~17.5 Hz, two time neighboring packets at 24.5 Hz (NP1 and

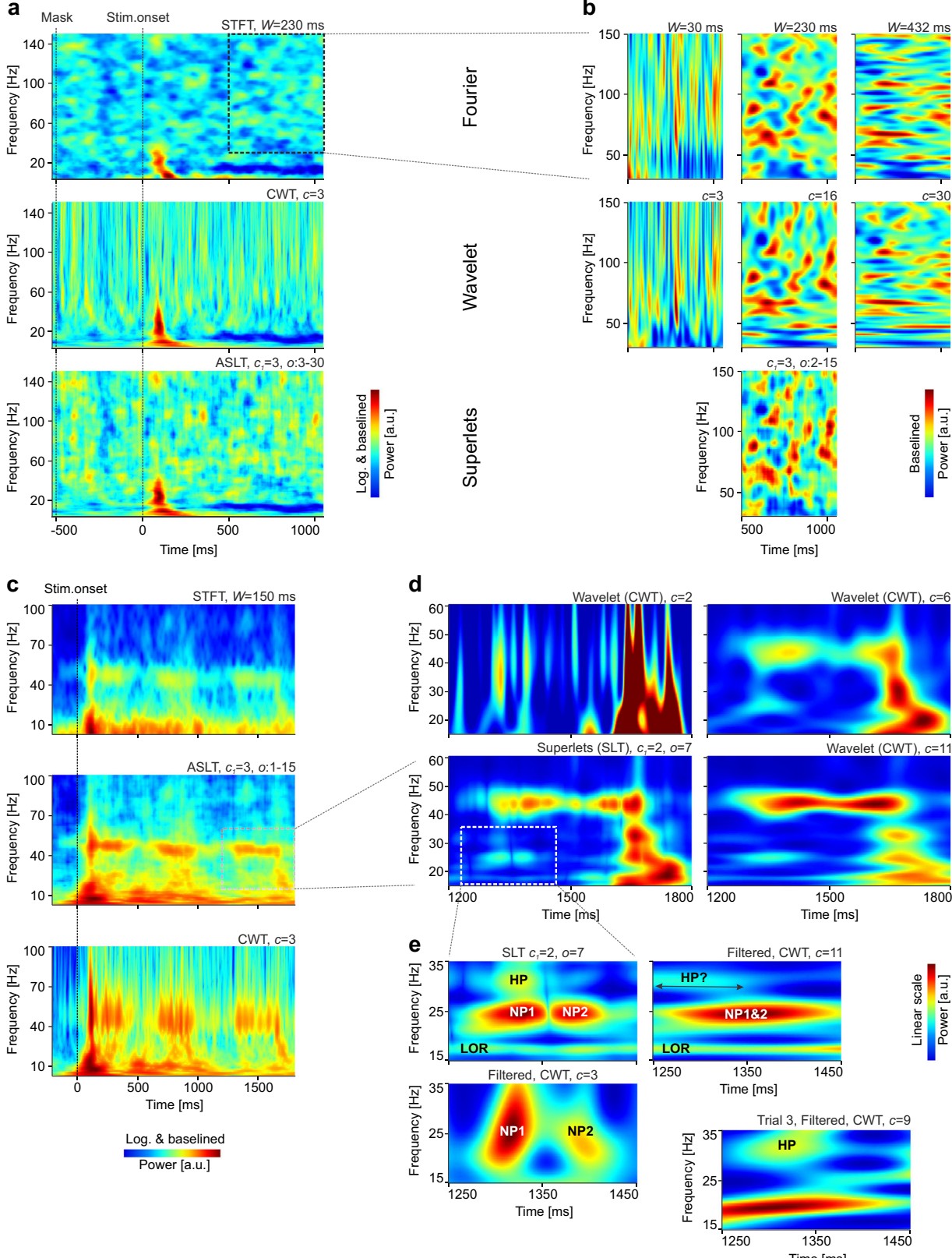

NP2), and one higher frequency packet at ~31 Hz (HP) (Fig. 4e, top left). To determine whether these features were actually present in the signal over the ten trials, we narrow-band filtered the signal (bidirectional IIR, order 3, bandpass 10–40 Hz) to remove frequency contamination plaguing the wavelet estimates in the γ-range. This enabled us to validate the presence of the TF

packets using narrow ($c = 3$) and wider ($c = 11$) wavelets (Fig. 4e bottom left and top right). However, although the frequency of HP could be identified, its clear temporal location could not be established with the CWT, irrespective of the parameters of the wavelet (Fig. 4e top right). We suspected that this may originate from averaging over ten trials such that time/frequency

**Fig. 4 Time-frequency analysis of EEG and acute electrophysiology signals.** Data was recorded over occipital electrode Oz for EEG (**a**, **b**) and in mouse visual cortex for acute electrophysiology (**c**–**e**). **a** Global time-frequency EEG power spectrum around stimulus onset computed using Fourier analysis (STFT; top), wavelets (CWT; middle), and adaptive additive superlets (ASLT; bottom). **b** Zoom-in analysis over the γ-frequency band (30–150 Hz) of data from **a** using STFT with various windows (top), CWT with different number of Morlet cycles (middle), and adaptive multiplicative superlets (bottom). Representations in **a** were first logarithmized (base 10) and both those in **a** and **b** were baselined (z-score) to 500 ms pre-stimulus period. Representations are averages across 61 trials. **c** Fourier (STFT; top), adaptive multiplicative superlets (ASLT; middle), and wavelet power spectra (CWT; bottom) around stimulus onset on mouse electrophysiology data. Representations were first logarithmized (base 10) and then baselined (z-score) to pre-stimulus period. **d** Zoom-in on a γ-burst from data in **c**, induced by the passage of the grating through the receptive field of cortical neurons. The SLT used multiplicative superlets of order 7 and $c_1 = 2$, optimized to provide high temporal and frequency resolution (bottom left). By comparison, individual wavelets optimized for time (top left), frequency (bottom right) or a compromise between the two (top right) cannot reveal all the details evidenced by the superlet. **e** Further zoom-in on a detail from data in **d** provided by the superlet (top left) reveals two time neighboring packets (NP1 and NP2), a higher frequency packet (HP), and a lower ongoing rhythm (LOR). Tuned wavelets on 10–40 Hz band-passed data indicate roughly the presence of the temporal (left bottom) and frequency (top right) components. The location of HP cannot be determined by wavelet analysis in the average time-frequency spectrum, but is recovered by single-trial analysis, indicating that superlets can correctly reveal very fine time-frequency details, which are smeared out in the average spectra by the other methods. Absolute power shown (linear scale, no baselining) in **d** and **e**. Representations in **c** and **d** are averages across ten trials.

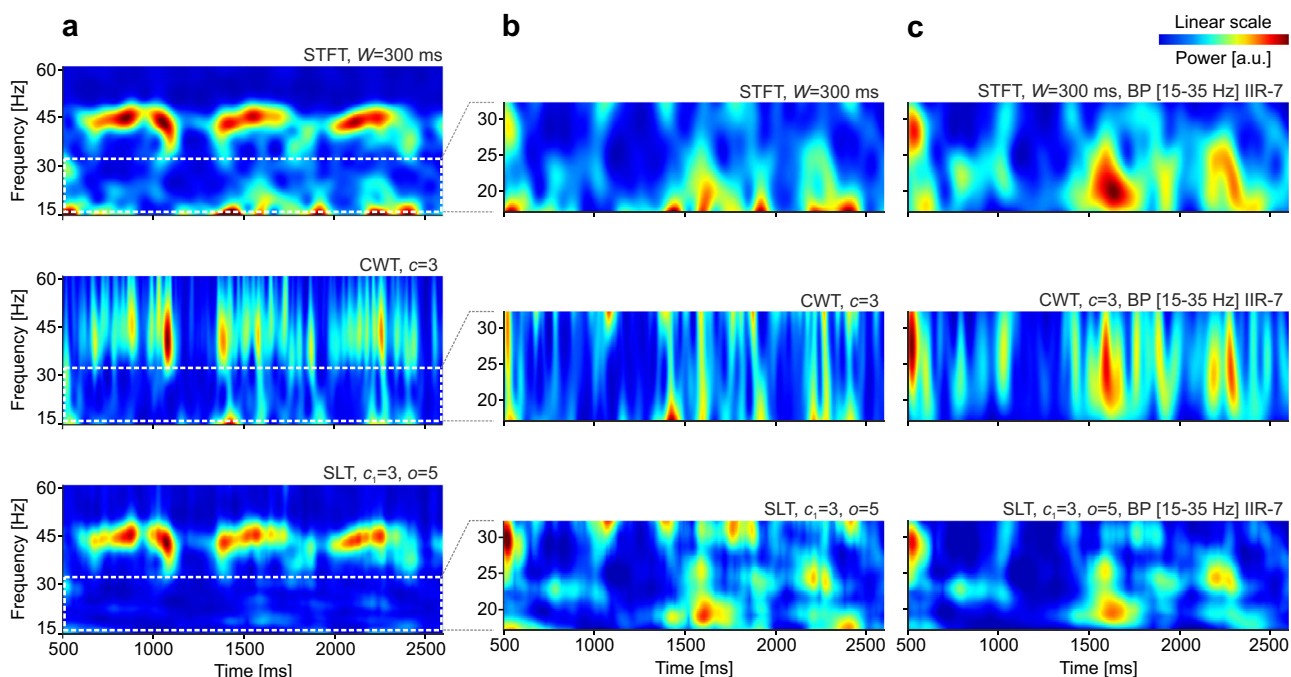

**Fig. 5 Effect of averaging time-frequency spectra. a** Fourier (STFT; top), wavelet power spectra (CWT; middle), multiplicative superlets (SLT; bottom) computed for the β- and low-γ-bands (15–60 Hz), and averaged across ten trials. Stimulus onset was excluded from analysis to focus on the fast oscillations induced by the passage of the drifting grating. **b** Zoom-in on the β-band. **c** Same as in **b** but computed after the signal was bandpass filtered in 15–35 Hz using a bidirectional Butterworth IIR filter of order 7. Absolute power shown (no baselining, linear scale).

smearing of the long/short CWT could hide this detail. Indeed, we found that HP was expressed clearly in at least one of the trials in the set (Fig. 4e bottom right). Thus, the packets in the TF detail revealed by SLs were actual features in the signal and, in addition, the TF concentration provided by SLs was able to reveal bursts expressed at single-trial level. The latter could not be identified by other methods because they were averaged out due to the time/frequency smearing in the wavelet or Fourier representations (see also Supplementary Fig. 2).

One of the hardest problems in the TF analysis of brain signals is to resolve weaker neighboring rhythms when strong, dominating oscillation bursts are present in a certain frequency band. For example, oscillations may occur simultaneously in the β- and γ-bands, and these are often hard to distinguish[45]. Figure 5 shows an example analysis on the same data from Fig. 4c–e but in response to an orthogonal grating direction (135°) and focused on the post-stimulus onset part of the response. All methods were able to reveal the strong γ-bursts occurring in response to the drifting grating

(Fig. 5a). The STFT displayed strong leaked-in power from the lower frequencies (Fig. 5a top), whereas the CWT had markedly poor frequency resolution (Fig. 5a middle). As before, the SLT provided good TF resolution (Fig. 5a bottom). Importantly, the latter also suggested the presence of TF structure in the β-band (12–30 Hz).

When we zoomed in on the β-band, the SLT revealed a rich TF structure with multiple packets located at various frequencies and temporal offsets (Fig. 5b bottom). By contrast, the STFT (Fig. 5b top) and CWT (Fig. 5b middle) were unable to properly resolve this frequency band, suffering from power leakage from below and above the band. When the signal was narrow-band filtered in the 15–35 Hz range, the representation in the STFT (Fig. 5c top) and the CWT (Fig. 5c middle) changed significantly, and became more similar to the SLT, roughly revealing the same TF structure. By contrast, the SLT maintained a similar representation after filtering, implying that it correctly captured the TF structure of the β-band, with minimal interference from bursts in the frequency bands below and above it.

**Comparison to MMCE**. An established TF super-resolution method is the Fourier-based MMCE[35]. The latter employs a set of windows with increasing size and combines the resulting spectrograms geometrically. We next compared how SLs faired with respect to the MMCE method. Figure 6a, b depicts a comparison of the two methods when applied to EEG data (same experiment as in Fig. 4a, b, different subject). Signals were aligned to either to the presentation of the visual stimulus (onset) or the response of the subject (button press) and analyses were grouped across three conditions, as a function of the response of the subject (object seen, uncertain, nothing seen)[40]. Unlike in the case of SLs, the choice of parameters for MMCE is not obvious. If the MMCE windows were matched with the extent of the shortest and longest wavelets in the ASLT, the MMCE representation would suffer from excessive time smearing (see Supplementary Fig. 3). We therefore experimented with various parameters that yielded a reasonably sharp MMCE representation: a set of seven windows, spanning from 30 to 700 ms. On the data aligned to stimulus onset, the MMCE (Fig. 6a) provided a less sharp representation compared to the ASLT (Fig. 6b). Although frequency resolution remained constant, relative time resolution degraded with increasing frequency. Also, a marked effect was the masking/ "dilution" of high frequency packets when powerful low-frequency components were present. SLs did not suffer from these problems and revealed a rich TF structure, independently of how the data were aligned (Fig. 6b). All analyses revealed interesting associations between α-, β-, and γ-activity, and the experimental condition, but SLs provided the sharpest picture. The increase in γ-bursts expression and in their frequency was associated with the perceptual process, whereas increased ongoing β was more sustained in association to response certainty in this particular subject. α was in general suppressed during stimulus processing.

We next compared the two methods on electrophysiology data. Figure 6c–e shows results on the same data as in Fig. 5, but for a larger time and frequency range. As was the case for EEG data, the MMCE displayed a progressive "dilution" of oscillation packets with increasing frequency (Fig. 6c). By contrast, ASLs displayed a milder "dilution" effect and provided robust details across the explored frequency range (Fig. 6d).

Depending on the slope of the order increase function in the ASLT, one can obtain representations that provide less, the same, or more dilution compared to MMCE. By contrast, the SLT does not display any dilution. To investigate the "dilution" effect, we optimized both the MMCE and SLT to estimate limited γ-bursts induced by the drifting grating (Fig. 6c–e, inset 1). In this limited range, both methods provided rather identical results, thus validating the fact that SLs provide locally consistent results with established methods. However, when the analysis window was expanded to incorporate a progressively larger frequency range (Fig. 6c–e, insets 2 and 3), the MMCE displayed a marked dilution of the higher frequencies while the SLT did not. The dilution effect is discussed in detail in Supplementary Information—"III. Superlets and redundancy suppression: towards multiscale TFRs/Interpretation of superlets and relation to MMCE."

**Detection of oscillation bursts expressed in single trials**. During cognitive processes, it is expected that γ-bursts are scattered in time and frequency in a single-trial-dependent manner. Therefore, we evaluated the ability of SLs to discover γ-bursts expressed in single trials. Three 11-cycle packets of 40, 80, and 120 Hz were inserted into only one of the 84 trials (Fig. 7a) recorded over the Pz electrode in condition "Nothing" of the dataset shown in Fig. 6a, b. We computed the averaged TF spectrum over the 84

trials using STFT (Fig. 7b), CWT (Fig. 7c), MMCE (Fig. 7d), and multiplicative SLT (Fig. 7e). SLs were able to robustly detect the presence of all three packets. The latter also revealed a rich TF structure in the upper frequencies (>50 Hz), which the other methods mostly missed.

**Comparison to other high-resolution methods**. We next evaluated how the high-resolution methods, based on the WVD or the directionally smoothed WVD, fared on EEG and electrophysiology data. As their representations can also contain negative values, applying the log was not an option. Therefore, all results were computed with linear scales. As before, representations for EEG data were baselined, to compensate for its pronounced 1/f property. Figure 8 shows the results, on the same data from Fig. 4a, c, computed with the WVD-based methods. In the WVD, cross-terms contaminated the entire TF landscape (Fig. 8a, d). The CW (Fig. 8b, e) and BJ (Fig. 8c, f) methods performed better, mostly revealing structure in the low frequencies. However, all these methods performed poorly overall, on both EEG (Fig. 8a–c) and electrophysiology (Fig. 8d–f) data. Well-known phenomena, such as entrainment of γ-bursts by passage of drifting gratings, are difficult to identify.

**Detecting packets embedded in noise and measuring resolution**. Previous results, especially those in Fig. 7, suggest that the SLT and ASLT are useful as detection tools when the data are noisy. It would therefore be important to have an objective measurement of how well different methods perform in detecting various types of target signals. We created a setup where a unit amplitude packet (a brief sine wave or a Gaussian atom—each spanning ten cycles) was progressively buried in white noise of increasing amplitude. To mimic realistic conditions, we generated datasets consisting of 50 trials with noise, and inserted the packet in only ten of these. We then considered a TF mask around the target signal and defined a "detection score" as the fraction of values within the mask which fell above the 95th percentile of the distribution of all values in the representation (Fig. 9a top). The logic was that the signal is detectable as long as its corresponding representation falls within the upper range of the global distribution. The detection score is always bounded to 1 and decreases as the detectability of the signal becomes increasingly impaired. We computed the detection score for all methods and for noise levels from 0.25 to 5 (5× larger than the amplitude of the signal), for both the sine packet (Fig. 9b) and the Gaussian atom (Fig. 9c). For each noise level, we analyzed 25 independently generated datasets. While the isolated atom was best detected by CW and BJ in low noise levels, both the sine and the atom were better detected by the ASLT as soon a significant amount of noise was added. The ASLT was also very close to CW and BJ on the atom in low noise conditions (Fig. 9c).

The quality of TF representations is sometimes evaluated using the "uncertainty product" (UCP), which localizes the representation in time and frequency, by computing the first and second moments on the marginals (for details, see Supplementary Information—"IV. Resolution of TFR/TSR representations"). It can be argued[49] that the UCP is not a proper measure of a method's resolution because resolution is related to the ability to distinguish among multiple, close components in a representation[50]. The UCP is not applicable on multi-component representations if the first and second moments become statistically flawed[51].

Approaches to measure resolution are usually based on the shape of the TF landscape when multiple, known signal components are added[50,51]. Here we took a similar approach and considered two fixed frequency signal components (sine

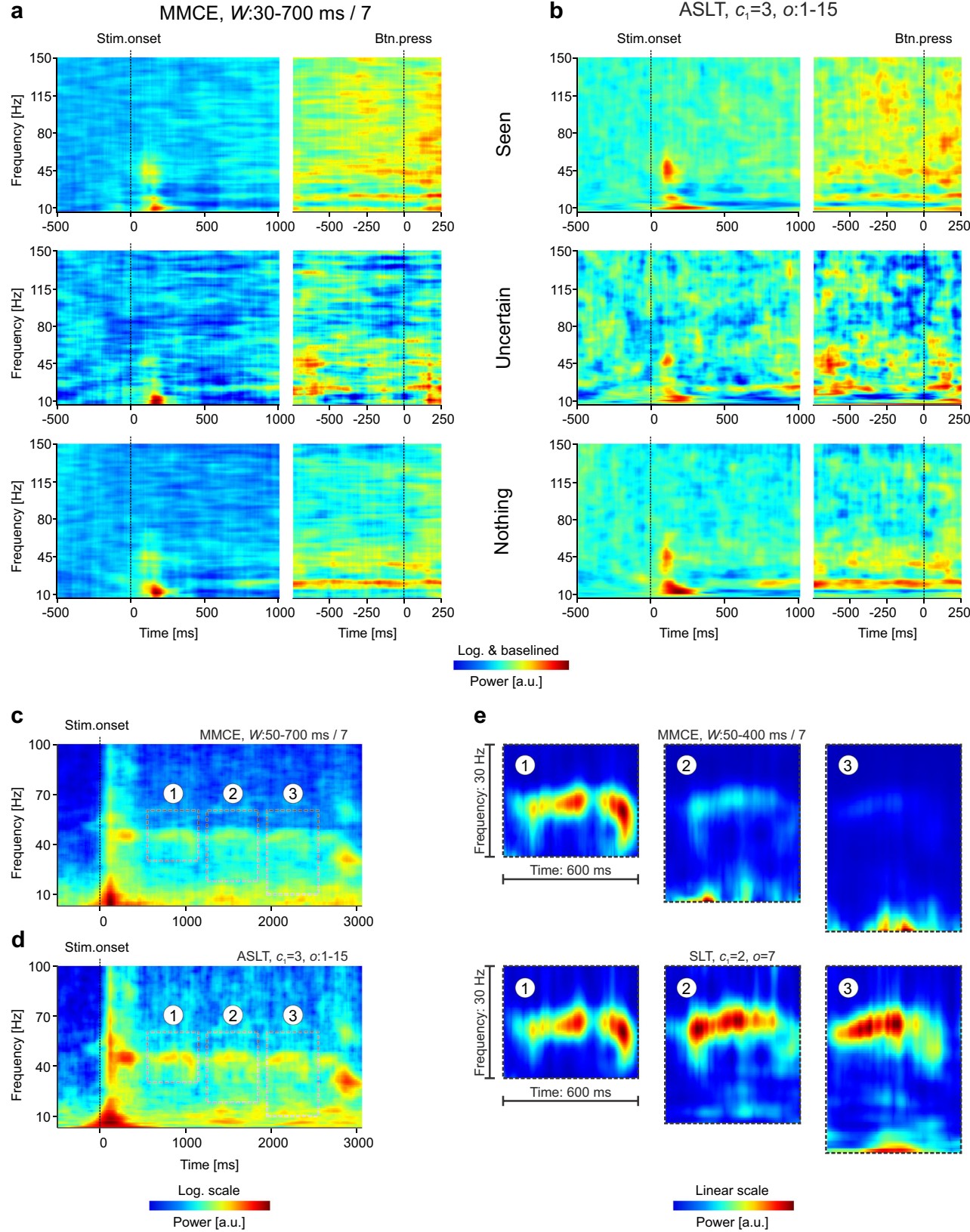

packet and Gaussian atom) that were brought progressively closer in time. Time resolution was measured as the fraction of "empty space" in the cross-section of the representation at the known frequency of the components (Fig. 9d, g top, horizontal line and Fig. 9d, g bottom, red profile line). Frequency resolution was measured by considering the inverse of the SD of frequency spread at the location of the known components (Fig. 9d, g top, vertical line). TF resolution was then defined as the product of the two measures. A high value of this resolution measure corresponds to a superior method, with the ability to better

**Fig. 6 Comparison between minimum mean cross-entropy (MMCE) and superlets. a** MMCE using 7 windows that span 30–700 ms on EEG data recorded by 15 occipital electrodes. **b** Adaptive multiplicative superlets on data from **a**, with orders spanning 1–15 and $c_1 = 3$ cycles. For each analysis, signals were aligned to stimulus onset (left) or subject response (right). Representations were first logarithmized (base 10) and then baselined (z-score) to the pre-stimulus period. Color scales were maintained identical across conditions (along columns) to facilitate comparisons. Number of analyzed trials per condition were: 84 for "seen," 20 for "uncertain," 100 for "nothing." **c** MMCE using 7 windows that span 50–700 ms on in vivo electrophysiology data (LFP, 10 trials). **d** Adaptive multiplicative superlets on the data from **c**, with orders spanning 1–15 and $c_1 = 3$ cycles. **e** Analyses on restricted windows with various locations and sizes (see insets in **c** and **d**) using MMCE (top) and superlets (bottom). The power scale is logarithmic in **c** and **d**, to facilitate exploration of a large spectral range, while it is linear in **e**, to enable precise quantitative comparison.

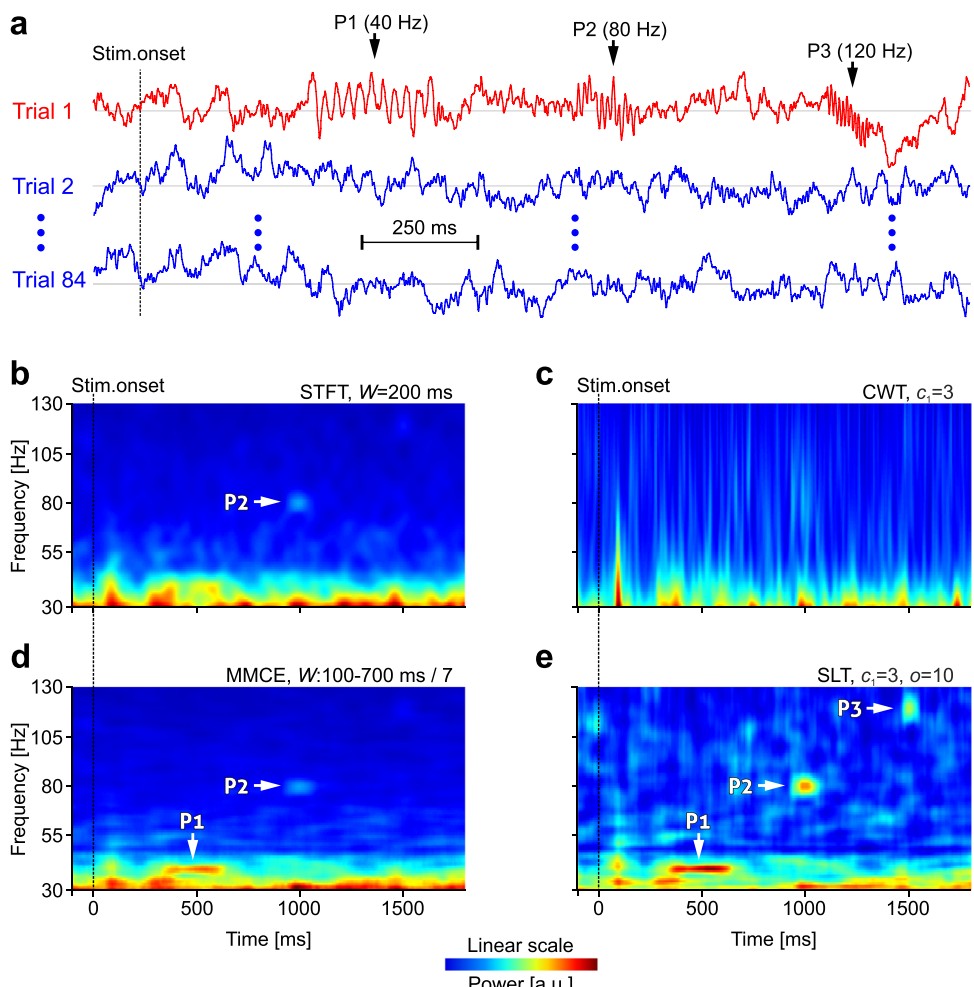

**Fig. 7 Detection of single-trial γ-bursts (packets). a** Three target packets extending 11 cycles and with frequency 40 (P1), 80 (P2), and 120 Hz (P3) were inserted into a single trial of a set of 84 trials of EEG recordings (Pz electrode). Real data shown. Averaged time-frequency spectrum over the 84 trials using: STFT (**b**), CWT (**c**), MMCE (**d**), multiplicative SLT (**e**). Arrows show the time-frequency location of detected target packets.

distinguish neighboring oscillation packets. We computed the resolution of various representations, based on STFT, CWT, MMCE, SLT, WVD, CW, and BJ. The window for STFT and the number of cycles for CWT were matched to the extent of each packet / atom. In addition, the parameters of MMCE and SLT were matched at the frequency of the packets/atoms. For two sine packets, the time resolution of SLT and MMCE was unsurpassed (Fig. 9e), whereas the TF resolution was even more prominently above that of the other methods (Fig. 9f). By contrast, due to strong cross-terms, the resolution of the WVD was zero. The other members of this family, CW and BJ, performed rather poorly, due to their directional smoothing, which decreased temporal resolution. In the case of the two Gaussian atoms, which were already maximally concentrated, the SLT, MMCE, CW, and BJ exhibited the best time (Fig. 9h) and TF (Fig. 9i) resolution for

the difficult case, when the atoms were very close. The CW and BJ quickly lost this advantage as soon as the packets were pulled apart, whereas the STFT and CWT gained a slight advantage over MMCE and SLT for larger separation between atoms. These results indicate that, for sinusoidal packets, where the SLT and MMCE can gain precision over single wavelet/window transforms, these methods are unsurpassed. For Gaussian atoms, the SLT and MMCE behave optimally at small separation, and very well at large separations.

When a single frequency or a limited range of frequencies is present in the data, the windows of the MMCE can be matched such that the shortest and longest window match the extent of the shortest and longest SL wavelets at that frequency, as shown in Fig. 9d–i. In such cases, the TF resolution of the two methods is very similar (Fig. 9f, i). It should be noted however that this

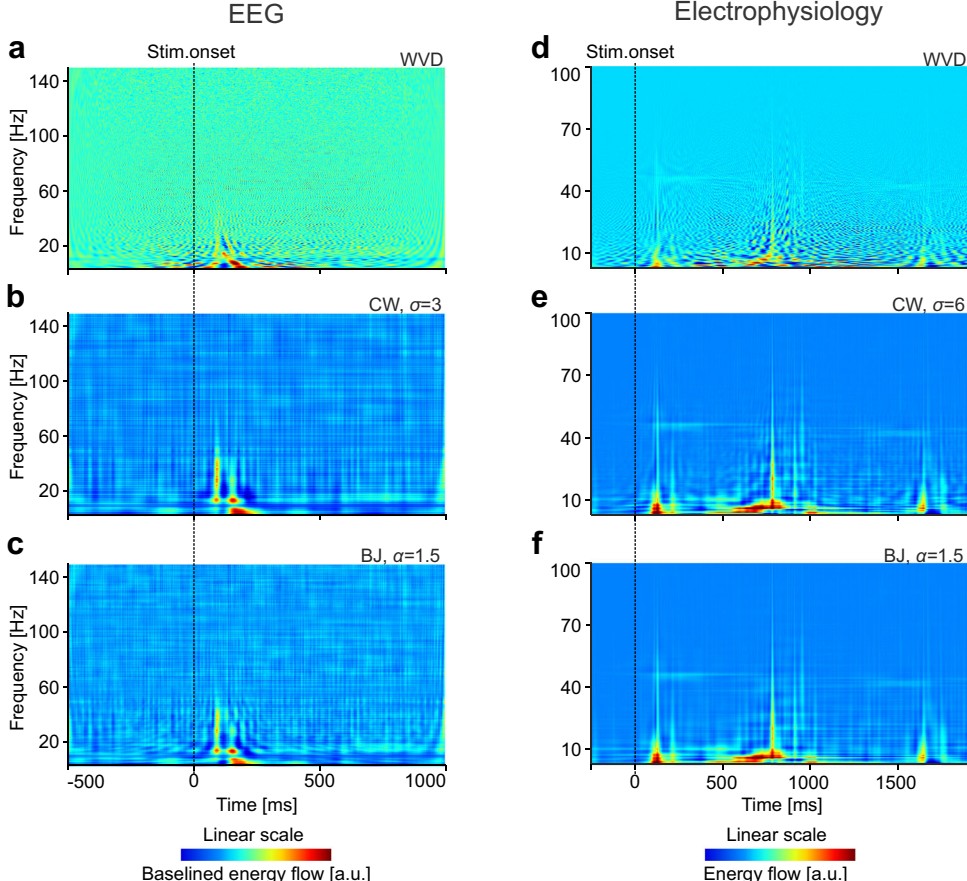

**Fig. 8 Time-frequency energy (flow) representations with Wigner–Ville Distribution (WVD) based methods. a–c** WVD, Choi–Williams (CW), and Born–Jordan (BJ) representations for the EEG data from Fig. 4a. **d–f** The same representations computed for the electrophysiology data from Fig. 4c. Baselining to pre-stimulus period was used for representations in **a–c** but not for those in **d–f**.

window matching is not usually helpful on real data, especially when the signal contains a wide range of frequencies that are explored.

**Validation using optogenetics**. Although different methods can reveal a rich TF landscape in brain signals, it is difficult to know the ground truth and therefore hard to distinguish which method performs better. To overcome this limitation, we used LFP data recorded in the visual cortex of transgenic animals where we applied periodic optogenetic stimulation using blue light. Figure 10 shows results obtained on data recorded in a Thy1-ChR2-YFP mouse, where blue light drives mainly principal, excitatory neurons[52–55]. Optogenetic stimulation was delivered by rectangular light pulses with a duty cycle of 50% by using a mechanical shutter and a blue laser (see "Methods"). Stimulation frequency covered the 5–105 Hz range, with a step of 10 Hz. We quantified the dominant power and frequency of the response in the LFP by computing baselined (Z-scored)[43] Welch spectra across all channels that had responses to light (23 out of 32) and evaluating the peak power and its corresponding frequency (Fig. 10a, "Peak pow. increase" and "Frequency @ peak"). In addition, we isolated multi-unit activity (MUA) across signals recorded by all these electrodes and computed the average firing rates (Fig. 10a, "MUA firing rate"). Periodic optogenetic stimulation entrained the cortical circuit into oscillations whose power increase peaked in the γ-range (30–80 Hz), matching previous reports[56,57]. Interestingly, at low stimulation frequency the cortex engages into induced γ-oscillations at significantly higher frequency than that of the

stimulating signal. At 5 Hz drive, we observed ~53–55 Hz and vigorous spiking (Fig. 10a, "Frequency" and "Firing rate"), as shown before by Tiesinga[58]. As stimulation frequency was increased, the cortex tended to engage into oscillations with a frequency closer and closer to that of the stimulus, switching from an induced regime to an evoked (locked) regime, where the responses followed the stimulation pulses. We evaluated the performance of the high-resolution methods on both these regimes.

To investigate the induced regime, we considered LFP signal from the deepest electrode (furthest away from photo-stimulation) during 5 Hz stimulation. Figure 10b shows the TF spectra computed with the SLT (top), the window-matched MMCE (second row), CW (third row), and BJ (bottom). The SLT displayed a rich structure in the γ-band and showed clear bursting at ~55 Hz. We focused on a clear, strong γ-burst (packet), located during the second cycle of stimulation (200–400 ms) and which could not be properly resolved by the MMCE, CW, or BJ (Fig. 10b, red arrow). To determine the veracity of this burst, we computed the autocorrelation of the signal using scaled correlation analysis (SCA), a time-domain method able to isolate correlations on fast timescales[59]. SCA revealed a clear oscillatory modulation at the target location (Fig. 10c top), whose period matched the period of the burst (Fig. 10c bottom). To further obtain an independent confirmation that the observed γ-burst is a genuine phenomenon, rooted in cortical dynamics, we extracted the MUA from the same channel. The peri-stimulus time histogram indeed revealed robust spiking following the

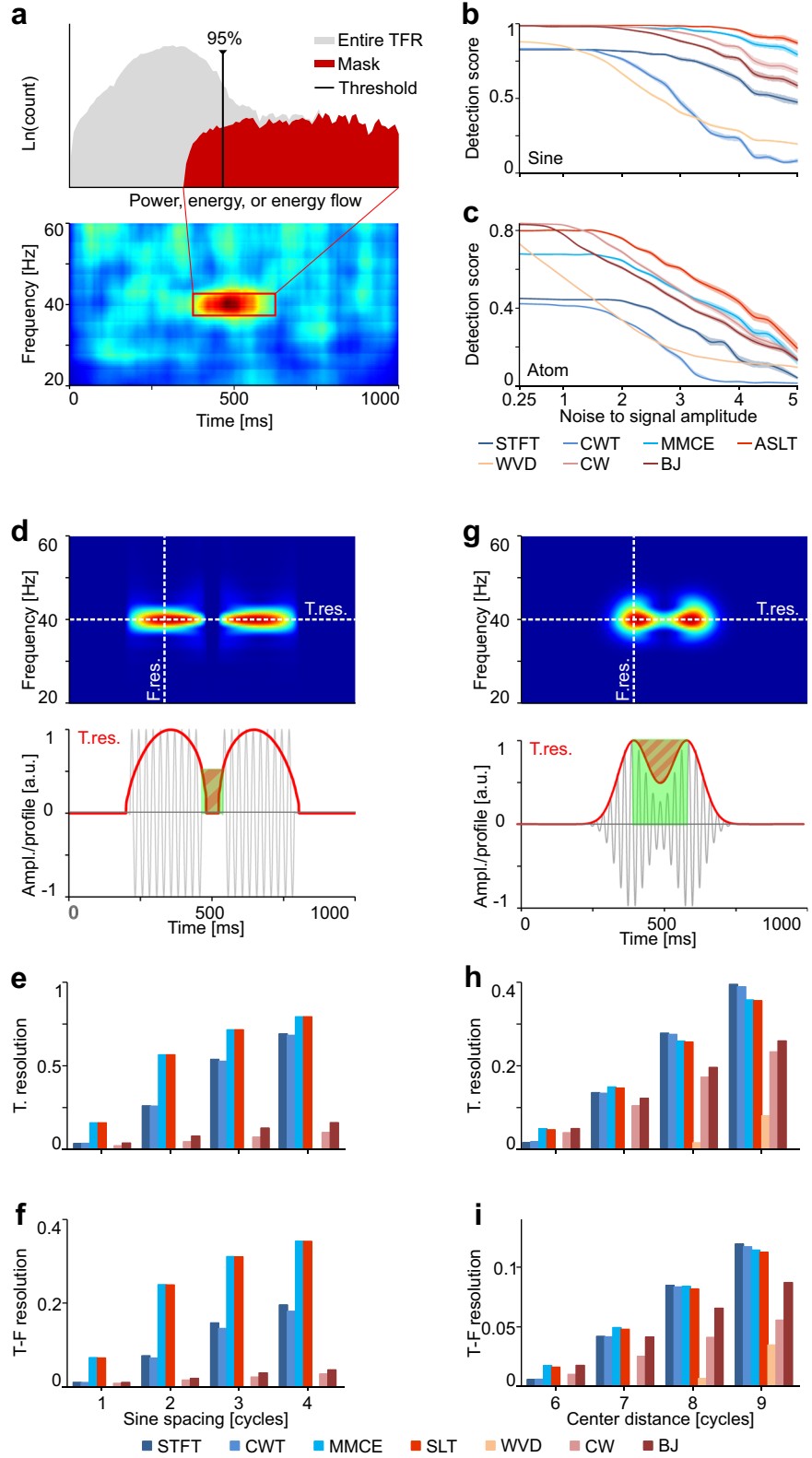

optogenetic stimulation (Fig. 10d top). Using the binary version of SCA[59], we computed the autocorrelation structure of the spike train and found that it displayed oscillatory modulation with the period matching the frequency of the burst observed by the SLT (Fig. 10d bottom). Given the tonic optogenetic drive of the excitatory population during the target period, it is likely that the observed γ-oscillations reflect a push-pull Pyramidal-INterneuron

Gamma interaction[48,60,61] between excitatory and inhibitory cells, whereby inhibition balances out excitation[62] (see Tiesinga and Sejnowski[63]). Thus, the SLT was able to identify a genuine γ-burst, originating from circuit dynamics, which the other methods largely missed.

To investigate the evoked (locked) regime, we considered the LFP recorded by a more superficial channel at 65 Hz rhythmic

**Fig. 9 Detection of isolated signal components under noisy conditions and resolution of various representations on sine packets and Gaussian atoms.**
**a** Procedure for computing the detection score based on the fraction of values in the target window that reside above the 95th percentile of the distribution of values of the entire representation. The natural logarithm (Ln) was applied to the distributions for visualization purposes only. **b** Detection score for a unit amplitude sine packet of 8 cycles @ 40 Hz, as a function of the amplitude of added white noise. **c** Same as in **b** but for a unit amplitude Gaussian atom of 10 cycles @ 40 Hz. Results in **b** and **c** were computed across 25 datasets for each noise level. Each dataset consisted of 50 trials with independent noise instantiations and the target packet was inserted in only ten trials. Error bands are SEM. **d** Extraction of time and frequency profiles from the representation (top) and measurement of time resolution (bottom) for two sine packets. **e** Temporal resolution for different representations of the signal from **d**. **f** Time-frequency resolution for the signal from **d**. **g**, **h**, **i** The same as in **d**, **e**, **f**, respectively, but for two Gaussian atoms. Time resolution was measured as the fraction of "empty space" in the cross-section of the representation, i.e., area of the shaded region divided by the area of the green box. The green box is defined as the rectangle that spans the space between the edges of the sine packets or between the peaks of the Gaussians, while its height is given by the value of the cross-section of the representation at the sine ending points or the peaks of the Gaussians, respectively.

optogenetic stimulation. At this stimulation frequency, the cortex locked more reliably to the stimulus (Fig. 10a) but in some cases this locking was transient. Such an example is shown in Fig. 10e–i.

All methods were able to reveal the transient locking to the stimulus at 65 Hz (Fig. 10e), revealed also by bandpass filtering the signal (Fig. 10f top) in the narrow 60–70 Hz band (Fig. 10f middle). In addition, the SLT revealed an interesting phenomenon that could not be resolved by the other methods: fine modulation at 130 Hz (double of stimulation frequency), with marked bursting, especially at the beginning and in the middle of the trial (Fig. 10e top, red arrows). Bandpass filtering (110–140 Hz; Fig. 10f bottom) and SCA (Fig. 10g) confirmed the existence of these high-frequency bursts.

To confirm the locking of the LFP to the light stimulus, we computed the event-related potentials (ERPs) aligned to the rising front of the light pulse and during periods with high (Fig. 10e, periods 1 and 3) and low (Fig. 10e, period 2) power in the SLT. Indeed, high power was correlated to high ERP amplitude and low power to low ERP amplitude (Fig. 10h), showing that the phenomenon of transient coupling revealed by the SLT was genuine. Moreover, we aligned the signal during the early high-frequency burst in Fig. 10e (first red arrow), with the light stimulation and found that, indeed, the LFP was locked to double the stimulation frequency (Fig. 10i). The ability of SLs to identify such high-frequency transients while simultaneously resolving the lower frequencies was unmatched by the other methods on these data.

Finally, we investigated the ability of high-resolution methods to detect very high-frequency bursts under a variety of optogenetic stimulation conditions. A gallery of examples is shown in Supplementary Figs. 13–15. Results indicate that SLs reliably detect high-frequency bursts, sometimes exceeding 200 Hz. Such bursts can occur under different stimulation conditions, even when the optogenetic drive is slow (Supplementary Fig. 14). Furthermore, an example of coupling at double of the stimulation frequency is shown in Supplementary Fig. 15. The other high-resolution methods performed poorly by comparison.

## Discussion
Increasing the resolution of joint TF estimation, especially for the case of non-stationary signals, has been a very active field of research in the past decades[11,15,32,64]. Notable techniques include the Cohen class methods[36,37] or the Fourier-based MMCE[30] and its Probabilistic Latent Component Analysis derivative[31]. SLs extend these efforts by using a simple wavelet-based approach. They take advantage of multiple estimates at a range of temporal resolutions and frequency bandwidths, which are combined geometrically (optimal entropic criterion) to evaluate the temporal and frequency location of finite oscillation packets.

In optics, the term super-resolution refers to the ability to resolve details beyond the diffraction (Rayleigh) limit[33] by taking advantage of multiple measurements[34]. Here we refer to super-resolution as the ability to resolve the joint TF density better than it is possible with a single estimate[32]. SLs provide super-resolution in the TF sense, even if their frequency resolution approaches (but does not exceed) the theoretical Rayleigh frequency. For frequency super-resolution, when time is irrelevant, other techniques are applicable, e.g., based on model fitting[65], polyphase analysis filter banks[66], Pisarenko harmonic decomposition[67], or multiple signal classification (MUSIC)[68]. These frequency super-resolution techniques ignore the temporal component and focus on the frequency dimension only.

SLs use the GM across a set of wavelet responses to sharpen TF localization. Intuitively, the GM multiplicatively combines responses with high temporal precision with those with high-frequency precision[29]. For example, if a narrow-bandwidth wavelet (many cycles) detects a narrow frequency component, this will be vetoed out in time if the short wavelet at a certain location has a low response, and vice versa. Quantitatively, it has been shown that using the GM to combine individual measurements improves the estimate of the joint TF density and is optimal in a cross-entropy sense[30,35]. This property is not shared by the arithmetic mean (see Supplementary Fig. 4), which corresponds to the minimum mean-squared solution[30].

The frequency resolution limit for a finite oscillation packet depends on the packet's duration but temporal resolution can be increased by increasing sampling rate. Typically, LFPs are obtained by low-pass filtering (@300 Hz) the electrophysiology signal sampled at much higher rates (32–50 kHz) and then downsampling the signal. When using SLs, one should keep a high sampling rate after downsampling (e.g., 2–4 kHz) to enable the method to resolve fine TF details (see Figs. 4d, 5, and 6e).

Cortical responses exhibit a significant trial-to-trial variability[69]. Therefore, results are typically averaged across multiple trials. For TF analysis this can pose significant problems[44,70,71], for several reasons. First, perceptual processes may be supported by high-frequency γ-bursts whose expression is not necessarily locked to the external events available for aligning the analysis (stimulus onset, button press etc.). As a result, γ-packets may be scattered throughout the TF spectrum and will not sum up coherently in the average. Second, due to the TF uncertainty, isolated packets can be masked out by strong neighboring packets whose estimate leaks over the target's representation. Because they concentrate the joint TF estimate in each individual trial and in a frequency-specific manner, SLs provide a sharp image of the TF landscape, revealing oscillation packets that may remain hidden with other estimation methods.

Our results indicate that, for averaged TF spectra, traditional methods (STFT and CWT) may fail to reveal the true TF structure within a certain band if strong spectral neighbors exist, as the representation of the latter leaks into the band of interest, compromising its estimation. Powerful oscillation packets can in principle be detected by many methods. However, estimating the surrounding, weaker packets, turns out to be difficult for classical

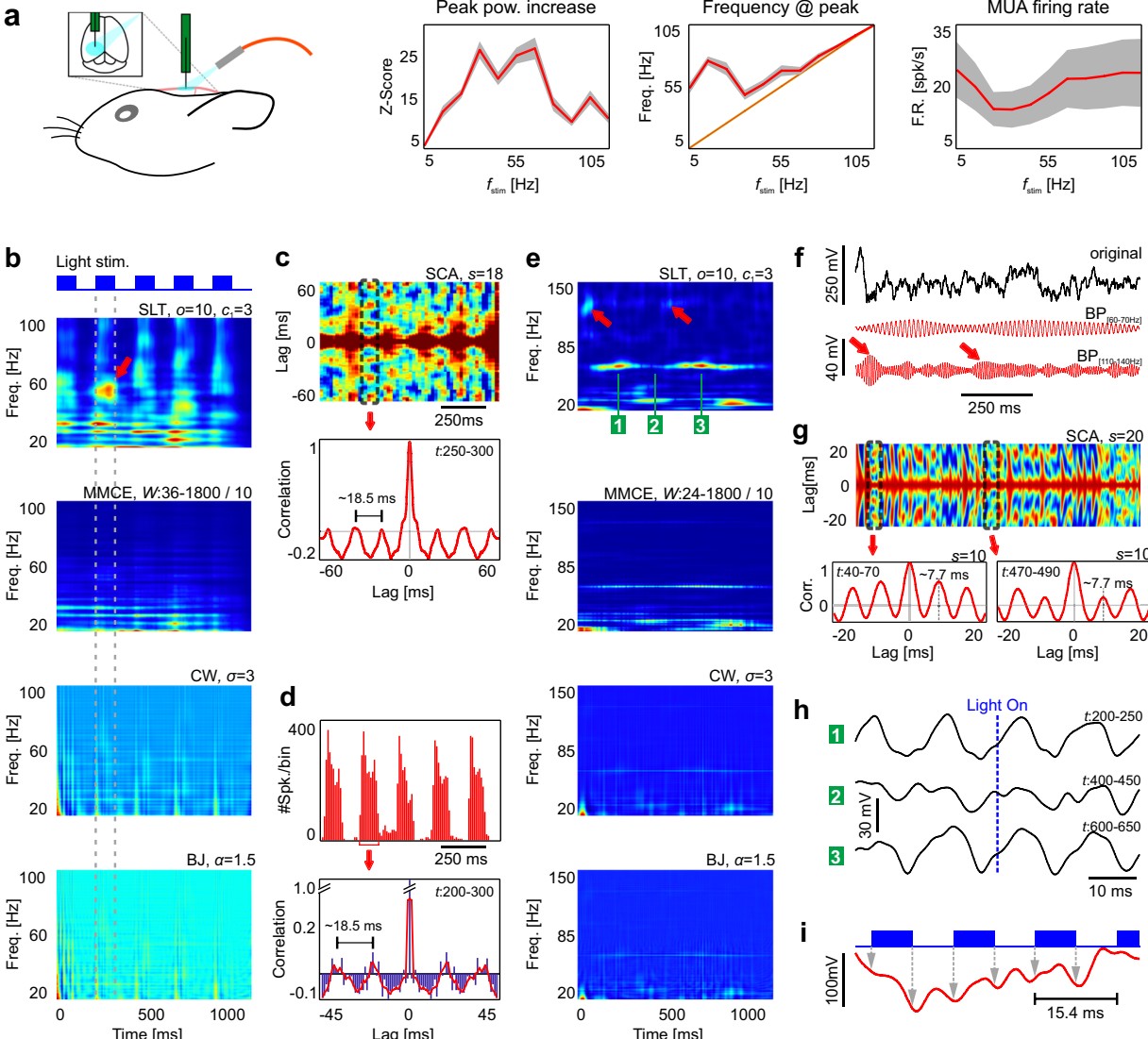

**Fig. 10 Optogenetic validation of high-frequency burst detection on in vivo electrophysiology data. a** Peak power increase, frequency at peak, and firing rate of MUA across the set of responsive electrodes as a function of optogenetic stimulation frequency. A schematic representation of the stimulation and recording site is shown on the left. Error bands represent SEM. **b** Time-frequency representations of LFP data, recorded with the deepest electrode, during optical stimulation with 5 Hz (50% duty cycle rectangular blue light pulses—top; ten trials). The range of MMCE window sizes was matched with the range spanned by the SLT wavelets. **c** Scaled correlation analysis (SCA) on the data from **b**. Top: time-resolved. Bottom: time-collapsed around the temporal location of the burst in **b**. **d** Analysis of multi-unit activity (MUA) recorded on the same electrode as the LFP in **b**. Top: Peri-Stimulus Time Histogram (PSTH); bottom: binary SCA on the MUA spikes. **e** Time-frequency representations of LFP data recorded during stimulation with 65 Hz, on a more superficial electrode (single trial). Green numbers denote periods with peak power at 65 Hz (1 and 3) and with reduced power between these peaks (2). Red arrows indicate two high frequency (130 Hz) transients. **f** The single trial trace from **e** (top), followed by its band-passed versions (Butterworth IIR, order 3, bidirectional) at 60–70 Hz (middle) and 110–140 Hz (bottom). **g** SCA on the LFP signal from **e** and **f**, with time-collapsed versions at the locations of the two bursts indicated by red arrows in **e**. **h** Event-related potentials (ERP) aligned to light onset computed on the LFP signal during periods 1, 2, and 3, indicated in **e**. **i** The LFP signal between 30–80 ms, at the location of the high-frequency burst (first red arrow in **e**), in relation to the light stimulation profile.

methods, including those based on the directionally filtered WVD, potentially impairing important discoveries about the simultaneous coordination of rhythms across neighboring bands. SLs provide a robust and elegant solution to this problem because the TF concentration of spectral power they provide minimizes the cross-band contamination during spectral estimation.

On the neuroscience data used to test the methods investigated here, SLs provided the best results. Neural data have a rich TF structure[2] and are often contaminated by noise (e.g., EEG). The reason why SLs outperform other methods on such data may have to do with the scale-free, fractal nature of brain signals[5–7]. Indeed, for such signals, oscillation bursts are scaled

(compressed or dilated), covering a wide range of frequencies, and wavelet analysis is the natural approach. When normalized such that temporally-scaled oscillation busts, with the same amplitude, yield the same peak response[21], wavelets become useful tools that reveal the scale-free nature of these bursts. SLs inherit these advantages and use a wavelet normalization that facilitates the detection of oscillation bursts as they compress with increasing frequency. When a fixed order is used (SLT), the resulting representation is "non-diluting", i.e., bursts with the same peak amplitude will receive the same peak "intensity" in the SL representation. By contrast, methods based on the Fourier transform, such as the spectrogram or the MMCE, use

one or multiple fixed analysis windows, respectively. As window size is not dependent on the continuous change of the timescale across the frequency domain, these methods progressively "dilute" finite oscillation packets that compress with the increase in frequency. Finally, techniques based on the directionally smoothed WVD, seem to significantly underperform on brain signals, most likely because the rich TF structure of these signals induce cross-terms that are difficult to suppress, even with advanced kernels.

SLs generalize the CWT to provide super-resolution in the TF sense. One of the major advantages of SLs is their simplicity and straightforward implementation, facilitating an intuitive understanding of what they do. Indeed, the base wavelet controls the time resolution, whereas the order determines how well the representation is sharpened in frequency. Unlike in the STFT or MMCE, choosing the parameters for SLs ($c_1$, $o$) is relatively easy, because one operates with oscillation cycles rather than absolute time windows. The SLs' base wavelet cycles can be chosen independently of the frequency range of the representation.

On neuroscience data SLs provide interesting results and it is expected that the method will be instrumental in identifying high-frequency oscillation bursts whose expression in brain signals is difficult to quantify. Indeed, the examples shown here, especially those in Fig. 10 and Supplementary Figs. 13–15 compellingly speak for the ability of this method to reveal transient oscillation frequencies that go well beyond the traditional frequency band usually explored (e.g., 0–100 Hz). Furthermore, SLs may find applications in the analysis of other types of signals whose TF landscape is complex.

## Methods

High-density EEG (Biosemi ActiveTwo 128 electrodes; recording software Acti-View for Windows version 6.05) data were recorded @1024 samples/s from healthy human volunteers freely exploring visual stimuli consisting of deformed lattices of dots that represented objects and were presented on a 22″ monitor (1680 × 1050@120 fps; distance 1.12 m).

We generated a set of 210 stimuli, consisting of dot lattices corresponding to 30 objects and deformed progressively to yield 7 levels of visibility[40]. For each of the 7 visibility levels (parameter range $g = 0$–$0.3$, in steps of 0.05), we generated 30 dots images corresponding to the 30 objects, grouping stimuli into 7 experimental blocks. Blocks were presented to subjects in ascending order of visibility (from nothing visible to easily visible). For each dot image presented on the screen, subjects were free to visually explore the stimulus for as long as they needed (free visual exploration paradigm), until they reached a perceptual decision, which they had to signal by pressing one of three buttons congruent with perception ("nothing," "uncertain," "seen"). Data were recorded from 11 subjects. Here we used data from two subjects. All subjects gave their written informed consent before the experiment. The protocol was approved by the Local Ethics Committee (approval 1/CE/08.01.2018). Data were collected in accordance with relevant legislation: Directive (EU) 2016/680 and Romanian Law 190/2018.

In vivo electrophysiology data were recorded with A32-tet probes (Neuro-Nexus Technologies, Inc.) at 32 kSamples/s (Multi Channel Systems MCS GmbH; recording software MC_Rack version 4.6.2) from primary visual cortex of anesthetized C57/Bl6 mice receiving monocular visual stimulation (1440 × 900@60fps; distance 10 cm) with full-field drifting gratings (0.11 cycles/deg; 1.75 cycles/s; contrast 25–100%; 8 directions in steps of 45°, each shown 10 times). Anesthesia was induced and maintained with a mixture of $O_2$ and iso-flurane (1.2%), and was constantly monitored based on heart and respiration rates and testing the pedal reflex. Within a stereotaxic device (Stoelting) a craniotomy (1 × 1 mm) was performed over visual cortex. To minimize animal use, multiple datasets were recorded over 6–8 h from each animal. Local field potentials were obtained by low-pass filtering the signals @300 Hz and down-sampling to 4 kHz.

For optogenetics experiments, we used genetically modified mice expressing light-activatable ion channels (ChR-2) in neuronal subsets of pyramidal neurons within the cortex, via the Thy-1 promoter[54]. Optical stimulation was delivered via a blue laser (Sanctity 473 nm DPSS Laser) to the primary visual cortex using stereotactic coordinates, in close vicinity to the recording site. Stimulation was delivered in repetitive pulses at variable frequencies (from 5 to 105 Hz) with a duty cycle of 50%, via a custom-built mechanical shutter. Laser power was adjusted in order to activate neuronal population, but not exceed 50 mW/mm². The artifact of

light pulses on the electrical contacts was minimized by placing the optical fiber at an angle, ensuring that the light did not reach the probe contacts, but did elicit a neural response in both LFP and MUA. We observed no effect of light pulses post-mortem or in animals that did not express ChR-2.

Mice were housed in a controlled environment, with temperature in the range of 21–23 °C and 50–60% humidity, with a dark/light cycle of 12/12 h. In vivo experiments were approved by the Local Ethics Committee (3/CE/02.11.2018) and the National Veterinary Authority (ANSVSA; 147/04.12.2018).

**Reporting summary.** Further information on research design is available in the Nature Research Reporting Summary linked to this article.

## Data availability

EEG and electrophysiology datasets analyzed for the present study are available from the corresponding author on reasonable request. EEG dataset codes are: Dots_30_001 (Figs. 4a, b and 8a–c) and Dots_30_002 (Figs. 6a, b and 7). In vivo electrophysiology codes are: M017_002 (Figs. 4c–e, 5, 6c–e, and 8d–f) and M022_004 (Fig. 10).

## Code availability

Data were analyzed using custom software. A freely available version, written in C++ version 11 with an interface for Matlab 2018b and including code for producing artificially generated data can be found at: https://github.com/TransylvanianInstituteOfNeuroscience/Superlets.

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

## Acknowledgements

The research leading to these results has received funding from: NO Grants 2014-2021, under Project contract number 20/2020 (RO-NO-2019-0504), two grants from the Romanian National Authority for Scientific Research and Innovation, CNCS-UEFISCDI (codes COFUND-NEURON-NMDAR-PSY and PN-III-P2-2.1-PED-2019-0277), and a National Science Foundation grant NSF-IOS-1656830 funded by the US Government. Any opinions, findings, and conclusions or recommendations expressed in this material are those of the authors and do not necessarily reflect the views of the National Science Foundation. We thank Wolf Singer and Gal Vishne for insightful comments on the method and manuscript.

## Author contributions

R.C.M. and V.V.M. developed the method. V.V.M., R.C.M., and H.B. generated the toy data. A.N.-D, H.B., and R.C.M. recorded the test data. H.B. and R.C.M. implemented and tested the method and performed the comparison to other methods. V.V.M. and R.C.M. performed the analytical derivations. R.C.M., V.V.M., H.B., and A.N.-D. wrote the manuscript.

## Competing interests

The authors declare no competing interests.
