## [Peer Review File · Nature Communications]

Reviewers' comments:

Reviewer #1 (Remarks to the Author):

The authors present a method, called superlets, for obtaining “super-resolution” timefrequency representations of biological signals, by combining multiple wavelet transforms. They claim that, “superlets outperform ... other super-resolution methods ... with unprecedented precision” and “superlets can reveal transient oscillation events that are hidden ... by other methods.” These are extraordinary claims that need to be substantiated by quantitative and fair comparisons to the best methods available, as discussed below.

1. Quantitative comparisons to other high-resolution methods

a) As the authors note, combining multiple time-frequency representations is not new. Different combinations have been proposed, using different optimality criteria as well as different time-frequency representations (e.g., in addition to ref. [17] in their manuscript, see [1] below). The authors consider just one of these methods. Hence, together with the next point, it is premature to claim their method yields “unprecedented precision” and uncovers time-frequency structure that no other method can.

b) For most of the examples (Figs. 1, 3, 4, 5, 6, S2), the authors compare their superlets to single spectrograms or wavelet transforms. Given that the time-frequency resolution of these methods is about as poor as it can be, it is not hard to do better than that! The combination of multiple spectrograms should be included in all cases, as should comparisons to other high-resolution methods, such as the Choi-Williams and RID time-frequency representations

c) Additional quantitative comparisons are needed to substantiate their claims. Since there is no unique definition of the time-varying spectrum of a signal, we do not know “ground truth” (even for known signals that we generate, like the authors’ test signal). Hence, comparing one representation against another to draw conclusions as to which one best represents the (unknown) time-frequency structure of a signal is a challenge. However, one important comparison that should be made, for which we do know the ground truth, is to compare the marginal densities of each representation against the true marginals (i.e., the magnitude-square of the signal and its Fourier transform). It would be helpful and informative for the authors to include plots of the marginals of each representation vs. the true marginals, as well as quantitative metrics of the error in each one.

d) Widely used test signals for evaluating time-frequency methods should be included in addition to the authors’ test signal, with quantitative comparisons. Namely, a complexvalued single tone burst, and a complex-valued single linear FM chirp: $s(t) = \exp(j(\omega_0 t + \beta t^2 / 2))$, $t_1 < t < t_2$ and $= 0$ otherwise. While even for these signals we do not know the true time-frequency structure, we do know the spectral density and the temporal density (i.e., the true marginals), the instantaneous frequency, and that these signals are zero outside a finite time interval, all of which should be accurately reflected in a ‘good’ time-frequency representation of the signal.

e) For fair comparisons, it is important that one combines representations that span the same time and frequency resolutions; i.e., the maximum and (especially) minimum resolutions in time and frequency of the set of spectrograms should match that of the set of wavelets. In addition, the various representations in each set should all be normalized to match the total signal power. It is not clear that

this was the case in the comparisons made (e.g., in Fig. 7, the power of the MMCE result is at the low-end of the heat map, while the superlets result is at the high-end (especially the first column in A v. the first column in B)). Similarly, the authors' comments about the spectrogram and 6 MMCE "diluting" power as frequency increases, resulting in poorer detection compared to the superlet, is likely an issue of improper normalization: if the spectrogram window is properly normalized (to unit-energy), two different-frequency but equal-amplitude tone pulses will have the same peak power in the spectrogram.

f) Regarding the detection results, given the statistical nature of detection and the dependence on SNR and threshold levels, one example is insufficient to claim one method is superior to another. Such claims need to be more extensively evaluated; for example, by computing ROC curves.

2. Define and calculate time-frequency resolution, and clarify restrictions of the uncertainty principle

a) It is important to define and quantify what is meant by the "time-frequency resolution" of the representation. The uncertainty principle constrains the product of a signal's duration and bandwidth (commonly taken as the standard deviations in time and frequency); as discussed in [2], those are global quantities. One can calculate analogous local quantities, namely the standard deviation in time at a given frequency, and vice versa. In general, there is no uncertainty constraint on these local resolution quantities, even though the global quantities are constrained. However, for the spectrogram and other windowed-methods, including wavelets, these local quantities are constrained by the uncertainty principle. Hence, it would be valuable for the authors to calculate the global and local uncertainty product for the various examples and representations.

b) The uncertainty product of the spectrogram and wavelets has been extensively studied, along with their marginals and moments [3]. In general, the results obtained with wavelets are vastly inferior in terms of these metrics compared to those for the spectrogram. As such, it is curious that the authors seem to be obtaining better results by combining wavelets v. spectrograms. Can the authors take into account the results of [3] and show that the combination of wavelets indeed yields better results than does combining spectrograms, in terms of these metrics? If this is indeed the case, it would be very valuable to explain why that is, given that the individual wavelets are worse than the individual spectrograms. (As noted previously, their comments about the spectrogram / MMCE "diluting" power may be an issue of improper normalization and hence would not explain the claimed advantages that emerge when combining wavelets.)

Reviewer #2 (Remarks to the Author):

This is quite an interesting paper on signal processing for neural data analysis. (It potentially has wider applicability in areas such as speech, but these are outside the scope of my review, and is not explicitly considered in the paper.)

One of the limiting factors in neural time series analysis is that many important neural events are best described by an optimal frequency range and time window. Conventionally these signals have either been analyzed using FFT based methods or wavelet methods, which make different fundamental trade-offs of time and frequency resolution and each can be performed with a different choice of time or frequency resolution (in FFT, by the window size, in wavelets, by the number of cycles). The paper does a nice job of explaining this problem.

The authors revive, extend, and nicely quantify the resolution limits of a technique based on taking

the geometric mean of time-frequency spectra with different time-frequency resolution to improve the information available (labelled "super-resolution"). That's probably overselling it, but the simulations and data analysis shown do support the claim of better time-frequency localization of signals. The original paper (Reference 16) did this based on two spectrograms obtained by FFT, the authors nicely optimize this method to multiple spectrograms based on wavelets.

The paper is potentially impactful in neuroscience and perhaps other fields, but my enthusiasm is severely limited because the authors don't really make an attempt to explain mathematically why this should and does work. There is an oblique reference to multi-resolution imaging, but I would like to see a mathematical analysis.

Neuroscience suffers from new techniques being proposed without much theoretical underpinning. These techniques get popular but then ultimately die because no one really knows what they are doing. I think the technique proposed here is valuable, but is not yet properly justified.

Minor comments

In equation 2, where does the 5 come from?

Particularly in the simulations, too much of the analysis provided visually where I would prefer to get goodness of fit measures, since the answer is exactly known.

Reviewer #3 (Remarks to the Author):

The authors develop an algorithm to, in simple terms, get the frequency resolution of a short-time Fourier transform (STFT) and the time resolution of a wavelet transform (CWT). In their approach they average wavelets with increasing width geometrically, and when there is a need to cover a broader frequency range they make the wavelets adaptive. The algorithm is applied to examples, to determine the interference when multiple transient signals are present (close in time or close in frequency) and apply it to example experimental data and compare it to STFT, CWT and another approach to combine measurements, albeit in frequency space (MMCE). Overall, they show that their method, SLT or ASLT, outperforms the STFT and CWT in time and frequency resolution, and is a little bit crisper than MMCE. What struck me as very useful is the ability to do single-trial analysis, or be sensitive to transient events that occur on a few trials amongst the ones included in the average. This is a useful contribution to the analysis of experimental data for cases where there is variability in time and background state in the activity patterns (which is the case in most cognitive neuroscience experiments).

Overall, the paper provides a useful contribution to the analysis of the experimental neuroscience data.

Specific comments:

There are other ways of combining the wavelets (for instance, difference weights in additive and different powers in geometric), and one could formulate an objective function measuring temporal and frequency resolution of event detection to be optimized across an ensemble of transient events. Why is the choice used in this paper optimal? And for what objective function is it optimal, what would I need to do when I value temporal resolution twice as important as frequency resolution?

The transient events in the model data have a length of 7 or 11 cycles — which in typical experimental data would probably be 2-5 cycles. Perhaps some of the figures can be tested on a couple of values for the length.

Page 10 — bottom: Fig2B/Fig2C should refer to figure 3 instead

NCOMMS-19-31444-T: Superlets: time-frequency super-resolution using wavelet sets

Vasile V. Moca, Harald Bârzan, Adriana Nagy-Dăbâcan, Raul C. Mureşan

Point-by-point reply to reviewers

We thank the editor and all reviewers for their insightful and very useful comments. We have struggled to address all the concerns and hope that the manuscript is now in a satisfactory shape.

Summary of the revision:

1. **Extensive mathematical treatment.** Our strategy for the revision was to keep the Main Text as light and easy to understand as possible, to make it accessible for a broad readership. Therefore, we deferred the mathematical details to an extensive Supplementary Information (SI), which now provides important insights, details, and discussions for those interested in the rigorous treatment of *superlets*. We struggled to write the mathematical derivation in SI as easy to follow as possible for readers with a minimal mathematical background. We feel that in many fields of science that use advanced signal processing methods, like neuroscience for example, there is still a superficial level of understanding and much confusion. In fact, even in the signal processing community some of the concepts related to time-frequency analysis are less clear than one may think. One of the significant issues related to *superlets* is the nature of the representation they provide and the way in which wavelets are normalized. In fact, we show that *superlets* are related to methods of wavelet ridge detection. We also go in great detail to set a general framework for understanding the relation between time-frequency (TFR) and time-scale representations (TSR), and we relate different techniques to general classes of representations. We hope the SI will help the average reader to get a coherent understanding of the “whats” and “whys” of time-frequency / time-scale analysis.
2. **Mathematical derivation of the superlets’ working principle.** Importantly, we have invested a significant effort in finding ways to analytically derive how *superlets* work. It turns out the method can be elegantly understood within the framework of band-pass filters. We show why *superlets* act as “frequency sharpeners”, “rejecting” redundant power and narrowing band-pass characteristics in the higher frequencies.
3. **Derivation of constant bandwidth superlets.** Using the analytical framework developed for *superlets*, we show that one can calculate the exact parameters of the ASLT to achieve constant frequency bandwidth. In fact, the constant frequency bandwidth ASLT is very similar to the MMCE method and provides very similar results. One may therefore consider *superlets* as a generalization that also encompasses the MMCE method as a special case.
4. **Dilution phenomenon in TFR/TSRs.** The SI contains a dedicated section where the difference between MMCE and *superlets* is discussed. We also provide the explanation for the “dilution” phenomenon, and show further examples based on Gaussian atoms.

5. **Marginals and Uncertainty Product (UCP).** An entire section of the SI is dedicated to the UCP, where we define it rigorously following Cohen's formalism. We also compute the TFR/TSR of various signals, from simple (atom, tone), to more complex ones (chirp, multi-packet landscape). We compare the representations and marginals computed with Wigner-Ville Distributions (WVD) and its directionally smoothed variants (Choi-Williams – CW; Born-Jordan – BJ), as well as *superlets* and MMCE. We show and discuss why marginals are not very useful to describing the quality or computing the resolution of a certain representation.
6. **Relation to generic TFR/TSR classes.** We discuss the relation of MMCE and *superlets* to generic Cohen class / bilinear methods and to the Rioul and Flandrin's "affine" smoothed TSD class [1]. While we do not insist too much in this direction, we hope that the basic concepts we provide may prompt novel research trying to understand better how the MMCE and *superlets* may be put in the context of more general formalisms.
7. **High-resolution methods on real data.** We now computed the WVD, CW, BJ representations for EEG and electrophysiology data. We provide the results in the Main Text. We should mention that these methods are not at all popular in neuroscience, and, after applying these on real data we also understood why. Biological data has a very rich and complex time-frequency landscape, such that methods that over-concentrate the representation become overwhelmed by cross-terms. Even methods like CW and BJ suffer from the directional smoothing to such extent that very obvious and well-known features of neuronal signals cannot be observed.
8. **Detection accuracy and resolution measurements.** We devised detection accuracy and resolution measurements and applied them on Gaussian atoms and sine packets. As found by previous empirical testing, the results confirm that *superlets* are superior to other methods. For a single frequency, *superlets* and MMCE with Gaussian window can be matched to be equivalent. In fact, we show this relation in SI and also provide the conversion formula between the two representations, at a given, fixed frequency.
9. **Improved clarity.** We struggled to also improve the clarity of the main text, by rephrasing here and there, and using a less ambiguous terminology. For example, to avoid confusion between the Morlet bandwidth parameter and the bandwidth of the wavelet (they are inversely proportional), we now use "time spread parameter" for the former and "bandwidth" for the latter. Although in the literature the bandwidth term is sometimes used inconsistently, referring to the bandwidth parameter, we feel this creates more confusion. We therefore resorted to a non-ambiguous terminology that is better grounded in wavelet theory.

We will provide our answers below to each point raised by reviewers. The reviewers' comments will be rendered in blue and our replies in black.

Replies to reviewer #1:

The authors present a method, called *superlets*, for obtaining “super-resolution” time-frequency representations of biological signals, by combining multiple wavelet transforms. They claim that, “superlets outperform ... other super-resolution methods ... with unprecedented precision” and “superlets can reveal transient oscillation events that are hidden ... by other methods.” These are extraordinary claims that need to be substantiated by quantitative and fair comparisons to the best methods available, as discussed below.

We agree with the reviewer and thank him for pointing us to the most relevant methods in time-frequency analysis. Many of these advanced methods, such as Choi-Williams (CW) and Born-Jordan (BJ), are rather unknown to neuroscientists and have witnessed, perhaps undeservedly, little attention in the community performing analysis of brain signals. However, there may be a good reason why these methods did not get traction in the analysis of biological signals – they perform rather poorly on these types of signals, as we learned during the revision.

To get a deep understanding, we implemented these methods ourselves, including the Wigner-Ville Distribution (WVD) and its directionally smoothed counterparts (CW and BJ) and we cross-checked with other mainstream tools, like Matlab. We also spent a significant effort reviewing the literature and relating to the outstanding work of Leon Cohen, and the very nice extensions by Patrick Flandrin and colleagues, or the excellent books by Boualem Boashash and Stéphane Mallat. Following the leads indicated by the reviewer, we were able to better relate our work to the significant advances in time-frequency analysis.

We found the comments of the reviewer extremely useful and struggled to address all of them in detail. In this process, we also understood our method better – including a thorough analytical treatment – and found ways to relate it more systematically to previous work. For a fairer relation to previous work, we also tempered the tone of the manuscript, as suggested by the reviewer. Please see our detailed answers below:

1. Quantitative comparisons to other high-resolution methods

- a) As the authors note, combining multiple time-frequency representations is not new. Different combinations have been proposed, using different optimality criteria as well as different time-frequency representations (e.g., in addition to ref. [17] in their manuscript, see [1] below). The authors consider just one of these methods. Hence, together with the next point, it is premature to claim their method yields “unprecedented precision” and uncovers time-frequency structure that no other method can.

We have removed the “hard” formulations from the Main Text and have added a greatly expanded Supplementary Information (SI), which explores in detail the place of *superlets* in the world of time-frequency / time-scale representations. We refer specifically to biological signals and show that, at least for neuroscience data, the new method is superior to others. The Main Text has 3 new Fig.s, while the SI has 8 new Fig.s with various results. We have tried really hard to compare our method as fairly as possible to many other techniques and have also derived analytically how it behaves. At least for neuroscience data, our conclusions hold – the *superlets* are unmatched by any of the methods we tested.

- b) For most of the examples (Figs. 1, 3, 4, 5, 6, S2), the authors compare their superlets to single spectrograms or wavelet transforms. Given that the time-frequency resolution of these methods is about as poor as it can be, it is not hard to do better than that! The combination of multiple spectrograms should be included in all cases, as should comparisons to other high-resolution methods, such as the Choi-Williams and RID time-frequency representations.

A first reason why most comparisons were made to spectrograms and scalograms is that these are overwhelmingly used in the analysis of brain signals today. As mentioned before, more advanced techniques, such as Choi-Williams and Born-Jordan have not witnessed widespread adoption in neuroscience. A second reason is that, for the general readership, explaining the limitations of the STFT and CWT and comparing them to higher resolution methods gives a strong intuitive understanding of what is going on. However, we do agree with the reviewer that a comparison to more advanced techniques is necessary. This is now extensively done (see Figs. 10, 11, 12, S10, and S11).

Anecdotically, we developed the *superlet* trying to fix the poor frequency resolution of the CWT for higher frequencies. More generally, our work is focused significantly on high-frequency gamma oscillations and we were not satisfied by the frequency resolution provided by the CWT. In fact, the idea to combine wavelets geometrically came out of intuition. Our first version of the manuscript is here: <https://www.biorxiv.org/content/10.1101/583732v1>. We only later learned about the MMCE, after independently inferring the usefulness of the geometric mean in spectral estimation.

Finally, comparing to the traditional STFT and CWT is also essential for the broad readership that may not be very technically knowledgeable about very advanced time-frequency techniques. In our opinion, using STFT and CWT as baselines, helps the broad readership understand how *superlets* (and other methods) work, and what are the limitations and challenges of time-frequency super-resolution in general. We therefore kept the main backbone of the paper but expanded it with new comparisons and also provided an extensive SI to frame our method rigorously.

- c) Additional quantitative comparisons are needed to substantiate their claims. Since there is no unique definition of the time-varying spectrum of a signal, we do not know “ground truth” (even for known signals that we generate, like the authors’ test signal). Hence, comparing one representation against another to draw conclusions as to which one best represents the (unknown) time-frequency structure of a signal is a challenge. However, one important comparison that should be made, for which we do know the ground truth, is to compare the marginal densities of each representation against the true marginals (*i.e.*, the magnitude-square of the signal and its Fourier transform). It would be helpful and informative for the authors to include plots of the marginals of each representation vs. the true marginals, as well as quantitative metrics of the error in each one.

The reviewer touches an important point here. We agree that, in general, it is hard to define a “ground truth” for the time-frequency spectrum of a signal. In the end, it all depends on how a certain representation is used to infer something about the properties of the signal. In neuroscience (and likely geoscience too) this is not so fuzzy: one would like to know where

oscillation bursts start, where they end, what is their frequency and magnitude/power. The reason is that these patterns of activity are generated by stereotypical neural circuits that produce well-known patterns in various frequency bands. What is particularly troublesome in neuroscience is the difficulty to distinguish between brief burst in both time and frequency, and to compensate for the sometimes overwhelming $1/f$ characteristic of the signal, which hinders estimation of higher frequencies. All this has to be done on signals that have a very rich time-frequency landscape and are plagued by noise and irrelevant activity. In that respect, two finite sine packets, with a sharp start/end and clear frequency have a rather clear “ideal representation” in the mind of the user, and the user expects to find such signature in the time-frequency representations of the signal.

Regarding marginals, we thank the reviewer for pointing us in this direction. Indeed, there is a large amount of work on this topic, especially by Cohen. We now added an entire section dedicated to marginals in the SI, and compute it on various representations, including the high-resolution ones like WVD, CW, and BJ. Unfortunately, as pointed out by Flandrin et al. (ICASSP '94)[2] and many other authors since, marginals aren't very useful for the characterization of a time-frequency representation. We provide a clear example where methods with beautiful marginals show a much poorer representation, including indistinguishable packets in time, than other representations whose marginals are not so sharp (see Figs. S10 and S11, “Neighbors” in SI).

- d) Widely used test signals for evaluating time-frequency methods should be included in addition to the authors' test signal, with quantitative comparisons. Namely, a complex-valued single tone burst, and a complex-valued single linear FM chirp: $s(t) = \exp(j(\omega_0 t + \beta t^2/2))$, $t_1 < t < t_2$ and $= 0$ otherwise. While even for these signals we do not know the true time-frequency structure, we do know the spectral density and the temporal density (*i.e.*, the true marginals), the instantaneous frequency, and that these signals are zero outside a finite time interval, all of which should be accurately reflected in a 'good' time-frequency representation of the signal.

These are all good points. We took the advice of the reviewer and computed the representations for all these signals with WVD, CW, BJ, MMCE, and ASLT. Results are shown in subsection “Marginals of high-resolution representations” and Figs. S10 and S11 in the SI. We would like to argue that methods which apparently behave optimally on simpler signals, may in fact be catastrophically impaired on real-world data – at least on some types of signals, like those originating from biological systems. Our results indicate just that: State-of-the-art time-frequency methods based on directionally smoothed WVD are mostly unusable on EEG and electrophysiology data. We hope that our analysis in Fig. S10 and S11 as well as the discussion throughout the SI shed more light on these issues.

- e) For fair comparisons, it is important that one combines representations that span the same time and frequency resolutions; *i.e.*, the maximum and (especially) minimum resolutions in time and frequency of the set of spectrograms should match that of the set of wavelets. In addition, the various representations in each set should all be normalized to match the total signal power. It is not clear that this was the case in the comparisons made (*e.g.*, in Fig. 7, the power of the MMCE result is at the low-end of the heat map, while the superlets result is at the high-end (especially the first column in A *v.* the first column in B)). Similarly, the authors' comments about the spectrogram and

MMCE “diluting” power as frequency increases, resulting in poorer detection compared to the superlet, is likely an issue of improper normalization: if the spectrogram window is properly normalized (to unit-energy), two different-frequency but equal-amplitude tone pulses will have the same peak power in the spectrogram.

In general, the MMCE and *superlets* cannot be matched, except at a single frequency – just like there is a single-frequency match between the spectrogram with Gaussian window and the Morlet-based scalogram. For the latter, we provide the exact matching formula (eq. s45). The reason why MMCE and SLT are not equivalent is that while one uses a single set of windows, the second uses a set of “windows” at each frequency. In addition, there are constraints to how small a window for MMCE one can choose, which are dictated by the lowest frequency (largest period) in the spectrum of interest. A too small window leads to aliasing. As a result, it is in general not possible to choose for the MMCE the “smallest” window that the SLT uses (at the highest frequency). These aspects are now discussed extensively in the SI “Interpretation of superlets and relation to MMCE”. We should mention that we always struggled to choose the MMCE parameters to get the closest match to the SLT. In addition, we now introduce the “constant bandwidth” ASLT, where orders are adaptively adjusted such that the frequency bandwidth of *superlets* is constant. This type of ASLT is almost identical to the MMCE, and therefore the ASLT may be viewed as a generalization whose special case is the MMCE.

Regarding normalization, this was properly done for all the transforms. For the spectrogram, we used unit energy windows and the integral of the representation was always the energy of the signal. One should note that even if the spectrograms are normalized, the integral of the MMCE will not yield the energy of the signal (because the sum of geometric means is not the geometric mean of the sums). Regarding scalograms, the normalization is in fact such an important issue, that we decided to dedicate it an entire section in the SI: see “Fundamentals of spectrograms and scalograms”. The scalogram does not integrate to the energy of the signal, but extra terms ($1/a^2$ or $1/a$, depending on the type of scalogram) have to be added to obtain the energy. The correct definition for the scalogram is given in Rioul and Flandrin [1], [3], in the original papers of Grossmann and Morlet [4], and countless papers and textbooks since, e.g. [5]. The scalogram does not preserve marginals [1] and its integral does not yield the energy, except when extra scaling is added in the integral. The correct implementation relies on the original definition, as is also done in Matlab and many mainstream tools.

Regarding Fig. 7, its label states how the color scales should be interpreted. These were not matched across methods but only across the 3 conditions for each method – i.e. the color intensity has the same interpretation only across A – left, across A – right, and the same for B – left and B – right. The purpose of the representation is to identify differences in the distribution of oscillation packets as induced by the experimental condition, not to determine how much total energy there is in the piece of signal. Indeed, the global scaling of the power is irrelevant for such applications, except when one attempts reconstruction from the representation. We should mention that the representations throughout the paper would not change at all if the representations were scaled, since the color scale always goes from min to max, and scaling has no effect. The only place where representations were matched across conditions (but not methods!) is Fig. 7. The “dilution” effect we mention has nothing to do with this global scaling – it refers to the relation between low and high frequencies.

Regarding the dilution phenomenon in the spectrogram, this is now discussed in detail in the SI (see Fig. S7 and the corresponding section in SI). The reviewer is right: two tone impulses with same amplitude but different frequency would have the same representation in the spectrogram. But this only holds if: either the two packets have equal duration, or each packet is longer than the size of the STFT window. In all examples we give, we introduce packets of equal cycle number (not equal absolute duration!) because this mimics the real case for biological data (and also for other types of data). In Fig. 7 the dilution appears precisely because oscillation bursts in higher frequencies have a shorter duration than those at lower frequencies and end up occupying just a fraction of the STFT window. This is precisely what has hindered the proper study of gamma oscillations for decades. The *superlets* come to solve exactly this problem and the paper is constructed fundamentally on these premises (see also Fig. 1, and all discussions in SI and Main Text).

- f) Regarding the detection results, given the statistical nature of detection and the dependence on SNR and threshold levels, one example is insufficient to claim one method is superior to another. Such claims need to be more extensively evaluated; for example, by computing ROC curves.

We fully agree! We now devised a novel test using Gaussian atoms and sine packets embedded in various amounts of noise. We measure how all the methods: WVD, CW, BJ, STFT, CWT, MMCE, ASLT fare on detecting the target signals as a function of the amount of added noise (see Fig. 11 and the corresponding description in Main Text).

2. Define and calculate time-frequency resolution, and clarify restrictions of the uncertainty principle

- a) It is important to define and quantify what is meant by the “time-frequency resolution” of the representation. The uncertainty principle constrains the product of a signal’s duration and bandwidth (commonly taken as the standard deviations in time and frequency); as discussed in [2], those are global quantities. One can calculate analogous local quantities, namely the standard deviation in time at a given frequency, and vice versa. In general, there is no uncertainty constraint on these local resolution quantities, even though the global quantities are constrained. However, for the spectrogram and other windowed-methods, including wavelets, these local quantities *are constrained* by the uncertainty principle. Hence, it would be valuable for the authors to calculate the global and local uncertainty product for the various examples and representations.

We took the reviewer’s advice seriously and examined in detail the marginals of various representations. We also computed the uncertainty product (UCP) on various signals for the *superlet* and its composing wavelets (see section “Resolution of TFR/TSR representations” in SI). The result is interesting in that the UCP of the SLT is smaller than the UCP of each individual wavelet in its set, especially on sine packets (see Fig. S9). We also discuss the issues of UCP and how to compute it correctly, as discussed in the literature (see Fig. S8 and corresponding text). Finally, the local UCP is not very useful either to evaluate the quality of the representation, because the UCP is based on first and second moments of the distribution which are statistically ill-defined for multi-modal distributions (see Flandrin, Baraniuk et al. [2], or

Stanković [6]). Instead, we introduce the concept of resolution in the sense of discrimination between neighboring components, as is done in optics and as was originally defined by Rayleigh. The same path is taken by others for TFRs, like Boashash and Sucic [7], or Stanković [6]s. Fig. 12 in Main Text shows these new results.

b) The uncertainty product of the spectrogram and wavelets has been extensively studied, along with their marginals and moments [3]. In general, the results obtained with wavelets are vastly inferior in terms of these metrics compared to those for the spectrogram. As such, it is curious that the authors seem to be obtaining better results by combining wavelets v. spectrograms. Can the authors take into account the results of [3] and show that the combination of wavelets indeed yields better results than does combining spectrograms, in terms of these metrics? If this is indeed the case, it would be very valuable to explain why that is, given that the individual wavelets are worse than the individual spectrograms. (As noted previously, their comments about the spectrogram / MMCE “diluting” power may be an issue of improper normalization and hence would not explain the claimed advantages that emerge when combining wavelets.)

Regarding the UCP on spectrograms and scalograms (wavelets), it should be noted that representations based on scalograms depend critically on the type of wavelet normalization (see “Fundamentals of spectrograms and scalograms” in the SI). A special kind of normalization (used in many studies and implemented in Matlab as the cwt function) conserves the integral rather than the energy of the mother wavelet. This kind of scalogram was used extensively for wavelet ridge detection [5], [8] and is also the one we employ for *superlets*.

The MMCE with Gaussian window and the Morlet SLT are precisely equivalent at a single frequency (when windows are matched). The reason why the SLT outperforms the MMCE is because the former is multi-resolution in the wavelet sense (adapts to the scale of each frequency) while the MMCE is not (a fixed set of windows covers all frequencies). This is now clearly explained in SI, “Interpretation of superlets and relation to MMCE”. Also, the analytical derivations in the SI now help to better understanding how the SLT works and why its UCP is dramatically improved compared to single wavelets (actually, it has the UCP of the optimally-sized wavelet for a certain signal component). The results we obtained are certainly not flawed by any normalization issue. They are now backed up by robust mathematical formulations and extensive testing on various cases that we have performed under the guidance of the reviewer.

Finally, we should mention that we are really grateful to the reviewer for these interesting and eye-opening comments. Our efforts to address them improved the manuscript significantly, in our opinion. And helped us understand much better what *superlets* do and how they compare to other techniques.

Replies to reviewer #2:

This is quite an interesting paper on signal processing for neural data analysis. (It potentially has wider applicability in areas such as speech, but these are outside the scope of my review, and is not explicitly considered in the paper.)

One of the limiting factors in neural time series analysis is that many important neural events are best described by an optimal frequency range and time window. Conventionally these signals have either been analyzed using FFT based methods or wavelet methods, which make different fundamental trade-offs of time and frequency resolution and each can be performed with a different choice of time or frequency resolution (in FFT, by the window size, in wavelets, by the number of cycles) . The paper does a nice job of explaining this problem.

The authors revive, extend, and nicely quantify the resolution limits of a technique based on taking the geometric mean of time-frequency spectra with different time-frequency resolution to improve the information available (labelled "super-resolution"). That's probably overselling it, but the simulations and data analysis shown do support the claim of better time-frequency localization of signals. The original paper (Reference 16) did this based on two spectrograms obtained by FFT, the authors nicely optimize this method to multiple spectrograms based on wavelets.

Following the suggestions of reviewer #1 we have toned down the wording of the paper to let the results speak for themselves. Regarding the term super-resolution, this is meant to refer to the simultaneous time and frequency resolution of the method. The same terminology is used in a set of seminal reviews, like that of Shafi et al. [9]. We believe that the new results in the revision also justify the usage of the "super-resolution" label for *superlets*.

The paper is potentially impactful in neuroscience and perhaps other fields, but my enthusiasm is severely limited because the authors don't really make an attempt to explain mathematically why this should and does work. There is an oblique reference to multi-resolution imaging, but I would like to see a mathematical analysis.

Neuroscience suffers from new techniques being proposed without much theoretical underpinning. These techniques get popular but then ultimately die because no one really knows what they are doing. I think the technique proposed here is valuable, but is not yet properly justified.

The reviewer is right. To be honest, initially we understood the method more intuitively rather than rigorously, in a mathematical sense. We took the reviewer's concerns very seriously and invested a great effort in trying to mathematically formulate and study superlets. In this process, we also realized how many confusions are in the field of signal processing applied to neuroscience. We therefore set up an extensive Supplementary Information, with ample mathematical background that unambiguously and coherently (we hope!) explains the place of *superlets* in the world of time-frequency / time-scale methods. We also managed to analytically derive why superlets reduce redundancy for the higher frequencies and this derivation proved very valuable to other inferences, on ASLT for example, which, for a particular parameter instantiation is equivalent to the MMCE. The Supplementary Information has about 37 pages and spans 103 equations. We hope this will be a useful material for the broad readership with a basic mathematical background. The Main Text was kept light and easy to digest, while the "harder" mathematical aspects are presented in Supplementary Information.

Minor comments

In equation 2, where does the 5 come from?

The 5 comes from a design choice for the wavelets. We now explain this in the Main Text: “In eq. (2), we set B_c such that the plane wave spans c full cycles within 5 standard deviations of the Gaussian envelope. In practice, for the convolution one then considers a Morlet window of 6 standard deviations (see Fig. S1).” We also modified Fig. S1 to explain graphically the design choices.

Particularly in the simulations, too much of the analysis provided visually where I would prefer to get goodness of fit measures, since the answer is exactly known.

In addition to the analytical treatment, which provides exact quantitative results (see especially section “Superlets and redundancy suppression: towards multi-resolution TFRs” in the Supplementary Information), we also came up with quantitative tests, as suggested by reviewer #1. These are now presented in Figs. 11, 12, S9, and S10/11.

We thank the reviewer for “pushing” us to provide clear mathematical proofs. We believe that this greatly contributed to our understanding of the method and helped us better relate *superlets* to other techniques.

Replies to reviewer #3:

The authors develop an algorithm to, in simple terms, get the frequency resolution of a short-time fourier transform (STFT) and the time resolution of a wavelet transform (CWT). In their approach they average wavelets with increasing width geometrically, and when there is a need to cover a broader frequency range they make the wavelets adaptive. The algorithm is applied to examples, to determine the interference when multiple transient signals are present (close in time or close in frequency) and apply it to example experimental data and compare it to STFT, CWT and another approach to combine measurements, albeit in frequency space (MMCE). Overall, they show that their method, SLT or ASLT, outperforms the STFT and CWT in time and frequency resolution, and is a little bit crisper than MMCE. What struck me as very useful is the ability to do single-trial analysis, or be sensitive to transient events that occur on a few trials amongst the ones included in the average. This is a useful contribution to the analysis of experimental data for cases where there is variability in time and background state in the activity patterns (which is the case in most cognitive neuroscience experiments).

Overall, the paper provides a useful contribution to the analysis of the experimental neuroscience data.

Specific comments:

There are other ways of combining the wavelets (for instance, difference weights in additive and different powers in geometric), and one could formulate an objective function measuring temporal and frequency resolution of event detection to be optimized across an ensemble of transient events. Why is the choice used in this paper optimal? And for what is objective function is it optimal, what would I need to do when I value temporal resolution twice as important as frequency resolution?

This is a good point. Indeed, there are multiple ways to combine time-frequency representations. The original paper of Loughlin (1994) describes the difference between the

geometric mean and arithmetic mean. The geometric mean is the closest to the representations in the set from a cross-entropy (informational) point of view. The arithmetic mean is minimizing the sum of squared errors between the set of representations and their average. Weighted arithmetic means have also been used in the context of directionally smoothed Wigner-Ville Distribution banks [10]. These aspects are now discussed in the Supplementary Information, “MMCE and *superlets* as special cases of generic TFD/TSD classes”.

The arithmetic and geometric mean provide very different representations. We opted for the geometric mean initially out of intuition (see replies to reviewer #1), and then realized that this is optimal from an information theoretic point of view and discovered the early work of Cheung and Lim [11], extended by Loughlin et al. [12]. Intuitively (and also confirmed by testing), the geometric mean provides the very advantages super-resolution requires: if any wavelet in the set has a zero response, e.g. the shortest wavelet located between two temporally separated packets, this will have the consequence that the entire *superlet's* response will be “vetoed” out at that particular time-frequency location. This is strongly supported by results in new Fig. 12 (and Fig. S9), which outlines the clear resolution advantage of both the MMCE and *superlets* on sinusoidal packets. Mathematically, the difference between the geometric and arithmetic averaging of representations has been elegantly demonstrated by Loughlin et al. [12]. Also, the analytical derivations in the Supplementary Information demonstrate why the geometric mean provides strong “frequency sharpening” (see subsection “Redundancy suppression by superlets – analytical derivation”).

Regarding the choice of parameters for the *superlets* as a function of temporal versus frequency resolution, we should mention that in general the method is very robust and requires little tweaking on real data. The c_1 parameter controls temporal resolution, while the order controls frequency concentration. A small c_1 (1-3) is sufficient for all applications we tested, and the order depends on how much one would like to sharpen the representation and on the choice of c_1 . These issues are now explained in the Main Text, where we provide 3 examples of *superlets* for Fig. 3. Also, the Supplementary Information now provides the mathematical background to precisely calculate the resolution of the method.

The transient events in the model data have a length of 7 or 11 cycles — which in typical experimental data would probably be 2-5 cycles. Perhaps some of the Fig.s can be tested on a couple of values for the length.

Regarding the size of the packets, this is a more complicated discussion. In principle, both the MMCE and *superlets* can increase frequency resolution as long as the packets have a certain length – both these methods are bounded by the Rayleigh frequency, as we show in Fig. 2 and discuss in the Main Text. The great advantage of both methods is their excellent time resolution, which determines the time-frequency super-resolution. This is born out of the properties of the geometric mean. When it comes to detecting short transients in the upper frequency bands, another advantage of the SLT is that it is multi-resolution and does not “dilute” the representation of short packets. The CWT has the same properties, but its frequency resolution is so bad in the upper frequencies that one cannot resolve the spectrum. By contrast, the MMCE is based on a fixed set of windows and dilutes the higher frequency, shorter packets. Only the SLT shares the advantage of both the CWT (multiresolution) and of the MMCE (good time and frequency resolution due to the geometric mean).

We considered the example in Fig. 9 and tested what happens with shorter packets, e.g. 5 cycles (see the Fig. below). The SLT was computed on the same data ((A) is taken from the main text, for comparison), and on data where we inserted the shorter, 5-cycle packets (B). As one can see, it is no problem for the method to detect the shorter packets in the high frequency range, although, as expected, the frequency localization is worse (as discussed above). The lowest frequency packet @40 Hz, although visible, suffers from interference with the lower frequency fluctuations that have high. Nevertheless, for this example, one should keep in mind that this is an extreme case: the packets were inserted into a single trial out of 84! Considering this, the results are quite good.

Page 10 — bottom: Fig2B/Fig2C should refer to Fig. 3 instead

The reviewer is right. We have fixed the typos.

Finally, we would like to also thank the third reviewer for the insightful and encouraging comments! We have struggled really hard to revise the paper such that all these concerns are adequately addressed by the precise mathematical derivations in the Supplementary Information and by the new results of the empirical and analytical testing.

References

- [1] O. Rioul and P. Flandrin, "Time-scale energy distributions: a general class extending wavelet transforms," *IEEE Transactions on Signal Processing*, vol. 40, no. 7, pp. 1746–1757, Jul. 1992, doi: 10.1109/78.143446.
- [2] P. Flandrin, R. G. Baraniuk, and O. Michel, "Time-frequency complexity and information," in *Proceedings of ICASSP '94. IEEE International Conference on Acoustics, Speech and Signal Processing*, 1994, vol. iii, p. III/329-III/332 vol.3, doi: 10.1109/ICASSP.1994.390031.
- [3] P. Flandrin and O. Rioul, "Affine smoothing of the Wigner-Ville distribution," in *International Conference on Acoustics, Speech, and Signal Processing*, 1990, pp. 2455–2458 vol.5, doi: 10.1109/ICASSP.1990.116088.
- [4] A. Grossmann, R. Kronland-Martinet, and J. Morlet, "Reading and Understanding Continuous Wavelet Transforms," in *Wavelets*, Berlin, Heidelberg, 1990, pp. 2–20, doi: 10.1007/978-3-642-75988-8_1.

- [5] S. Mallat, *A Wavelet Tour of Signal Processing: The Sparse Way*, 3 edition. Amsterdam ; Boston: Academic Press, 2008.
- [6] L. Stanković, "A measure of some time–frequency distributions concentration," *Signal Processing*, vol. 81, no. 3, pp. 621–631, Mar. 2001, doi: 10.1016/S0165-1684(00)00236-X.
- [7] B. Boashash and V. Susic, "Resolution measure criteria for the objective assessment of the performance of quadratic time-frequency distributions," *IEEE Transactions on Signal Processing*, vol. 51, no. 5, pp. 1253–1263, May 2003, doi: 10.1109/TSP.2003.810300.
- [8] N. Delprat, B. Escudie, P. Guillemain, R. Kronland-Martinet, P. Tchamitchian, and B. Torresani, "Asymptotic wavelet and Gabor analysis: extraction of instantaneous frequencies," *IEEE Transactions on Information Theory*, vol. 38, no. 2, pp. 644–664, Mar. 1992, doi: 10.1109/18.119728.
- [9] I. Shafi, J. Ahmad, S. I. Shah, and F. M. Kashif, "Techniques to Obtain Good Resolution and Concentrated Time-Frequency Distributions: A Review," *EURASIP J. Adv. Signal Process.*, vol. 2009, no. 1, p. 673539, Jun. 2009, doi: 10.1155/2009/673539.
- [10] P.-L. Shui, H.-Y. Shang, and Y.-B. Zhao, "Instantaneous frequency estimation based on directionally smoothed pseudo-Wigner-Ville distribution bank," *Sonar Navigation IET Radar*, vol. 1, no. 4, pp. 317–325, Aug. 2007, doi: 10.1049/rsn:20060123.
- [11] S. Cheung and J. S. Lim, "Combined multiresolution (wide-band/narrow-band) spectrogram," *IEEE Transactions on Signal Processing*, vol. 40, no. 4, pp. 975–977, Apr. 1992, doi: 10.1109/78.127970.
- [12] P. Loughlin, J. Pitton, and B. Hannaford, "Approximating time-frequency density functions via optimal combinations of spectrograms," *IEEE Signal Processing Letters*, vol. 1, no. 12, pp. 199–202, Dec. 1994, doi: 10.1109/97.338752.

Reviewers' comments:

Reviewer #1 (Remarks to the Author):

In this resubmission (revision of NCOMMS-19-31444-T), the authors have toned down their claims somewhat and added new material, including a 40+ page supplement, to address some of the concerns raised previously by the reviewers. That is appreciated, although some of my more significant previous concerns remain, especially with regard to fair and accurate quantitative comparisons. The issues of proper normalization and matching (minimum) resolutions is particularly problematic; when done properly, as shown here, **there are no resolution advantages of an MMCE combination of wavelets versus an MMCE combination of spectrograms**. The new material also raises additional concerns with regards to inaccuracies and misconceptions.

Moreover and importantly, much of the new material now makes this manuscript more of a “signal processing” paper than a neuroscience paper. As the authors themselves note in their rebuttal, the underlying theory and math are likely beyond the general readership of this journal. Hence, I question the appropriateness of *Nature Communications* for the publication of this manuscript, and feel that it would be more appropriately vetted by, and published in, the signal processing community (especially the vast majority of the supplemental material).

Having said all of that, I feel that there could potentially be a valuable contribution here to the neuroscience community, but it will require a substantial re-write and change of focus. Specifically, rather than writing the paper from the perspective of a new method that beats all others (since it does not), the authors may wish to consider introducing super resolution time-frequency methods in general to the community, with a clean application to neural data to illustrate the basic ideas. As appropriate, they could potentially discuss and contrast fixed-bandwidth v. constant-Q analysis and when one approach might be more appropriate than the other. In particular, constant-Q (wavelet) analysis has been applied to so-called ‘long memory processes,’ which have slowly decaying auto correlation functions [1]. Perhaps there is some relevance here to neural data (questions of stationarity notwithstanding). Note, though, that any advantage is not one of ‘resolution’ of constant-Q v. fixed-bandwidth, as shown below.

I summarize here my major concerns, and then expand on each below.

1. Coverage of similar past work on combining time-frequency representations is still lacking.
2. Fair and quantitative comparisons is still lacking. The issues of proper normalization and matching (minimum) resolutions is particularly concerning here; without it, you are comparing apples to oranges and can not make claims that one method is better or worse than another. With it, there are no resolution advantages of MMCE combinations of wavelets over spectrograms, as shown below.
3. The detection results are incomplete, raise concerns, and should be removed.
4. The supplemental material contains errors and misconceptions, and much of it is beyond the readership of this journal. Moreover, yet another review of time-frequency analysis and wavelets is not needed, but any such review should be fully vetted by experts in that community.

Additionally, I have minor concerns about misconceptions about the uncertainty principle as it pertains to time-frequency resolution that should be corrected.

1. **Placing MMCE wavelets (“superlets”) in appropriate context.** The authors now make comparisons to other methods of time-frequency analysis. However, they still present “superlets” as a significant new idea and improvement over any other method available, especially for the analysis of biological signals.

- a) As I noted previously, ‘super-resolution’ time-frequency analysis via optimal combinations of time-frequency representations is decades old, using different optimality criteria and different representations [2]. The authors continue to focus on the MMCE combination of spectrograms, but I think the other methods should be cited.
- b) I think it is a misnomer to refer to the MMCE combination of spectrograms as the “MMCE” method; MMCE is an optimality criterion, and it has been applied to combine representations other than spectrograms. The authors use it to combine wavelet transforms; as such, their superlets are also an “MMCE” method.
- c) With regard to the Choi-Williams and other modern methods of time-frequency analysis, the authors claim that “these techniques have not found significant traction in the analysis of biological signals.” This statement is simply not true, as a quick google scholar search of “choi-williams biomedical” will reveal, with hundreds of papers dealing with analysis of a variety of biomedical signals such as EEG, ECG, EMG, heart sounds, etc.

2. **Fair and accurate quantitative comparisons.** Although the authors include additional comparisons to more modern methods (mostly in the supplement), I still have serious concerns, especially with regard to proper normalization; matching minimum resolutions between the wavelets and the spectrograms that are being combined; and the quantitative measures that I previously suggested. The authors can’t have it both ways: they can not include comparisons that favor their approach and dismiss those that don’t.

a) MMCE spectrogram comparisons. The authors resist including additional MMCE-spectrogram comparisons in the body of the paper as previously suggested, instead relegating these to the supplement (which many readers will skim at best, if they even look at it). They defend this decision under the premise that the basic neuroscientist is familiar with spectrograms but “may not be very technically knowledgeable about very advanced time-frequency techniques.” I think if the reader is savvy enough to understand the MMCE combination of wavelets, they will have no trouble understanding the MMCE combination of spectrograms.

b) The marginals and proper normalization. The authors downplay quantitative comparison of the marginals of the superlets and MMCE to the true temporal and spectral densities of the signal. They cite a paper by Flandrin (ICASSP ’94) in which they claim he points out that “marginals aren’t very useful for the characterization of a time-frequency representation.” First off, Flandrin makes no such claim in that paper. But even if he had, one can find other papers making the opposite point (e.g., [3]).

The point is as follows [3, 4]: **If** one wishes to make the claim that some time-frequency function $P(t, \omega)$ shows the distribution of the energy of a signal $x(t)$ over time and frequency, then it better be the case that the total energy of the representation, $E = \iint P(t, \omega) dt d\omega$, equals the total energy of the signal, $E_x = \int x(t)^2 dt$. Then, the marginal $P(t) = \int P(t, \omega) d\omega$, compares to the known and true temporal energy it is fair and appropriate to ask how the energy over time of the representation, i.e., density of the signal, $x(t)^2$; and how the energy over frequency of the representation, i.e., the marginal $P(\omega) = \int P(t, \omega) dt$, compares to the known and true spectral energy density, $\frac{1}{2\pi} \int x(t) e^{-i\omega t} dt$. If the total energy and marginals are erroneous, then to

claim that $P(t, \omega)$ is an accurate representation of the joint distribution of signal energy over time and frequency is questionable at best. (Of course, the converse is not true: while the marginals are an important characteristic, they do not fully characterize the joint distribution.)

The authors further dismiss the value of such comparisons by noting that the normalization of the superlet is not the same as the MMCE spectrogram combination. Well, then they are comparing apples to oranges, and to say that one is better or worse than the other is meaningless! They can easily (and should) renormalize the superlet, just as the MMCE-spectrogram should be normalized [2, 5], so that they sum to give the total signal energy. With that, comparison of the marginals of each representation, including quantitative measures of the error, is appropriate and necessary to substantiate claims that superlets are more accurate than MMCE-spectrograms and other methods.

Fig. 8 is particularly troubling in this regard. The superlet indicates higher energy over the time-frequency regions labeled (1), (2) and (3) than does the MMCE-spectrogram. Accordingly, at least one of these representations is misleading; the authors believe it to be the MMCE-spectrogram, but on what evidence they base this belief is unclear. A quantitative comparison of the marginals of each representation will address this issue (assuming, again, one normalizes each representation properly, so that comparisons are appropriate and meaningful): which marginals are closer to the true signal densities? Going back to the previous point, if the marginals of representation A are less accurate than those of representation B, how can one claim that A is a more accurate representation of the joint energy than B? [3]

- c) **Matching temporal and spectral resolutions.** As I mentioned in my original review, for fair comparisons, it is important that one combines representations that span the same time and frequency resolutions. In particular, the maximum and (especially) minimum resolutions (e.g., standard deviations) in time and frequency of the set of spectrogram windows should match those of the set of wavelets. The authors replied that this is difficult to do because the resolution of wavelets scales with frequency. While it is true that the variances of the wavelet scale with frequency, it is nevertheless still possible to match the min and max resolutions of the set of wavelets to equal those of the spectrograms over the range of times and frequencies of interest. To do otherwise is again comparing apples to oranges: if the minimum resolution of the wavelets is better than that of the spectrograms, then of course the MMCE wavelet combination will have better resolution than that of the MMCE spectrogram combination.

Significantly, as shown here, if the minimum resolutions are matched, then the supposed resolution advantage of MMCE-wavelets over MMCE-spectrograms disappears; in fact, because of the constant-Q nature of wavelets, the MMCE-wavelets will have poorer resolution with increasing frequency.

Specifically, let the “mother wavelet” $\psi(t)$ have temporal standard deviation σ_t and spectral standard deviation σ_ω . Then at the frequency $\omega = \omega_0$, the scaled wavelet $\psi(t/a)$ will have standard deviations $\sigma_t(\omega_0) = \frac{\omega_m \omega_0}{\omega} \sigma_t$ and $\sigma_\omega(\omega_0) = \frac{\omega_0}{\omega_m \sigma_\omega}$ (where we

have expressed the scale parameter in terms of frequency by $a = \omega_m/\omega$). Accordingly, across the set of wavelets being combined, the minimum frequency resolution will be given by the minimum of the set of $\sigma_\omega(\omega_0)$ at the lowest frequency ω_0 of interest, and the minimum temporal resolution will be given by the minimum of the set of $\sigma_t(\omega_0)$ at the highest frequency of interest. The spectrogram windows should be so chosen to match these minimum resolutions.

I have done just this for the 30 Hz + 80 Hz example in the Supplement of the manuscript, using Gaussian windows for the spectrograms and Gabor wavelets (eq. (S20) in the manuscript). Fig. 1 plots narrowband and wideband spectrograms and their MMCE combination, as well as two wavelet analyses and their MMCE combination, for this signal. Since the spectrograms and wavelets do not change over time for this signal (assuming the cross-terms are negligible), results at one time are shown. In this figure, the peak values are normalized to 1 to aid visual comparisons of resolutions.

Figure 1: Wavelets, spectrograms and MMCE combinations for equal-strength 2-tone example of Fig. S4 in manuscript, with peak values normalized to 1. The resolutions of the spectrograms and wavelets were chosen here to be comparable; specifically, the maximum width of the wavelets (dashed blue at $f = 80$ Hz, left) matches that of the spectrograms (dashed blue, right), and the minimum width of the wavelets (dashed black at $f = 30$ Hz, left) matches that of the spectrograms (dashed black, right). With these comparable resolutions, the MMCE combination of spectrograms yields better resolution than that of the wavelets (dotted v. solid red at $f = 80$ Hz, bottom), because of the constant-Q nature of wavelets.

In this example, care was taken to match the minimum and maximum resolutions of the wavelets with those of the spectrograms, as should be clear from the plots. In particular, the individual wavelets were chosen so that one has a reference frequency matching the tone at $f_1 = 30$ Hz, and the other has a reference frequency matching the tone at $f_2 = 80$ Hz. The spectrogram resolutions were chosen to match the minimum and maximum resolutions from these two wavelets; namely, letting σ denote the resolution of the mother wavelet, then the narrowband spectrogram was computed with a Gaussian window with spectral width $\sigma_{NB} = f_2 \sigma$, and the wideband spectrogram

used a Gaussian window with $\sigma_{WB} = f_2 f \sigma$.

As apparent from the plots, this resolution matching results in spectrograms having comparable min and max resolutions as the wavelets, over the set. As further seen in the plots, this resolution matching negates any resolution advantages of the MMCE combination of wavelets. Indeed, because the MMCE combination preserves the constant-Q nature of wavelets, the resolution decreases as frequency increases, such that the resolution at frequency $f = f_2$ is inferior to that of the MMCE spectrogram.

Figure 2: Same example as in fig. 1, but each representation is now normalized to the total signal power. Since the signal power is totally concentrated at $f = 30$ Hz and $f = 80$ Hz for this 2-tone example, it is clear that the error in the frequency marginal of the MMCE combination of wavelets is greater than the error in that of the MMCE combination of spectrograms (bottom, solid v. dashed red).

This example further allows us to address the normalization and marginals issues discussed in the previous point. As plotted in Fig. 1, the total power of each representation is not the same. If we normalize the power across representations,¹ so that each one sums to the true signal power, the results are as shown in Fig. 2. Given that the true spectral density is concentrated at $f = f_1$ and $f = f_2$ for this example,

¹We normalize power here rather than energy since the energy is infinite for this infinite-duration 2-tone signal.

it is clear that the error in the spectral marginals is greater for the MMCE combination of wavelets than for the MMCE combination of spectrograms. Hence, with proper normalization and matching resolutions, the advantages of MMCE-wavelets over MMCE-spectrograms is questionable.

On the other hand, if one *does not* properly match resolutions across the set of spectrograms and wavelets, then one can readily (but incorrectly) conclude that wavelets yield “better” results, as illustrated in Fig. 3. In this example, the spectrogram combinations were kept as in the previous figures, but now the wavelets were chosen such that the minimum resolution of this set of wavelets was smaller than that of the narrowband spectrogram. Hence, the resolution of the MMCE-wavelets in this case is better at $f = f_1 = 30$ Hz here than in the MMCE-spectrogram case. But, this comparison is inappropriate as the resolutions of the underlying methods being combined were different to begin with.

Figure 3: Same example as in fig. 2, but now unmatched in minimum resolutions. Specifically, the minimum resolution of the wavelets (dashed black at $f = 30$ Hz, left) is smaller than that of the spectrograms (dashed black, right). In this case, one would conclude the MMCE-wavelet combination has better resolution than that of the spectrogram combination (solid v. dashed red, bottom), but that is only because the minimum resolution of the set of wavelets was smaller than that of the spectrograms. (Note vertical scales differ.)

3. Detection results. The detection results summarized in Fig. 11 of the manuscript are very problematic: many of the methods have poor performance for low SNR, which is rather odd and troublesome. Moreover, detection is characterized not only by the probability of detecting a signal when it is present (P_D), but also by the rate of false de-tectons/false alarms (P_{FA}). These vary with threshold settings, which is why I suggested previously that the authors consider a ROC curve analysis. The results presented here are operating at one (unknown) point on the ROC curve, and the corresponding P_{FA} is

likewise unknown/not reported. However, this issue is beyond the scope of this paper and deserves far more detailed consideration than given here, such that it would be best to remove these incomplete results from this paper.

4. **Supplemental Material.** As noted at the outset of my review, there are significant errors and misconceptions in the Supplemental Material. I note some of these below. More importantly, there are many review articles and books on time-frequency methods, such that the need for yet another one is highly questionable. But if it is to be published, it should be reviewed by experts in that field and published in a more appropriate journal.

- a) Eq. (s20) is a Gabor wavelet, not a Morlet wavelet.
- b) Eq. (s21) is *not* the Fourier transform of (s20): the Fourier transform of a Gaussian is a Gaussian, not a truncated Gaussian.
- c) Related to the previous point, it follows that Eq. (s28) is an approximation: it assumes that the term for $\omega < 0$ is negligible. For that to be the case, it requires that the ratio of the mean to the standard deviation of the Gaussian is greater than roughly 2.33 (for less than 1% error in the approximation). Similarly with (s38), (s43) and (s54).
- d) Eq. (s58) does not follow from (s57); they are equivalent only for $\sigma \delta \gg 2$.
- e) “*The time resolution of the scalogram is excellent.*” No, it isn’t: just as the frequency resolution decreases with increasing frequency, the temporal resolution decreases with decreasing frequency.
- f) “*computing the square root of the product of the two spectrograms...correlates the two spectrograms.*” No, it doesn’t: multiplying spectrograms is not the same as correlating them.
- g) “*MMCE still suffers from the fact that it is not a multi-resolution method*” and “*the SLT is a multi-resolution method, while the MMCE is not.*” These statements are not correct: the MMCE method combines spectrograms (or wavelets) of different temporal and spectral resolutions. Hence, it is a multi-resolution method, regardless of whether the underlying representations are fixed-bandwidth or constant-Q.
- h) “*It should be noted however that the UCP is only statistically sound when the distributions in time and frequency are unimodal and the first and second moments make sense.*” This and similar statement in the manuscript are not correct; many multi-modal distributions have well-behaved second moments. The u.p. applies to (the magnitude-square of) a function and (the magnitude-square of) its Fourier transform; as long as these decay faster than $1/t^2$ and $1/\omega^2$, respectively, the variances will be finite (and when they don’t, the product of the variances = ∞ which still exceeds the lower limit of the u.p.!).
- i) “*We introduce the uncertainty product single side, UCPS, as the uncertainty product only on the positive side of the spectrum.*” This is not a new idea [7].

Minor points

Heisenberg and time-frequency resolution. Finally, and relatively minor compared to the points above, but nevertheless important to correct, there are misconceptions about

the Heisenberg uncertainty principle (hereafter abbreviated u.p.) and time-frequency resolution. For example, the authors state, “Importantly, this limit applies to a single measurement.” This and other comments about the u.p. and time-frequency resolution need correction/clarification. In signal analysis, the u.p. is perhaps more properly called the duration-bandwidth theorem, because there is nothing “uncertain” (i.e., probabilistic) about it [6]: it states that a function and its Fourier transform can not both be arbitrarily narrow [7], which is typically quantified by the product of the temporal and spectral variance of the signal, $\sigma_t \sigma_\omega \geq \frac{1}{4}$. No matter how many times we measure a signal and compute its Fourier transform, this lower bound will not be violated; it applies to all measurements. To suggest that the u.p. can be overcome by multiple measurements is incorrect.

It is important to appreciate that the u.p. / duration-bandwidth theorem applies to a signal and its Fourier transform. If one applies a method of spectral analysis that is not the Fourier transform of the signal, then there is no inherent u.p. / duration-bandwidth limitation. As discussed previously, if the representation $P(t, \omega)$ yields the correct marginals – which are the magnitude-square of underlying Fourier transform pairs – then $P(t, \omega)$ satisfies the u.p. However, if the marginals are not related via a Fourier transform, then there is no inherent lower-limit; but that does not mean the method has broken or exceeded the u.p., because the u.p. does not apply.

With regard to combining multiple $P(t, \omega)$ to obtain a ‘superresolution’ result that breaks the u.p.: again, if the underlying marginals of this representation are not a Fourier transform pair, then the u.p. does not apply. If on the other hand the final representation is constrained to yield the correct marginals, then it will satisfy the u.p. in that the product of the temporal and spectral variances of $P(t, \omega)$ is equal to those of the signal. Note that when combining spectrograms or wavelets, the product of the variances of each representation is *greater* than that of the signal; the geometric combination reduces this product, but not below that of the signal. Again, one is not beating or exceeding the u.p. by combining multiple spectrograms or wavelets.

Local uncertainty products. I noted previously that a better measure of the time-frequency resolution of $P(t, \omega)$ are the local or conditional variances, as opposed to the global variances constrained by the u.p. [6]. The authors dismissed this suggestion, citing a paper they say shows it “is not applicable on multi-component representations, as the first and second moments become statistically flawed.” As with the claim about marginals and reference to the Flandrin paper, I do not see anywhere in this paper where the authors make this statement. The low-order global and local moments of the spectrogram and MMCE combination are well-behaved and readily computed for any measurable signal. They should also be computable for wavelets and superlets – if not, that’s a concern and limitation that should be noted [8].

References

- [1] D. Percival and P. Gottorp, “Long-Memory Processes, the Allan Variance and Wavelets,” in *Wavelet Analysis and Its Applications*, 1994.
(<https://doi.org/10.1016/B978-0-08-052087-2.50018-9>)

- [2]C. Detka, P. Loughlin and A. El-Jaroudi, "On combining evolutionary spectral estimates," *IEEE Seventh SP Workshop on Statistical Signal and Array Processing*, 1994. (<https://doi.org/10.1109/SSAP.1994.572489>)
- [3]L. Cohen and P. Loughlin, "The marginals and time-frequency distributions," *Proc. SPIE, Advanced Signal Processing Algorithms, Architectures, and Implementations XIII*, Vol. 5205, 2003. (<https://doi.org/10.1117/12.513899>)
- [4]L. Cohen, *Time-Frequency Analysis*. Prentice-Hall, 1995.
- [5]P. Loughlin, J. Pitton and B. Hannaford, "Approximating time-frequency density functions via optimal combinations of spectrograms," *IEEE Sig. Process. Ltrrs.*, vol. 1, no. 12, 1994. (<https://doi.org/10.1109/97.338752>)
- [6]P. Loughlin and L. Cohen, "The uncertainty principle: global, local or both?," *IEEE Trans. Sig. Process.*, 2004. (<https://doi.org/10.1109/TSP.2004.826160>)
- [7]G. Folland and A. Sitaram, "The uncertainty principle: A mathematical survey," *The Journal of Fourier Analysis and Applications* 3, 207238 (1997). (<https://doi.org/10.1007/BF02649110>)
- [8]L. Cohen, "The Wavelet Transform and Time-Frequency Analysis." In: L. Debnath (ed), *Wavelets and Signal Processing*. Birkhuser, Boston, MA, 2003. (https://doi.org/10.1007/978-1-4612-0025-3_1)

In brief, there are significant errors and misconceptions in this revised paper. As I show in my review, if one matches resolutions and normalizes appropriately (as I previously suggested), then there is no advantage of 'superlets' over previously proposed MMCE methods using spectrograms.

In addition, much of the material here, especially in the supplement, makes this more of a 'signal processing' paper than a neuroscience paper, such that I question the appropriateness of its publication in Nature Comms. The material, especially in the supplement, should be more fully vetted by experts in the signal processing / time-frequency analysis fields.

Reviewer #2 (Remarks to the Author):

Id like to thank the reviewers for taking my comments (and the other reviewer's comments) so seriously and thoroughly.

The authors have addressed my concerns and I would favor publication of the paper itself.

However, I have significant concerns about the publication of the supplemental materials in their current form. There are two reasons for this.

First, I think the analytic treatment in the supplemental material should be submitted as a separate paper in a different journal. They should be reviewed thoroughly on their own merits. I tried to do that here and confirmed that I agree with their basic approach, but I dont think its realistic to ask me to do a detailed analysis like this in a re-review. I would suggest the authors send it as a stand alone paper to a specialist journal like J Neuro Method, or a Signal Processing journal (the latter choice though

may make it invisible to the neuroscience community).

So I am saying section II and III of the supplement should be removed or largely reduced.

Second, the supplemental material needs to bring focus to the comparisons to existing techniques.

Finally equation 2 should have 5 as a parameter rather than as a number which gives it the appearance of a physical constant chosen by a higher power.

Reviewer #3 (Remarks to the Author):

The authors have revised the manuscript quite completely and added valuable mathematical motivation in the supplementary documents. My only comment would be to clarify the description of figure 12. State in the text that high values of the resolution measure are good and include in the caption the meaning of the shadings so we can understand what time resolution means. Perhaps change in the title of SI subsections Fundament  Foundation? Other than that I am satisfied with the revision.

NCOMMS-19-31444-T: Superlets: time-frequency super-resolution using wavelet sets

Vasile V. Moca, Harald Bârzan, Adriana Nagy-Dăbâcan, Raul C. Mureşan

Point-by-point reply to reviewers

We thank the editor and all reviewers for their insightful and very useful comments. We have struggled to address all the concerns and hope that the manuscript is now in a satisfactory shape.

Summary of the revision:

1. **Extensive mathematical treatment.** Our strategy for the revision was to keep the Main Text as light and easy to understand as possible, to make it accessible for a broad readership. Therefore, we deferred the mathematical details to an extensive Supplementary Information (SI), which now provides important insights, details, and discussions for those interested in the rigorous treatment of *superlets*. We struggled to write the mathematical derivation in SI as easy to follow as possible for readers with a minimal mathematical background. We feel that in many fields of science that use advanced signal processing methods, like neuroscience for example, there is still a superficial level of understanding and much confusion. In fact, even in the signal processing community some of the concepts related to time-frequency analysis are less clear than one may think. One of the significant issues related to *superlets* is the nature of the representation they provide and the way in which wavelets are normalized. In fact, we show that *superlets* are related to methods of wavelet ridge detection. We also go in great detail to set a general framework for understanding the relation between time-frequency (TFR) and time-scale representations (TSR), and we relate different techniques to general classes of representations. We hope the SI will help the average reader to get a coherent understanding of the “whats” and “whys” of time-frequency / time-scale analysis.
2. **Mathematical derivation of the superlets’ working principle.** Importantly, we have invested a significant effort in finding ways to analytically derive how *superlets* work. It turns out the method can be elegantly understood within the framework of band-pass filters. We show why *superlets* act as “frequency sharpeners”, “rejecting” redundant power and narrowing band-pass characteristics in the higher frequencies.
3. **Derivation of constant bandwidth superlets.** Using the analytical framework developed for *superlets*, we show that one can calculate the exact parameters of the ASLT to achieve constant frequency bandwidth. In fact, the constant frequency bandwidth ASLT is very similar to the MMCE method and provides very similar results. One may therefore consider *superlets* as a generalization that also encompasses the MMCE method as a special case.
4. **Dilution phenomenon in TFR/TSRs.** The SI contains a dedicated section where the difference between MMCE and *superlets* is discussed. We also provide the explanation for the “dilution” phenomenon, and show further examples based on Gaussian atoms.

5. ***Marginals and Uncertainty Product (UCP)***. An entire section of the SI is dedicated to the UCP, where we define it rigorously following Cohen's formalism. We also compute the TFR/TSR of various signals, from simple (atom, tone), to more complex ones (chirp, multi-packet landscape). We compare the representations and marginals computed with Wigner-Ville Distributions (WVD) and its directionally smoothed variants (Choi-Williams – CW; Born-Jordan – BJ), as well as *superlets* and MMCE. We show and discuss why marginals are not very useful to describing the quality or computing the resolution of a certain representation.
6. ***Relation to generic TFR/TSR classes***. We discuss the relation of MMCE and *superlets* to generic Cohen class / bilinear methods and to the Rioul and Flandrin's "affine" smoothed TSD class [1]. While we do not insist too much in this direction, we hope that the basic concepts we provide may prompt novel research trying to understand better how the MMCE and *superlets* may be put in the context of more general formalisms.
7. ***High-resolution methods on real data***. We now computed the WVD, CW, BJ representations for EEG and electrophysiology data. We provide the results in the Main Text. We should mention that these methods are not at all popular in neuroscience, and, after applying these on real data we also understood why. Biological data has a very rich and complex time-frequency landscape, such that methods that over-concentrate the representation become overwhelmed by cross-terms. Even methods like CW and BJ suffer from the directional smoothing to such extent that very obvious and well-known features of neuronal signals cannot be observed.
8. ***Detection accuracy and resolution measurements***. We devised detection accuracy and resolution measurements and applied them on Gaussian atoms and sine packets. As found by previous empirical testing, the results confirm that *superlets* are superior to other methods. For a single frequency, *superlets* and MMCE with Gaussian window can be matched to be equivalent. In fact, we show this relation in SI and also provide the conversion formula between the two representations, at a given, fixed frequency.
9. ***Improved clarity***. We struggled to also improve the clarity of the main text, by rephrasing here and there, and using a less ambiguous terminology. For example, to avoid confusion between the Morlet bandwidth parameter and the bandwidth of the wavelet (they are inversely proportional), we now use "time spread parameter" for the former and "bandwidth" for the latter. Although in the literature the bandwidth term is sometimes used inconsistently, referring to the bandwidth parameter, we feel this creates more confusion. We therefore resorted to a non-ambiguous terminology that is better grounded in wavelet theory.

We will provide our answers below to each point raised by reviewers. The reviewers' comments will be rendered in blue and our replies in black.

Replies to reviewer #1:

The authors present a method, called *superlets*, for obtaining “super-resolution” time-frequency representations of biological signals, by combining multiple wavelet transforms. They claim that, “superlets outperform ... other super-resolution methods ... with unprecedented precision” and “superlets can reveal transient oscillation events that are hidden ... by other methods.” These are extraordinary claims that need to be substantiated by quantitative and fair comparisons to the best methods available, as discussed below.

We agree with the reviewer and thank him for pointing us to the most relevant methods in time-frequency analysis. Many of these advanced methods, such as Choi-Williams (CW) and Born-Jordan (BJ), are rather unknown to neuroscientists and have witnessed, perhaps undeservedly, little attention in the community performing analysis of brain signals. However, there may be a good reason why these methods did not get traction in the analysis of biological signals – they perform rather poorly on these types of signals, as we learned during the revision.

To get a deep understanding, we implemented these methods ourselves, including the Wigner-Ville Distribution (WVD) and its directionally smoothed counterparts (CW and BJ) and we cross-checked with other mainstream tools, like Matlab. We also spent a significant effort reviewing the literature and relating to the outstanding work of Leon Cohen, and the very nice extensions by Patrick Flandrin and colleagues, or the excellent books by Boualem Boashash and Stéphane Mallat. Following the leads indicated by the reviewer, we were able to better relate our work to the significant advances in time-frequency analysis.

We found the comments of the reviewer extremely useful and struggled to address all of them in detail. In this process, we also understood our method better – including a thorough analytical treatment – and found ways to relate it more systematically to previous work. For a fairer relation to previous work, we also tempered the tone of the manuscript, as suggested by the reviewer. Please see our detailed answers below:

1. Quantitative comparisons to other high-resolution methods

- a) As the authors note, combining multiple time-frequency representations is not new. Different combinations have been proposed, using different optimality criteria as well as different time-frequency representations (e.g., in addition to ref. [17] in their manuscript, see [1] below). The authors consider just one of these methods. Hence, together with the next point, it is premature to claim their method yields “unprecedented precision” and uncovers time-frequency structure that no other method can.

We have removed the “hard” formulations from the Main Text and have added a greatly expanded Supplementary Information (SI), which explores in detail the place of *superlets* in the world of time-frequency / time-scale representations. We refer specifically to biological signals and show that, at least for neuroscience data, the new method is superior to others. The Main Text has 3 new Fig.s, while the SI has 8 new Fig.s with various results. We have tried really hard to compare our method as fairly as possible to many other techniques and have also derived analytically how it behaves. At least for neuroscience data, our conclusions hold – the *superlets* are unmatched by any of the methods we tested.

- b) For most of the examples (Figs. 1, 3, 4, 5, 6, S2), the authors compare their superlets to single spectrograms or wavelet transforms. Given that the time-frequency resolution of these methods is about as poor as it can be, it is not hard to do better than that! The combination of multiple spectrograms should be included in all cases, as should comparisons to other high-resolution methods, such as the Choi-Williams and RID time-frequency representations.

A first reason why most comparisons were made to spectrograms and scalograms is that these are overwhelmingly used in the analysis of brain signals today. As mentioned before, more advanced techniques, such as Choi-Williams and Born-Jordan have not witnessed widespread adoption in neuroscience. A second reason is that, for the general readership, explaining the limitations of the STFT and CWT and comparing them to higher resolution methods gives a strong intuitive understanding of what is going on. However, we do agree with the reviewer that a comparison to more advanced techniques is necessary. This is now extensively done (see Figs. 10, 11, 12, S10, and S11).

Anecdotically, we developed the *superlet* trying to fix the poor frequency resolution of the CWT for higher frequencies. More generally, our work is focused significantly on high-frequency gamma oscillations and we were not satisfied by the frequency resolution provided by the CWT. In fact, the idea to combine wavelets geometrically came out of intuition. Our first version of the manuscript is here: <https://www.biorxiv.org/content/10.1101/583732v1>. We only later learned about the MMCE, after independently inferring the usefulness of the geometric mean in spectral estimation.

Finally, comparing to the traditional STFT and CWT is also essential for the broad readership that may not be very technically knowledgeable about very advanced time-frequency techniques. In our opinion, using STFT and CWT as baselines, helps the broad readership understand how *superlets* (and other methods) work, and what are the limitations and challenges of time-frequency super-resolution in general. We therefore kept the main backbone of the paper but expanded it with new comparisons and also provided an extensive SI to frame our method rigorously.

- c) Additional quantitative comparisons are needed to substantiate their claims. Since there is no unique definition of the time-varying spectrum of a signal, we do not know “ground truth” (even for known signals that we generate, like the authors’ test signal). Hence, comparing one representation against another to draw conclusions as to which one best represents the (unknown) time-frequency structure of a signal is a challenge. However, one important comparison that should be made, for which we do know the ground truth, is to compare the marginal densities of each representation against the true marginals (*i.e.*, the magnitude-square of the signal and its Fourier transform). It would be helpful and informative for the authors to include plots of the marginals of each representation vs. the true marginals, as well as quantitative metrics of the error in each one.

The reviewer touches an important point here. We agree that, in general, it is hard to define a “ground truth” for the time-frequency spectrum of a signal. In the end, it all depends on how a certain representation is used to infer something about the properties of the signal. In neuroscience (and likely geoscience too) this is not so fuzzy: one would like to know where

oscillation bursts start, where they end, what is their frequency and magnitude/power. The reason is that these patterns of activity are generated by stereotypical neural circuits that produce well-known patterns in various frequency bands. What is particularly troublesome in neuroscience is the difficulty to distinguish between brief burst in both time and frequency, and to compensate for the sometimes overwhelming $1/f$ characteristic of the signal, which hinders estimation of higher frequencies. All this has to be done on signals that have a very rich time-frequency landscape and are plagued by noise and irrelevant activity. In that respect, two finite sine packets, with a sharp start/end and clear frequency have a rather clear “ideal representation” in the mind of the user, and the user expects to find such signature in the time-frequency representations of the signal.

Regarding marginals, we thank the reviewer for pointing us in this direction. Indeed, there is a large amount of work on this topic, especially by Cohen. We now added an entire section dedicated to marginals in the SI, and compute it on various representations, including the high-resolution ones like WVD, CW, and BJ. Unfortunately, as pointed out by Flandrin et al. (ICASSP '94)[2] and many other authors since, marginals aren't very useful for the characterization of a time-frequency representation. We provide a clear example where methods with beautiful marginals show a much poorer representation, including indistinguishable packets in time, than other representations whose marginals are not so sharp (see Figs. S10 and S11, “Neighbors” in SI).

- d) Widely used test signals for evaluating time-frequency methods should be included in addition to the authors' test signal, with quantitative comparisons. Namely, a complex-valued single tone burst, and a complex-valued single linear FM chirp: $s(t) = \exp(j(\omega_0 t + \beta t^2/2))$, $t_1 < t < t_2$ and $= 0$ otherwise. While even for these signals we do not know the true time-frequency structure, we do know the spectral density and the temporal density (*i.e.*, the true marginals), the instantaneous frequency, and that these signals are zero outside a finite time interval, all of which should be accurately reflected in a 'good' time-frequency representation of the signal.

These are all good points. We took the advice of the reviewer and computed the representations for all these signals with WVD, CW, BJ, MMCE, and ASLT. Results are shown in subsection “Marginals of high-resolution representations” and Figs. S10 and S11 in the SI. We would like to argue that methods which apparently behave optimally on simpler signals, may in fact be catastrophically impaired on real-world data – at least on some types of signals, like those originating from biological systems. Our results indicate just that: State-of-the-art time-frequency methods based on directionally smoothed WVD are mostly unusable on EEG and electrophysiology data. We hope that our analysis in Fig. S10 and S11 as well as the discussion throughout the SI shed more light on these issues.

- e) For fair comparisons, it is important that one combines representations that span the same time and frequency resolutions; *i.e.*, the maximum and (especially) minimum resolutions in time and frequency of the set of spectrograms should match that of the set of wavelets. In addition, the various representations in each set should all be normalized to match the total signal power. It is not clear that this was the case in the comparisons made (*e.g.*, in Fig. 7, the power of the MMCE result is at the low-end of the heat map, while the superlets result is at the high-end (especially the first column in A *v.* the first column in B)). Similarly, the authors' comments about the spectrogram and

MMCE “diluting” power as frequency increases, resulting in poorer detection compared to the superlet, is likely an issue of improper normalization: if the spectrogram window is properly normalized (to unit-energy), two different-frequency but equal-amplitude tone pulses will have the same peak power in the spectrogram.

In general, the MMCE and *superlets* cannot be matched, except at a single frequency – just like there is a single-frequency match between the spectrogram with Gaussian window and the Morlet-based scalogram. For the latter, we provide the exact matching formula (eq. s45). The reason why MMCE and SLT are not equivalent is that while one uses a single set of windows, the second uses a set of “windows” at each frequency. In addition, there are constraints to how small a window for MMCE one can choose, which are dictated by the lowest frequency (largest period) in the spectrum of interest. A too small window leads to aliasing. As a result, it is in general not possible to choose for the MMCE the “smallest” window that the SLT uses (at the highest frequency). These aspects are now discussed extensively in the SI “Interpretation of superlets and relation to MMCE”. We should mention that we always struggled to choose the MMCE parameters to get the closest match to the SLT. In addition, we now introduce the “constant bandwidth” ASLT, where orders are adaptively adjusted such that the frequency bandwidth of *superlets* is constant. This type of ASLT is almost identical to the MMCE, and therefore the ASLT may be viewed as a generalization whose special case is the MMCE.

Regarding normalization, this was properly done for all the transforms. For the spectrogram, we used unit energy windows and the integral of the representation was always the energy of the signal. One should note that even if the spectrograms are normalized, the integral of the MMCE will not yield the energy of the signal (because the sum of geometric means is not the geometric mean of the sums). Regarding scalograms, the normalization is in fact such an important issue, that we decided to dedicate it an entire section in the SI: see “Fundamentals of spectrograms and scalograms”. The scalogram does not integrate to the energy of the signal, but extra terms ($1/a^2$ or $1/a$, depending on the type of scalogram) have to be added to obtain the energy. The correct definition for the scalogram is given in Rioul and Flandrin [1], [3], in the original papers of Grossmann and Morlet [4], and countless papers and textbooks since, e.g. [5]. The scalogram does not preserve marginals [1] and its integral does not yield the energy, except when extra scaling is added in the integral. The correct implementation relies on the original definition, as is also done in Matlab and many mainstream tools.

Regarding Fig. 7, its label states how the color scales should be interpreted. These were not matched across methods but only across the 3 conditions for each method – i.e. the color intensity has the same interpretation only across A – left, across A – right, and the same for B – left and B – right. The purpose of the representation is to identify differences in the distribution of oscillation packets as induced by the experimental condition, not to determine how much total energy there is in the piece of signal. Indeed, the global scaling of the power is irrelevant for such applications, except when one attempts reconstruction from the representation. We should mention that the representations throughout the paper would not change at all if the representations were scaled, since the color scale always goes from min to max, and scaling has no effect. The only place where representations were matched across conditions (but not methods!) is Fig. 7. The “dilution” effect we mention has nothing to do with this global scaling – it refers to the relation between low and high frequencies.

Regarding the dilution phenomenon in the spectrogram, this is now discussed in detail in the SI (see Fig. S7 and the corresponding section in SI). The reviewer is right: two tone impulses with same amplitude but different frequency would have the same representation in the spectrogram. But this only holds if: either the two packets have equal duration, or each packet is longer than the size of the STFT window. In all examples we give, we introduce packets of equal cycle number (not equal absolute duration!) because this mimics the real case for biological data (and also for other types of data). In Fig. 7 the dilution appears precisely because oscillation bursts in higher frequencies have a shorter duration than those at lower frequencies and end up occupying just a fraction of the STFT window. This is precisely what has hindered the proper study of gamma oscillations for decades. The *superlets* come to solve exactly this problem and the paper is constructed fundamentally on these premises (see also Fig. 1, and all discussions in SI and Main Text).

- f) Regarding the detection results, given the statistical nature of detection and the dependence on SNR and threshold levels, one example is insufficient to claim one method is superior to another. Such claims need to be more extensively evaluated; for example, by computing ROC curves.

We fully agree! We now devised a novel test using Gaussian atoms and sine packets embedded in various amounts of noise. We measure how all the methods: WVD, CW, BJ, STFT, CWT, MMCE, ASLT fare on detecting the target signals as a function of the amount of added noise (see Fig. 11 and the corresponding description in Main Text).

2. Define and calculate time-frequency resolution, and clarify restrictions of the uncertainty principle

- a) It is important to define and quantify what is meant by the “time-frequency resolution” of the representation. The uncertainty principle constrains the product of a signal’s duration and bandwidth (commonly taken as the standard deviations in time and frequency); as discussed in [2], those are global quantities. One can calculate analogous local quantities, namely the standard deviation in time at a given frequency, and vice versa. In general, there is no uncertainty constraint on these local resolution quantities, even though the global quantities are constrained. However, for the spectrogram and other windowed-methods, including wavelets, these local quantities *are constrained* by the uncertainty principle. Hence, it would be valuable for the authors to calculate the global and local uncertainty product for the various examples and representations.

We took the reviewer’s advice seriously and examined in detail the marginals of various representations. We also computed the uncertainty product (UCP) on various signals for the *superlet* and its composing wavelets (see section “Resolution of TFR/TSR representations” in SI). The result is interesting in that the UCP of the SLT is smaller than the UCP of each individual wavelet in its set, especially on sine packets (see Fig. S9). We also discuss the issues of UCP and how to compute it correctly, as discussed in the literature (see Fig. S8 and corresponding text). Finally, the local UCP is not very useful either to evaluate the quality of the representation, because the UCP is based on first and second moments of the distribution which are statistically ill-defined for multi-modal distributions (see Flandrin, Baraniuk et al. [2], or

Stanković [6]). Instead, we introduce the concept of resolution in the sense of discrimination between neighboring components, as is done in optics and as was originally defined by Rayleigh. The same path is taken by others for TFRs, like Boashash and Sucic [7], or Stanković [6]s. Fig. 12 in Main Text shows these new results.

b) The uncertainty product of the spectrogram and wavelets has been extensively studied, along with their marginals and moments [3]. In general, the results obtained with wavelets are vastly inferior in terms of these metrics compared to those for the spectrogram. As such, it is curious that the authors seem to be obtaining better results by combining wavelets v. spectrograms. Can the authors take into account the results of [3] and show that the combination of wavelets indeed yields better results than does combining spectrograms, in terms of these metrics? If this is indeed the case, it would be very valuable to explain why that is, given that the individual wavelets are worse than the individual spectrograms. (As noted previously, their comments about the spectrogram / MMCE “diluting” power may be an issue of improper normalization and hence would not explain the claimed advantages that emerge when combining wavelets.)

Regarding the UCP on spectrograms and scalograms (wavelets), it should be noted that representations based on scalograms depend critically on the type of wavelet normalization (see “Fundamentals of spectrograms and scalograms” in the SI). A special kind of normalization (used in many studies and implemented in Matlab as the *cwt* function) conserves the integral rather than the energy of the mother wavelet. This kind of scalogram was used extensively for wavelet ridge detection [5], [8] and is also the one we employ for *superlets*.

The MMCE with Gaussian window and the Morlet SLT are precisely equivalent at a single frequency (when windows are matched). The reason why the SLT outperforms the MMCE is because the former is multi-resolution in the wavelet sense (adapts to the scale of each frequency) while the MMCE is not (a fixed set of windows covers all frequencies). This is now clearly explained in SI, “Interpretation of superlets and relation to MMCE”. Also, the analytical derivations in the SI now help to better understanding how the SLT works and why its UCP is dramatically improved compared to single wavelets (actually, it has the UCP of the optimally-sized wavelet for a certain signal component). The results we obtained are certainly not flawed by any normalization issue. They are now backed up by robust mathematical formulations and extensive testing on various cases that we have performed under the guidance of the reviewer.

Finally, we should mention that we are really grateful to the reviewer for these interesting and eye-opening comments. Our efforts to address them improved the manuscript significantly, in our opinion. And helped us understand much better what *superlets* do and how they compare to other techniques.

Replies to reviewer #2:

This is quite an interesting paper on signal processing for neural data analysis. (It potentially has wider applicability in areas such as speech, but these are outside the scope of my review, and is not explicitly considered in the paper.)

One of the limiting factors in neural time series analysis is that many important neural events are best described by an optimal frequency range and time window. Conventionally these signals have either been analyzed using FFT based methods or wavelet methods, which make different fundamental trade-offs of time and frequency resolution and each can be performed with a different choice of time or frequency resolution (in FFT, by the window size, in wavelets, by the number of cycles) . The paper does a nice job of explaining this problem.

The authors revive, extend, and nicely quantify the resolution limits of a technique based on taking the geometric mean of time-frequency spectra with different time-frequency resolution to improve the information available (labelled "super-resolution"). That's probably overselling it, but the simulations and data analysis shown do support the claim of better time-frequency localization of signals. The original paper (Reference 16) did this based on two spectrograms obtained by FFT, the authors nicely optimize this method to multiple spectrograms based on wavelets.

Following the suggestions of reviewer #1 we have toned down the wording of the paper to let the results speak for themselves. Regarding the term super-resolution, this is meant to refer to the simultaneous time and frequency resolution of the method. The same terminology is used in a set of seminal reviews, like that of Shafi et al. [9]. We believe that the new results in the revision also justify the usage of the "super-resolution" label for *superlets*.

The paper is potentially impactful in neuroscience and perhaps other fields, but my enthusiasm is severely limited because the authors don't really make an attempt to explain mathematically why this should and does work. There is an oblique reference to multi-resolution imaging, but I would like to see a mathematical analysis.

Neuroscience suffers from new techniques being proposed without much theoretical underpinning. These techniques get popular but then ultimately die because no one really knows what they are doing. I think the technique proposed here is valuable, but is not yet properly justified.

The reviewer is right. To be honest, initially we understood the method more intuitively rather than rigorously, in a mathematical sense. We took the reviewer's concerns very seriously and invested a great effort in trying to mathematically formulate and study superlets. In this process, we also realized how many confusions are in the field of signal processing applied to neuroscience. We therefore set up an extensive Supplementary Information, with ample mathematical background that unambiguously and coherently (we hope!) explains the place of *superlets* in the world of time-frequency / time-scale methods. We also managed to analytically derive why superlets reduce redundancy for the higher frequencies and this derivation proved very valuable to other inferences, on ASLT for example, which, for a particular parameter instantiation is equivalent to the MMCE. The Supplementary Information has about 37 pages and spans 103 equations. We hope this will be a useful material for the broad readership with a basic mathematical background. The Main Text was kept light and easy to digest, while the "harder" mathematical aspects are presented in Supplementary Information.

Minor comments

In equation 2, where does the 5 come from?

The 5 comes from a design choice for the wavelets. We now explain this in the Main Text: “In eq. (2), we set B_c such that the plane wave spans c full cycles within 5 standard deviations of the Gaussian envelope. In practice, for the convolution one then considers a Morlet window of 6 standard deviations (see Fig. S1).” We also modified Fig. S1 to explain graphically the design choices.

Particularly in the simulations, too much of the analysis provided visually where I would prefer to get goodness of fit measures, since the answer is exactly known.

In addition to the analytical treatment, which provides exact quantitative results (see especially section “Superlets and redundancy suppression: towards multi-resolution TFRs” in the Supplementary Information), we also came up with quantitative tests, as suggested by reviewer #1. These are now presented in Figs. 11, 12, S9, and S10/11.

We thank the reviewer for “pushing” us to provide clear mathematical proofs. We believe that this greatly contributed to our understanding of the method and helped us better relate *superlets* to other techniques.

Replies to reviewer #3:

The authors develop an algorithm to, in simple terms, get the frequency resolution of a short-time fourier transform (STFT) and the time resolution of a wavelet transform (CWT). In their approach they average wavelets with increasing width geometrically, and when there is a need to cover a broader frequency range they make the wavelets adaptive. The algorithm is applied to examples, to determine the interference when multiple transient signals are present (close in time or close in frequency) and apply it to example experimental data and compare it to STFT, CWT and another approach to combine measurements, albeit in frequency space (MMCE). Overall, they show that their method, SLT or ASLT, outperforms the STFT and CWT in time and frequency resolution, and is a little bit crisper than MMCE. What struck me as very useful is the ability to do single-trial analysis, or be sensitive to transient events that occur on a few trials amongst the ones included in the average. This is a useful contribution to the analysis of experimental data for cases where there is variability in time and background state in the activity patterns (which is the case in most cognitive neuroscience experiments).

Overall, the paper provides a useful contribution to the analysis of the experimental neuroscience data.

Specific comments:

There are other ways of combining the wavelets (for instance, difference weights in additive and different powers in geometric), and one could formulate an objective function measuring temporal and frequency resolution of event detection to be optimized across an ensemble of transient events. Why is the choice used in this paper optimal? And for what is objective function is it optimal, what would I need to do when I value temporal resolution twice as important as frequency resolution?

This is a good point. Indeed, there are multiple ways to combine time-frequency representations. The original paper of Loughlin (1994) describes the difference between the

geometric mean and arithmetic mean. The geometric mean is the closest to the representations in the set from a cross-entropy (informational) point of view. The arithmetic mean is minimizing the sum of squared errors between the set of representations and their average. Weighted arithmetic means have also been used in the context of directionally smoothed Wigner-Ville Distribution banks [10]. These aspects are now discussed in the Supplementary Information, “MMCE and *superlets* as special cases of generic TFD/TSD classes”.

The arithmetic and geometric mean provide very different representations. We opted for the geometric mean initially out of intuition (see replies to reviewer #1), and then realized that this is optimal from an information theoretic point of view and discovered the early work of Cheung and Lim [11], extended by Loughlin et al. [12]. Intuitively (and also confirmed by testing), the geometric mean provides the very advantages super-resolution requires: if any wavelet in the set has a zero response, e.g. the shortest wavelet located between two temporally separated packets, this will have the consequence that the entire *superlet's* response will be “vetoed” out at that particular time-frequency location. This is strongly supported by results in new Fig. 12 (and Fig. S9), which outlines the clear resolution advantage of both the MMCE and *superlets* on sinusoidal packets. Mathematically, the difference between the geometric and arithmetic averaging of representations has been elegantly demonstrated by Loughlin et al. [12]. Also, the analytical derivations in the Supplementary Information demonstrate why the geometric mean provides strong “frequency sharpening” (see subsection “Redundancy suppression by superlets – analytical derivation”).

Regarding the choice of parameters for the *superlets* as a function of temporal versus frequency resolution, we should mention that in general the method is very robust and requires little tweaking on real data. The c_1 parameter controls temporal resolution, while the order controls frequency concentration. A small c_1 (1-3) is sufficient for all applications we tested, and the order depends on how much one would like to sharpen the representation and on the choice of c_1 . These issues are now explained in the Main Text, where we provide 3 examples of *superlets* for Fig. 3. Also, the Supplementary Information now provides the mathematical background to precisely calculate the resolution of the method.

The transient events in the model data have a length of 7 or 11 cycles — which in typical experimental data would probably be 2-5 cycles. Perhaps some of the Fig.s can be tested on a couple of values for the length.

Regarding the size of the packets, this is a more complicated discussion. In principle, both the MMCE and *superlets* can increase frequency resolution as long as the packets have a certain length – both these methods are bounded by the Rayleigh frequency, as we show in Fig. 2 and discuss in the Main Text. The great advantage of both methods is their excellent time resolution, which determines the time-frequency super-resolution. This is born out of the properties of the geometric mean. When it comes to detecting short transients in the upper frequency bands, another advantage of the SLT is that it is multi-resolution and does not “dilute” the representation of short packets. The CWT has the same properties, but its frequency resolution is so bad in the upper frequencies that one cannot resolve the spectrum. By contrast, the MMCE is based on a fixed set of windows and dilutes the higher frequency, shorter packets. Only the SLT shares the advantage of both the CWT (multiresolution) and of the MMCE (good time and frequency resolution due to the geometric mean).

We considered the example in Fig. 9 and tested what happens with shorter packets, e.g. 5 cycles (see the Fig. below). The SLT was computed on the same data ((A) is taken from the main text, for comparison), and on data where we inserted the shorter, 5-cycle packets (B). As one can see, it is no problem for the method to detect the shorter packets in the high frequency range, although, as expected, the frequency localization is worse (as discussed above). The lowest frequency packet @40 Hz, although visible, suffers from interference with the lower frequency fluctuations that have high. Nevertheless, for this example, one should keep in mind that this is an extreme case: the packets were inserted into a single trial out of 84! Considering this, the results are quite good.

Page 10 — bottom: Fig2B/Fig2C should refer to Fig. 3 instead

The reviewer is right. We have fixed the typos.

Finally, we would like to also thank the third reviewer for the insightful and encouraging comments! We have struggled really hard to revise the paper such that all these concerns are adequately addressed by the precise mathematical derivations in the Supplementary Information and by the new results of the empirical and analytical testing.

References

- [1] O. Rioul and P. Flandrin, "Time-scale energy distributions: a general class extending wavelet transforms," *IEEE Transactions on Signal Processing*, vol. 40, no. 7, pp. 1746–1757, Jul. 1992, doi: 10.1109/78.143446.
- [2] P. Flandrin, R. G. Baraniuk, and O. Michel, "Time-frequency complexity and information," in *Proceedings of ICASSP '94. IEEE International Conference on Acoustics, Speech and Signal Processing*, 1994, vol. iii, p. III/329-III/332 vol.3, doi: 10.1109/ICASSP.1994.390031.
- [3] P. Flandrin and O. Rioul, "Affine smoothing of the Wigner-Ville distribution," in *International Conference on Acoustics, Speech, and Signal Processing*, 1990, pp. 2455–2458 vol.5, doi: 10.1109/ICASSP.1990.116088.
- [4] A. Grossmann, R. Kronland-Martinet, and J. Morlet, "Reading and Understanding Continuous Wavelet Transforms," in *Wavelets*, Berlin, Heidelberg, 1990, pp. 2–20, doi: 10.1007/978-3-642-75988-8_1.

- [5] S. Mallat, *A Wavelet Tour of Signal Processing: The Sparse Way*, 3 edition. Amsterdam ; Boston: Academic Press, 2008.
- [6] L. Stanković, "A measure of some time–frequency distributions concentration," *Signal Processing*, vol. 81, no. 3, pp. 621–631, Mar. 2001, doi: 10.1016/S0165-1684(00)00236-X.
- [7] B. Boashash and V. Susic, "Resolution measure criteria for the objective assessment of the performance of quadratic time-frequency distributions," *IEEE Transactions on Signal Processing*, vol. 51, no. 5, pp. 1253–1263, May 2003, doi: 10.1109/TSP.2003.810300.
- [8] N. Delprat, B. Escudie, P. Guillemain, R. Kronland-Martinet, P. Tchamitchian, and B. Torresani, "Asymptotic wavelet and Gabor analysis: extraction of instantaneous frequencies," *IEEE Transactions on Information Theory*, vol. 38, no. 2, pp. 644–664, Mar. 1992, doi: 10.1109/18.119728.
- [9] I. Shafi, J. Ahmad, S. I. Shah, and F. M. Kashif, "Techniques to Obtain Good Resolution and Concentrated Time-Frequency Distributions: A Review," *EURASIP J. Adv. Signal Process.*, vol. 2009, no. 1, p. 673539, Jun. 2009, doi: 10.1155/2009/673539.
- [10] P.-L. Shui, H.-Y. Shang, and Y.-B. Zhao, "Instantaneous frequency estimation based on directionally smoothed pseudo-Wigner-Ville distribution bank," *Sonar Navigation IET Radar*, vol. 1, no. 4, pp. 317–325, Aug. 2007, doi: 10.1049/rsn:20060123.
- [11] S. Cheung and J. S. Lim, "Combined multiresolution (wide-band/narrow-band) spectrogram," *IEEE Transactions on Signal Processing*, vol. 40, no. 4, pp. 975–977, Apr. 1992, doi: 10.1109/78.127970.
- [12] P. Loughlin, J. Pitton, and B. Hannaford, "Approximating time-frequency density functions via optimal combinations of spectrograms," *IEEE Signal Processing Letters*, vol. 1, no. 12, pp. 199–202, Dec. 1994, doi: 10.1109/97.338752.

NCOMMS-19-31444-T: Superlets: time-frequency super-resolution using wavelet sets

Vasile V. Moca, Harald Bârzan, Adriana Nagy-Dăbâcan, Raul C. Mureşan

Revision #2

Point-by-point reply to reviewers

We thank the editor and all reviewers again for their insightful and very useful comments.

While we found the comments useful, we feel Reviewer #1 has misunderstood the purpose and scope of the *superlets* method. We struggled to clarify and improve the manuscript further and will provide here further solid mathematical proofs in support of our conclusions. We begin by clarifying once more the major claims of the manuscript. We will then move on to the summary of the revision and the point-by-point reply to reviewers' comments.

Major claims of the manuscript:

The method we introduced is based on a simple principle (even though the mathematics behind may be complex): to overcome the poor frequency resolution of wavelet analysis in the higher frequency bands by combining multiple wavelets in a set. In addition, it features a type of wavelet normalization (L^1 norm—see below) that favors **detection** of oscillation packets. This provides the following advantages:

- a) The method inherits the advantage of wavelets, benefitting from their excellent temporal resolution (relative to scale).
- b) Compared to single wavelets, the method gains frequency resolution in the upper bands (without breaking the theoretical uncertainty limit however).
- c) In the joint TF domain, the method offers superior resolution to classical methods—a property coined as time-frequency super-resolution in a number of papers and reviews.
- d) Due to its multiresolution (multiscale) nature, and because of L^1 normalization, *superlets* do not dilute oscillation packets which compress with scale in the higher frequency bands, enabling the detection of single-trial oscillation bursts in neural data—very important for neuroscience!
- e) We also show that the method can be flexibly generalized and that results provided by some of the other analysis techniques can be obtained from particular instantiations (special cases) of *superlets*. For example, the CWT corresponds to *superlets* of order 1, while the MMCE yields similar results to constant bandwidth adaptive *superlets*.

The manuscript shows that *superlets* behave well on electrophysiology brain data, which has a rich time-varying spectrum. They outperform high-resolution methods based on the Wigner Transform, which suffer extensively from cross-term problems, even with advanced kernels. The method most readily comparable to *superlets* is the minimum mean cross-entropy (MMCE) based on spectrograms, which was developed in the 1990's by Cheung and by extended by Loughlin et al. The MMCE uses a set of spectrograms with different window sizes, which are combined into a representation by computing their geometric mean. **The main arguments of**

Reviewer#1 for criticizing our work have revolved around the relation between MMCE and superlets. We will expand significantly on this issue below.

Summary of the revision:

1. **Significant shortening of the Supplementary Information.** Following the suggestions of Reviewer #2, we shortened the Supplementary Information (SI) by i) keeping the mathematical background and proofs to a minimum; ii) removing a section altogether (former section “IV. MMCE and superlets as special cases of generic TFD/TSD classes”). We kept only the parts that are essential for the reader to understand where the method stands in relation to other TFRs/TSRs and how it achieves frequency concentration.
2. **Increase in clarity and use of more appropriate terminology.** We struggled to simplify the mathematical derivations in section III of the SI and took a more straightforward route to calculate the response of the *superlet* to a tone. Results are identical, demonstrating once more that the mathematical derivation was correct. Also, in the Main Text, we defined the *superlet* in the complex domain, thus yielding a more general and simple definition. This way, the SLT of order 1 is in fact the CWT (with just a $\times 2$ scaling). In addition, extensions of the method are now very easy, because one can calculate the phase of the SLT and implement coherence or phase-locking methods, two important developments we intend to pursue. We also clarified several terminological issues, on which we dwell extensively below.

Before we dive into a complex mathematical argumentation, we would like to present an example of analysis on electrophysiology data (mouse V1), **with perfectly matched SLT (left) and MMCE (right) parameters**. The MMCE has the same window sizes as the shortest and longest wavelets (48 ms and 1680 ms), and uses L¹-norm, like the SLT. Results speak for themselves!

Figure 1. SLT (left) and MMCE (right) with matched parameters and identical normalization. Note the very close power scales.

Point-by-point replies to reviewers:

We will provide our answers below to each point raised by reviewers. The reviewers' comments will be rendered in blue and our replies in black. We will start with our replies to Reviewers #2 and #3, and defer our much longer answer to Reviewer #1 to the end of the document.

Replies to Reviewer #2:

I'd like to thank the reviewers for taking my comments (and the other reviewer's comments) so seriously and thoroughly.

The authors have addressed my concerns and I would favor publication of the paper itself.

However, I have significant concerns about the publication of the supplemental materials in their current form. There are two reasons for this.

First, I think the analytic treatment in the supplemental material should be submitted as a separate paper in a different journal. They should be reviewed thoroughly on their own merits. I tried to do that here and confirmed that I agree with their basic approach, but I don't think it's realistic to ask me to do a detailed analysis like this in a re-review. I would suggest the authors send it as a stand-alone paper to a specialist journal like *J Neuro Method*, or a Signal Processing journal (the latter choice though may make it invisible to the neuroscience community).

So I am saying section II and III of the supplement should be removed or largely reduced.

Second, the supplemental material needs to bring focus to the comparisons to existing techniques.

We agree with the reviewer that the SI was too long and too complex for the broad readership that is likely to target the paper. We now significantly shortened sections II and III. With the permission of the reviewer, we would like to keep these shortened sections as they seem critical to understanding of the method, for the following reasons.

Section II: "Fundamentals of spectrograms and scalograms" is now much shorter, with just three essential points: i) explaining the difference between TFRs and TSRs; ii) discussing the classical L^2 (energy) versus modified, L^1 (amplitude) normalizations, **which is absolutely critical to the message of the paper**; iii) introducing the modified, L^1 -norm scalogram, which is at the foundation of *superlets*. We tried to keep the mathematics in this section very light and simple.

Section III: "Superlets and redundancy suppression: towards multiscale TFRs" was also shortened and simplified. We came up with a more straightforward calculation of the SLT's frequency characteristic. We also defined here the constant absolute bandwidth *superlets* and this is very important because it compares directly with the MMCE. In addition, we discussed the relation between SLT and MMCE. **Section III is, in our opinion, the most important in the entire paper.** Without it, the solidity of the arguments in the Main Text would not be the same.

We removed former section IV. The new section IV discusses resolution issues in the framework of the uncertainty product (UCP). As we argue and conclude in this section, we find the UCP framework particularly problematic for understanding how the methods behave on real data, with rich time-frequency structure (as also indicated by other authors before us [1], [2]). Instead, **the Main Text focuses extensively on comparisons with other methods and attempts to**

do so in an appropriate framework (e.g., see Figures 11 and 12). We would like to respectfully point out that the comparisons to existing techniques are the focus of the Main Text.

Finally equation 2 should have 5 as a parameter rather than as a number which gives it the appearance of a physical constant chosen by a higher power.

We agree. We introduced a parameter called k_{SD} and explained that we fixed it to 5. This is a design parameter for wavelets (we also used it for Gaussian atoms) and it is never changed.

In fact, this parameter, together with the number of cycles of the wavelet, has important implications for the admissibility of the wavelet. We also discuss this in our response to Reviewer #1 (below).

We thank the reviewer for the insightful comments! We hope the reviewer agrees that Sections II and III of the SI are essential for the message of the paper.

Replies to Reviewer #3:

The authors have revised the manuscript quite completely and added valuable mathematical motivation in the supplementary documents.

My only comment would be to clarify the description of figure 12. State in the text that high values of the resolution measure are good and include in the caption the meaning of the shadings so we can understand what time resolution means.

We thank the reviewer for pointing this out. We updated the Main Text with the following explanation: "A high value of this resolution measure corresponds to a superior method, with the ability to better distinguish neighboring oscillation packets."

In the caption of figure 12, we wrote: "Time resolution was measured as the fraction of "empty space" in the cross-section of the representation, i.e., area of the shaded region divided by the area of the green box."

Perhaps change in the title of SI subsections Fundament  Foundation? Other than that I am satisfied with the revision.

We would like to keep the word "Fundament", as we feel this is closer to the meaning we intended. In the Meriam Oxford Dictionary, *fundament* is defined as "an underlying ground, theory, or principle"¹. Section II in the SI is supposed to present exactly that: the underlying theory / principle of spectrograms and scalograms. We hope the reviewer agrees.

We thank the reviewer once more!

Replies to Reviewer #1:

Please see the attached pdf file for my complete review.

¹ <https://www.merriam-webster.com/dictionary/fundament>

In brief, there are significant errors and misconceptions in this revised paper. As I show in my review, if one matches resolutions and normalizes appropriately (as I previously suggested), then there is no advantage of 'superlets' over previously proposed MMCE methods using spectrograms.

In addition, much of the material here, especially in the supplement, makes this more of a 'signal processing' paper than a neuroscience paper, such that I question the appropriateness of its publication in Nature Comms. The material, especially in the supplement, should be more fully vetted by experts in the signal processing / time-frequency analysis fields.

We respectfully disagree with the reviewer. We believe that the reviewer has not understood the main purpose and scope of the paper. We will attempt here to clarify these aspects once more by providing both numerical and analytical proofs in support of our claims.

From the outset, we would like to stress out the following points:

- a) The purpose of the method is to detect oscillation bursts (packets) in natural signals, where **the duration of these events compresses as frequency increases** (i.e., there is a multiscale expression of such events). This is the core construct of the manuscript.
- b) **The manuscript never claimed that the resolution of superlets is better than that of the MMCE.** In fact, they are very similar overall—but their time and frequency resolutions are different in different parts of the TF plane (see below, and Figure 12 in Main Text).
- c) The problem of the MMCE is the “dilution” of packets that have fixed amplitude but whose duration decreases with the increase in frequency. **This is NOT due to improper normalization!** We will prove this mathematically below.
- d) The *superlets* do not dilute the representation of such packets because **they use L^1 normalization of the wavelets** (also called amplitude normalization), which captures power at scale. If the reviewer is not familiar with this type of normalization, we respectfully suggest the reference Lilly 2017 [3], page 7 (Optimization principle) and Liu et al. 2007 [4]. Very importantly, the MMCE using L^1 normalization still dilutes the representation (see Fig. 1, on page 2, above)—this is the major difference between the MMCE and SLT!
- e) This brings us to the final point: the SLT and MMCE cannot be matched and are different techniques, useful for different applications. Below, we prove mathematically that **it is not possible to match the windows of the MMCE and SLT in general**, and that this is a bad idea.

We selected the major points from the reviewer’s document and will expand on them below.

1. Terminology

Multiresolution

The reviewer claims MMCE is a multiresolution technique—because it employs spectrograms computed with different window sizes. We argue that established terminology in the signal processing literature consistently uses the term “multiresolution” with a different meaning, i.e., to refer to the continuous adaptation of resolution to the scale of the process (see the 1988 paper of Ingrid Daubechies on the concept of multiresolution analysis [5]). The overwhelming

majority of papers and textbooks use multiresolution in the context of wavelets [6]–[9]—and that is also how we used the term.

For time-frequency analysis, multiresolution means using analysis windows that scale with frequency, **i.e. they adapt to the scale of each process**. More accurately, this can also be called **multiscale**. While this may be a minor terminology issue, we would like to point out that the reviewer considered this matter important. In time-frequency analysis, multiresolution analysis (as used systematically in the literature) means **continuously** adapting the analysis window to the frequency—a property not shared by the spectrogram, or by the combination of spectrograms for that matter! For a rigorous treatment of multiresolution, one should refer to the book of Barth, Chan, and Haimes, “Multiscale and Multiresolution Methods – Theory and Applications”, Springer-Verlag 2002 [8], and refer to the thousands of papers and textbooks on wavelets. Nevertheless, to avoid any kind of confusion, it is very easy to replace multiresolution by multiscale in the text of the paper—**we now systematically use multiscale throughout the manuscript and SI, because it has a less ambiguous meaning.**

MMCE

Indeed, the MMCE is a family of methods, which minimize mean cross-entropy. **However, for historical reasons and clarity, we kept this acronym to mean the geometric combination of spectrograms, as originally used in Loughlin et al. [10].** We clarified this in the text: “While any such combination of representations can be called MMCE, for historical reasons we will use the term MMCE to refer to the geometric combination of spectrograms throughout the rest of the paper.”

Morlet or Gabor

The reviewer claims we are misnaming the Gabor wavelet as Morlet. For the wavelet in eq. s15 in SI, we used the “Morlet” naming, as in [11] and countless other papers, for a simple reason: the Morlet wavelet is the general family of wavelets to which this one also belongs. **While also called Gabor wavelet, the formula in eq. s15 is in fact the modified Morlet wavelet**, used when the correction term to render the wavelet admissible is negligible. We have now clarified this in the Main Text “[...] and one can define the modified Morlet (also called Gabor) wavelet, as [...]”.

2. The resolution issue

The results of the paper show that the *superlets* and MMCE are different methods and do not provide identical results. **But the reason why *superlets* are superior is not resolution!** Our paper never claimed that, in general, *superlets* provide better resolution than the MMCE.

The joint time-frequency resolution of *superlets* and MMCE is in fact rather very similar, as we have shown in Figure 12, but the way this is expressed across the TF plane is different. This stems from the central difference between multiscale (multiresolution) and fixed scale analysis. Here’s why:

- a) The MMCE has a fixed **absolute temporal** and frequency resolution, being based on a set of spectrograms where, in each spectrogram, the **same analysis window is used for all frequencies**.
- b) The *superlet* (SLT) has variable temporal and frequency resolution, because it is based on wavelets – **this is the essence of multiscale analysis!** It has good frequency resolution at low frequencies and good absolute temporal resolution at high frequencies.
- c) Importantly, in the case of wavelets/SLT the **relative temporal resolution** is fixed. Indeed, the cycle duration relative to the “analysis window” of the wavelet is constant, being independent of frequency. **In a relative sense, wavelets do indeed have excellent temporal resolution**, as temporal resolution should be judged relative to the process’ characteristic timescale.
- d) **The relative temporal resolution of MMCE degrades with increasing frequency**: the cycle duration decreases relative to analysis window size. This is the MMCE’s weakness.

These multiscale concepts are the central construction axis of the paper!

In the last review report, the reviewer provided a set of three figures, computing the frequency response of the MMCE and *superlet* for two tone signals. As frequency increases, the MMCE provides better frequency resolution for a tone than the *superlet*. However, what the reviewer seems to miss is that **at the same time, the *superlet* provides increasingly better time resolution with increasing frequency**. **What matters in the end is the joint time-frequency resolution, as stated throughout the paper.**

It makes no sense to compare the resolution of different TFRs on tone signals. In the analytical derivation in the SI, we used tones to compute the frequency resolution of the SLT compared to single wavelets, not to compare the *superlet* with other TFR methods! The latter comparison only makes sense for data with time-frequency structure. For a single tone, one can make the resolution of the SLT arbitrarily high, by increasing the number of base cycles or the order. Moreover, it is pointless to use any of these methods on tones: there are dedicated frequency super-resolution techniques which can perform much better (see Discussions in the Main Text).

The reviewer also ignores the adaptive SLT (ASLT) we introduced, and in particular the constant bandwidth ASLT, which gives constant frequency resolution (see pages 19-20 in SI). The constant bandwidth ASLT yields very similar results to the MMCE. In this respect, *superlets* can also be viewed as a generalization across a wider class of methods.

While explained in great depth in the paper and SI, we depict here again the exact fundamental difference between the methods. To make the case relevant for time-frequency analysis, we consider the relevant signals, **which have both a temporal and a frequency dimension**: Gaussian atoms. **We show here that the TF resolution of the two methods is expressed differently across the TF-plane** but that, resolution-wise, there is no dramatic difference between methods.

For simplicity, we consider 2 atoms (10 cycles each) at 30 and 80 Hz, respectively, and the multiplicative SLT of order 2, with $c_1 = 5$. This yields two wavelets with 5 and 10 cycles. As the reviewer insisted, we matched the smallest and largest wavelet “windows” with the Gaussian window of the MMCE (see Table 1 below).

Already from this table, an obvious phenomenon, which was the subject of recurring contradiction with the reviewer, is that while the “fill factor” (window filled by atom signal) for the SLT does not depend on frequency, it drops dramatically for MMCE from low to high frequency (see Table 1, bottom, right). We called this the “dilution phenomenon” that plagues the MMCE. In the next section, we explain this in depth and derive an analytical proof.

Absolute window sizes				
Frequency [Hz]	SLT: Wavelet 1 size (6 s.d.) [ms]	SLT: Wavelet 2 size (6 s.d.) [ms]	MMCE: Window 1 size (6 s.d.) [ms]	MMCE: Window 2 size (6 s.d.) [ms]
30	200	400	75	400
80	75	150		
Relative window sizes				
Frequency [Hz]	SLT: Relative size Atom/Wlt. 1	SLT: Relative size Atom/Wlt. 2	MMCE: Relative size Atom/Win. 1	MMCE: Relative size Atom/Win. 2
30	2	1	5.33	1
80	2	1	2	0.375

Table 1. Absolute and relative window sizes for matched SLT and MMCE of order 2.

As stressed in the manuscript, neuroscience data (and other natural signals) contain oscillation packets whose duration scales down with increasing frequency—**hence overcoming the dilution phenomenon was the central motivation of developing the superlets technique!** We very much hope that now the reviewer takes into account this central point of the paper.

In Fig. 2, we show the TFR of the two atoms using the matched SLT and MMCE.

Figure 2. Comparison of time-frequency resolution of SLT and MMCE on Gaussian atoms. A. Raw signal with two Gaussian atoms, A1 @30Hz and A2 @80Hz. **B.** Sketch of Gaussian “window” sizes at the location of the atoms (amplitude not to scale). **C.** TFRs with SLT and window-matched MMCE. **D.** Peak-normalized (scaled to 1) time and frequency marginals for each atom. The s.d. for each is calculated (top-

right corner). **E-G.** Uncertainty products of the two methods for atoms spanning 10 (E), 7 (F), and 5 (G) cycles, respectively. Single-sided uncertainty products (UCPS) were calculated for each, isolated atom. Note: to obtain the UCP, the s.d. in frequency was multiplied by 2π (frequency expressed in radians).

Fig. 2A depicts the signals (atoms A1 and A2), and Fig. 2B shows a sketch with the actual extent of Gaussians corresponding to the SLT (left) and MMCE (right) at the target time-frequency locations (blue stars). Fig. 2C shows the actual TFRs, while Fig. 2D shows the time and frequency marginals on the two atoms, scaled to unit peak to facilitate comparison.

As expected, the MMCE dilutes A2 (Fig. 2C, bottom). Results in Fig. 2D indicate that, for low frequency (A1), the SLT has higher frequency resolution than MMCE, while this is reversed at high frequency (A2). Also, the SLT has a slightly better temporal resolution at high frequency (A2). The uncertainty product confirms these observations (Fig. 2E). We performed the same analysis for shorter atoms, with 7 cycles (Fig. 2F) and 5 cycles (Fig. 2G). **Results indicate that the two methods gain a slight advantage one over the other depending on the location of the atom in the TF plane, as well as on the temporal extent of the atom.**

As stated throughout the paper and SI, the SLT and MMCE are not very different in terms of resolution (see also Fig. 12 in Main Text). **The reason why the SLT is superior to MMCE pertains to its non-dilution property for scaling packets, because the SLT is multiscale and the MMCE is not** (see below).

3. The energy normalization debate (mathematical proof)

The central point of disagreement with Reviewer#1 relates to the window normalization for time-frequency representations. In particular, there are two major classes of normalizations: L^1 -norm, or unit modulus integral (also called amplitude normalization [3]), and L^2 -norm, or unit energy normalization. Both normalizations are used in signal processing. The L^1 -norm is useful for estimation of instantaneous power while the L^2 -norm ensures preservation of energy in the representation. To clarify why *superlets* use the L^1 -norm we devoted an entire section in the SI (pages 4-13).

We never claimed that the *superlets* method was supposed to be an energy-conserving time-frequency representation! As stated in countless textbooks and papers there are many quadratic time-frequency representations (QTFRs) that are useful even if they do not preserve energy or marginals. The most evident such representation is the scalogram. Contrary to the usage in [12], the scalogram is the square modulus of the CWT as coined by Flandrin and Rioul in 1990 [13], and defined in countless textbooks and papers [7], [13]–[15]. **The scalogram is not the square modulus of the CWT divided by a^2 .**

We do not understand why the reviewer insists that the marginals need to be preserved. **The integral of the scalogram is not the energy of the signal**, but further normalizations are necessary, to a^2 [see eq. (s10) in SI]. Normalizing the scalogram to recover energy generates a different representation, which is not the scalogram as defined and used by the overwhelming majority in the field. Indeed, it is clearly stated by Rioul and Flandrin [13] (page 1751): **“Because the smoothing function associated to the scalogram is itself a WVD, it cannot be perfectly**

concentrated in both time and frequency. As a result, several properties, such as marginals, are lost”.

Further, the reviewer insists on this normalization, arguing that otherwise the SLT and MMCE are not comparable. The reviewer also claims that the dilution phenomenon seen in MMCE but not in SLT is due to improper normalization of the latter. **We will show here that this claim is wrong—the dilution phenomenon in MMCE has nothing to do with normalization**, but it stems from the fixed-scale nature of the MMCE method. Also, we will show that even with identical L^1 normalization for both MMCE and SLT, the first method exhibits dilution while the second does not—see also Fig. 1, on page 2, above.

[REDACTED]

=====

[REDACTED]

=====

=====

=====

[REDACTED]

[REDACTED]

[REDACTED]

—
————
—
—
—
—
—————

[REDACTED]

==

— — —
— — —

[REDACTED]

[REDACTED]

— —

[REDACTED]

==== —

=====

— —

:[REDACTED]

—

[REDACTED]

[REDACTED]

[REDACTED]

As mentioned in the paper and SI, both L^1 and L^2 normalizations are used with the CWT, to obtain useful TSRs/TFRs. The same is true regarding STFT-based methods, where one can use L^2 norm to obtain the spectrogram (energy distribution) and L^1 norm to obtain instantaneous power. In particular, using the L^1 or L^2 norms does not make a difference for MMCE (except by introducing a global scaling), whereas it makes a big difference for wavelets (SLT), and we explained why (SI, pages 4-13).

As shown here, even when the MMCE uses an IDENTICAL (L^1) normalization to that employed by the SLT, the MMCE displays dilution, while the SLT does not. This stems from the fundamental difference of fixed scale versus multiscale analysis and is not a normalization mistake!

Indeed, as we have shown mathematically (eq. 25-29), for the MMCE (spectrogram, or any other STFT-based representation for that matter) the manner in which its windows are normalized has absolutely no involvement into the dilution phenomenon. The latter occurs because the windows do not adapt to scale as the frequency is increasing, nor does their normalization. By contrast, wavelets can be normalized such that they do not dilute, using L^1 -norm, which corresponds to the power captured by the wavelet (energy per scale or power at scale) [3]. **This provides a scale-invariant power representation.** STFT-based methods, like the MMCE, cannot achieve this scale-invariance, irrespective of their normalization. This central point is very clear in the manuscript and SI. We hope very much that the reviewer now acknowledges this point!

4. Why matching *superlets* and MMCE windows is generally impossible

We argued that matching the size of the MMCE windows with the “effective windows” of the smallest and largest wavelets in the *superlet* is generally not possible—and a bad idea! The reviewer disagrees, so we will explain here why the reviewer is wrong.

There are multiple reasons why matching the analysis windows across SLT and MMCE is problematic. Consider again the example in Fig. 2. First, the two methods can be made equivalent at a single frequency – if we chose the window sizes to be 75 and 150ms, the SLT and MMCE would obviously give identical results at 80Hz (if the normalization of the window is identical—see section 3 above). Second, across a frequency range, one can choose two different reference frequencies (like in Fig. 2) and calibrate the smallest and largest windows of MMCE to match the shortest and longest wavelet of the SLT at these two frequencies. **The problem is, this match does not hold at other frequencies.** For the case in Fig. 2, at 10Hz (lowest frequency) the SLT has “windows” of size 600 and 1200ms, while the MMCE uses 75 and 400ms, thus not even overlapping.

Finally, and most importantly, the match of MMCE with SLT can be ill-posed when the range of explored frequencies is large. Take for example the TFRs in Fig. 7 in the main text, that span a range of frequencies between 8-150Hz. The shortest wavelet (3 cycles) is located at 150Hz and spans (6 standard deviations) only 24ms! **We hope it is obvious that using a window size of 24 ms as the smallest window of the MMCE to estimate power at 8Hz (125ms period) is a bad idea!** By comparison, at 8 Hz, the shortest wavelet in the SLT spans 450 ms.

5. Mathematical derivations and their accuracy

The reviewer expresses concerns about the formula we used to compute the Fourier Transform (FT) of the wavelet (eq. s16 in SI). The Heaviside function in the computation of the wavelet’s FT can be found in many papers, for example [11], and as early as 1992 [16] (a paper with 3343 citations!). **The reason Heaviside is used stems from the admissibility criterion, which ensures that the Gaussian is located well in the positive domain of the spectrum and that the negative side of the spectrum is negligible.** As the frequency of the wavelet decreases, so does its variance in the Fourier spectrum, such that admissible modified Morlet (Gabor) wavelets always have negligible components in the negative side of the spectrum. We thought this was obvious—but

for clarity we also included the proper citation to [11] when we introduced the wavelet (eq. s15) and its Fourier transform (eq. s16). Now we explain this in the SI: “where, $H(\omega)$ is the Heaviside step function. The latter reflects the fact that the wavelet’s Fourier transform is a Gaussian shifted to ω_m in the positive part of the frequency spectrum (since ω_m is positive and should be large enough to render the mother wavelet admissible). Indeed, due to the admissibility criterion one can neglect the negative side of this wavelet’s spectrum [16]”.

The reviewer claims eq. s51 (former s58) is wrong, when in fact it is correct—one can easily compute this using the so-called “square pyramidal number” formula², a particular case of Faulhaber's formula. **Importantly, the mathematical derivations in the SI were performed independently by two of the authors, and results were checked and cross-checked, as well as validated numerically!**

6. ROC analysis – detection theory

The reviewer provides self-contradictory arguments regarding the application of detection theory to determine performance of different techniques. **First of all, ROC analysis is well-defined if one knows what one is looking for in the signal, i.e. the detection target is known.** The big problem here, as specified by the reviewer, is that *<<Since there is no unique definition of the time-varying spectrum of a signal, we do not know “ground truth” (even for known signals that we generate, like the authors’ test signal).>>* During the revision, we brainstormed extensively about how to implement ROC analysis to compare TFRs with different methods. **We concluded that since one does not know the target (ground truth), quantifying its presence using a binary “present/absent” decision is impossible—therefore, traditional ROC analysis, as requested by the reviewer, is impossible to apply.**

We thus resorted to a much simpler, but more relevant setup where we measured how a target oscillation packet is progressively masked by a random background—we called this “detection score”. We argue that this measure is not “detection probability”! The detection score is a statistical measure that simply determines the fraction of values within the mask which fell above the 95th percentile of the distribution of all values in the representation. This score decreases continuously as background noise is progressively added. **This is indeed not ROC, as it says nothing about the “perception” of a detector (with definable true and false positives, for that matter), but rather a simple, robust, and telling statistical measure.**

The setup and measure we provided illustrates beautifully why the *superlets* provide better results on neuroscience data: they concentrate the power bursts (at scale!) optimally, such that random background provides the least masking of all the tested methods. Furthermore, the reviewer claims *“The detection results summarized in Fig. 11 of the manuscript are very problematic: many of the methods have poor performance for low SNR”*—we do not understand what the reviewer means by “very problematic”? **We argue that the results in Fig. 11 in fact summarize and quantify very well what the other analyses show throughout the rest of the paper (e.g., Figs. 3-10).** We therefore consider that the detection score, as defined, and the test setup, as presented, are very relevant for the comparison of the different methods.

² https://en.wikipedia.org/wiki/Square_pyramidal_number

7. High-resolution methods in neuroscience

Our results indicate that high-resolution methods based on the Wigner-Ville Distribution (WVD), such as the Choi-Williams Distribution (CWD) perform rather poorly on rich spectrum neurophysiology data, such as EEG and intracortical LFP. The application of such methods on brain data is scarce at best. We claimed that: *“While interesting theoretically and very useful for several applications, like radar, these techniques have not found significant traction in the analysis of biological signals, which still rely mostly on STFT and CWT. As we will see below, directionally smoothed WVDs perform rather poorly on neuroscience data.”*

WVD-based techniques behave well only on a limited segment of biomedical signals, notably the electrocardiogram (EKG or ECG, in some papers), which is a very stereotypical signal and has a sparse, stereotypical spectrum, enabling successful application of WVD-based methods whose cross-terms are not significantly plaguing the representation. Similarly, these methods are also useful for analysis of electromyographic (EMG) muscle signals. **However, they perform poorly in resolving brief spectral components across multiple trials and in data that has a rich time-frequency structure, like brain signals—we maintain this claim.**

The reviewer suggested we search on Google Scholar for “choi-williams biomedical”. We did—from the first 30 results returned, 18 papers deal with actual applications on biomedical data, of which 15 are related to EKG/EMG, 4 to EEG, and 3 to other types of data (artery flow, etc.). Of the **only 4 papers on brain signals:**

- ⇒ one reports that **results obtained with CWD are inferior to those obtained with wavelets** [17];
- ⇒ the second deals only with **a narrow frequency band** (alpha) [18];
- ⇒ the third deals with a brain-machine interface where a classifier can learn from a TFR, which is **otherwise very contaminated with cross-terms** [19];
- ⇒ the last paper computes Shannon Entropy on the CWD to detect changes in EEG between eyes-open and eyes-closed conditions, or between ictal and non-ictal events [20]—**both have conditions that produce such dramatic changes in the spectrum that they are detectable even with the poorest TFRs.**

We are not aware of a single major impact neuroscience paper that has used CWD to analyze brain data. It is true, however, that we might have missed the very few that attempted to apply such methods—what is clear is that WVD-based techniques are rarely applied in neuroscience.

We reformulated the Main Text to be clearer about these issues, as follows: *“While interesting theoretically and very useful for several applications, like radar, these techniques have found limited traction in the analysis of biological (especially brain) signals, which still rely mostly on STFT and CWT. As we will see below, directionally smoothed WVDs perform rather poorly on neuroscience data.”*

8. Uncertainty product and resolution

Regarding the applicability of the uncertainty product (UCP) on multi-modal distributions, the reviewer mentions that many multi-modal distributions have well-behaved second moments.

The fact that certain multi-modal distributions have well-behaved moments does not mean that the UCP is also useful in such cases, especially when one discusses about resolution. For real-world cases (e.g., see Fig. 1, on page 2, above), when the spectrum displays a rich time-frequency structure, we believe it is obvious that the UCP is hardly of any use to estimate resolution—it is not even certain that the first and second moments are defined in such cases.

That the UCP is not fit for estimating resolution is obvious even on simpler setups. Consider for example two Gaussian atoms, neighboring in time. The UCP will increase as the distance between these atoms is increased—hence any inference about the resolution of a TFR is utterly flawed. This is what we meant.

We now clarified this in the SI: “*It should be noted however that the UCP is only interpretable in the context of resolution when the distributions in time and frequency are unimodal*”. We now also cite reference Folland and Sitaram, 1997 [21], in the context of the single-sided uncertainty product, as suggested by the reviewer.

To conclude, we thank the reviewer for the thorough review of the manuscript. We hope that the evidence, explanations, and proofs we have provided here are sufficient to convince the reviewer that our arguments are correct.

Please also see the Annex below, after the references.

References

- [1] B. Boashash and V. Susic, “Resolution measure criteria for the objective assessment of the performance of quadratic time-frequency distributions,” *IEEE Transactions on Signal Processing*, vol. 51, no. 5, pp. 1253–1263, May 2003, doi: 10.1109/TSP.2003.810300.
- [2] L. Stanković, “A measure of some time–frequency distributions concentration,” *Signal Processing*, vol. 81, no. 3, pp. 621–631, Mar. 2001, doi: 10.1016/S0165-1684(00)00236-X.
- [3] J. M. Lilly, “Element analysis: a wavelet-based method for analysing time-localized events in noisy time series,” *Proc Math Phys Eng Sci*, vol. 473, no. 2200, Apr. 2017, doi: 10.1098/rspa.2016.0776.
- [4] L. Liu, H. Hsu, and E. W. Grafarend, “Normal Morlet wavelet transform and its application to the Earth’s polar motion,” *Journal of Geophysical Research: Solid Earth*, vol. 112, no. B8, 2007, doi: 10.1029/2006JB004895.
- [5] I. Daubechies, “Orthonormal bases of compactly supported wavelets,” *Communications on Pure and Applied Mathematics*, vol. 41, no. 7, pp. 909–996, 1988, doi: 10.1002/cpa.3160410705.
- [6] S. Mallat, *A Wavelet Tour of Signal Processing: The Sparse Way*, 3 edition. Amsterdam ; Boston: Academic Press, 2008.
- [7] B. Boashash, *Time-Frequency Signal Analysis and Processing: A Comprehensive Reference*. Academic Press, 2015.

- [8] T. J. Barth, T. Chan, and R. Haimes, *Multiscale and Multiresolution Methods: Theory and Applications*. Springer Science & Business Media, 2001.
- [9] I. Daubechies, *Ten Lectures on Wavelets*. SIAM, 1992.
- [10] P. Loughlin, J. Pitton, and B. Hannaford, "Approximating time-frequency density functions via optimal combinations of spectrograms," *IEEE Signal Processing Letters*, vol. 1, no. 12, pp. 199–202, Dec. 1994, doi: 10.1109/97.338752.
- [11] C. Torrence and G. P. Compo, "A Practical Guide to Wavelet Analysis," *Bulletin of the American Meteorological Society*, vol. 79, no. 1, pp. 61–78, Jan. 1998, doi: 10.1175/1520-0477(1998)079<0061:APGTWA>2.0.CO;2.
- [12] L. Cohen, "The Uncertainty Principle for the Short-Time Fourier Transform and Wavelet Transform," in *Wavelet Transforms and Time-Frequency Signal Analysis*, L. Debnath, Ed. Boston, MA: Birkhäuser, 2001, pp. 217–232.
- [13] O. Rioul and P. Flandrin, "Time-scale energy distributions: a general class extending wavelet transforms," *IEEE Transactions on Signal Processing*, vol. 40, no. 7, pp. 1746–1757, Jul. 1992, doi: 10.1109/78.143446.
- [14] A. Grossmann, R. Kronland-Martinet, and J. Morlet, "Reading and Understanding Continuous Wavelet Transforms," in *Wavelets*, Berlin, Heidelberg, 1990, pp. 2–20, doi: 10.1007/978-3-642-75988-8_1.
- [15] F. Hlawatsch and G. F. Boudreaux-Bartels, "Linear and quadratic time-frequency signal representations," *IEEE Signal Processing Magazine*, vol. 9, no. 2, pp. 21–67, Apr. 1992, doi: 10.1109/79.127284.
- [16] M. Farge, "Wavelet transforms and their applications to turbulence," *Annual review of fluid mechanics*, vol. 24, no. 1, pp. 395–458, 1992.
- [17] L. Fraiwan, K. Lweesy, N. Khasawneh, H. Wenz, and H. Dickhaus, "Automated sleep stage identification system based on time–frequency analysis of a single EEG channel and random forest classifier," *Computer Methods and Programs in Biomedicine*, vol. 108, no. 1, pp. 10–19, Oct. 2012, doi: 10.1016/j.cmpb.2011.11.005.
- [18] M. Hansson-Sandsten, "Multitaper Wigner and Choi–Williams distributions with predetermined Doppler-lag bandwidth and sidelobe suppression," *Signal Processing*, vol. 91, no. 6, pp. 1457–1465, Jun. 2011, doi: 10.1016/j.sigpro.2010.10.010.
- [19] R. Alazrai, H. Alwanni, Y. Baslan, N. Alnuman, and M. I. Daoud, "EEG-Based Brain-Computer Interface for Decoding Motor Imagery Tasks within the Same Hand Using Choi-Williams Time-Frequency Distribution," *Sensors*, vol. 17, no. 9, Art. no. 9, Sep. 2017, doi: 10.3390/s17091937.
- [20] U. Melia, F. Claria, M. Vallverdu, and P. Caminal, "Measuring Instantaneous and Spectral Information Entropies by Shannon Entropy of Choi-Williams Distribution in the Context of Electroencephalography," *Entropy*, vol. 16, no. 5, Art. no. 5, May 2014, doi: 10.3390/e16052530.
- [21] G. B. Folland and A. Sitaram, "The uncertainty principle: A mathematical survey," *The Journal of Fourier Analysis and Applications*, vol. 3, no. 3, pp. 207–238, May 1997, doi: 10.1007/BF02649110.

[REDACTED]

- -

==== - -

==== - -

[REDACTED]

==== - -

====

[REDACTED]

[REDACTED]

[REDACTED]

[REDACTED]

REVIEWER COMMENTS

Reviewer #1 (Remarks to the Author):

In this second resubmission, the authors have changed little in response to my second review, noting that they “respectfully disagree” and that I have “not understood” their paper. I understand just fine when they write that “Superlets provide super-resolution in the time-frequency space and may become instrumental in discovering new phenomena in many fields of science.” This statement is a gross exaggeration, particularly since there are significant errors and misconceptions in the paper. I stand by my previous reviews and assessment that this paper is more of a signal processing paper that should be thoroughly vetted by experts in that field. As the authors noted in their previous rebuttal, “We feel that in many fields of science that use advanced signal processing methods, like neuroscience for example, there is still a superficial level of understanding and much confusion.” This paper will add to that confusion and misunderstanding.

They continue to defend their claims by making illogical “apples to oranges” comparisons. They present Fig. 1 in their rebuttal with the statement, “Results speak for themselves!” I gather that one is supposed to conclude that, because their method shows more ‘hot spots,’ it must clearly be more accurate? They fail to appreciate that in this example and all other examples, the true time-frequency structure of the signal is *unknown* (even for the synthetic signal examples where they think they know it, as I previously pointed out). As I have repeatedly noted, the marginal distributions of the different representations should be compared, as they are the *only* things known with certainty about the time-frequency structure of signals. If their method shows a hot spot that is not reflected in the power spectrum or the instantaneous power of the signal, how can it be more accurate?

Finally, the authors’ statement that they “never claimed that the resolution of superlets is better than that of the MMCE” and other time-frequency methods is astounding, and untrue! In addition to the quote in the first paragraph above, here are a few more excerpts from their submissions:

“Superlets outperform the STFT, CWT, and other super-resolution methods ... resolving time-frequency details with unprecedented precision.”

“As before, the SLT provided the best time-frequency resolution.”

“The closest time-frequency super-resolution method to superlets is the Fourier-based MMCE.”

“...the MMCE provided poorer frequency resolution...”

“...superlets provided a sharper picture.”

“Increasing the resolution of joint time-frequency estimation...has been a very active field of research in the past decades. Notable techniques include the Fourier-based MMCE ... Superlets extend these efforts ... They provide remarkable time-frequency resolution...”

The word “resolution” appears over 100 times in the manuscript! It is disingenuous that the authors now state that I have misunderstood, that they are not claiming to have invented a *super-resolution* method that is superior to MMCE and all other methods such that it may alter the course of history by “discovering new phenomena in many fields of science”!

Reviewer #3 (Remarks to the Author):

As I was asked specifically to look at the veracity of the rebuttal, let me preface my evaluation by saying that I have a quantitative background, but I am not a statistician or mathematician. One concern was that the superlet method was quite similar to an earlier method (MMCE) but not necessarily better. I think the authors have highlighted the difference — adaptive footprint versus fixed footprint — and the consequences for the amplitude with which a time-frequency blob appears — constant versus varying/dilution), which would make superlets better, but not in terms of frequency resolution. One may argue that the new method represents an incremental improvement, but it pushes the use of wavelets where otherwise short-time fourier transforms were used, which to me is sufficiently new, and speaking as a neuroscientist, I find superlets interesting enough to apply to the data we are analysing. (There bursts —time-frequency blobs — are challenging and methods we are using such as template matching were not good enough to get clean results.) So overall I was in favor of publication and that still is my opinion. Maybe the opinion of the other referee would be more supportive if the sentence in the abstract ‘Superlets outperform the STFT, CWT, and other super-resolution methods’ is made more nuanced. (And the same for similar statements appearing in the main text).

Comments on Reviewer 1's report:

Apart from toning down claims, the other issue is which method is better and in what way. The authors rely on the result that a blob is better visible with their method than the other, which does not entirely convince the reviewer. I am not sure how to improve that because for interesting data (experimental) you do not have a ground truth. An alternative would be to redo some experiments and use optogenetics to put in a transient signal of a particular frequency (similar to the seminal Cardin et al 2009 Nature paper) and ask, for which method the estimated amplitude of the blob corresponds better to the stimulation amplitude, and which method 2-alternative forced-choice task for detecting whether there was a stimulation or not. It is not appropriate to ask the authors to do this unless they have some data like that laying around. This situation can also be modeled using an autoregressive (AR) model in which you transiently vary the parameters to add in frequency content and ask the same questions, but that may again be less convincing for the reviewer since it is not experiment.

NCOMMS-19-31444C: Superlets: time-frequency super-resolution using wavelet sets

Vasile V. Moca, Harald Bârzan, Adriana Nagy-Dăbâcan, Raul C. Mureşan

Revision #3

Point-by-point reply to reviewers

We thank the editor and the two reviewers again! We did our best to address their latest concerns and to provide a clear demonstration of the utility of the new method on neuroscience data.

Summary of the revision:

1. As indicated by both reviewers, we applied a significant toning down of the claims by revisiting the entire manuscript. We now refocused the text to make it clear that we refer mainly to neuroscience data, where we show many examples of the performance of *superlets* (we also added a gallery on optogenetics data, mentioned below). In addition, we made the manuscript more compact and struggled to increase clarity. We now merged Fig. 7 and 8 and added a new Fig. 12.

2. As suggested by Reviewer #1 (R1), perhaps the super-resolution issue was overemphasized. This is because the initial version of the manuscript was focused on comparing the SLT with the traditional STFT and CWT. The comparisons with more capable techniques came only later. As the manuscript evolved, other arguments than resolution have come into play, like the non-dilution property, ability to detect bursts in noisy data, etc. In the new version of the manuscript, we reformulated the text to reduce focus on super-resolution and provide a fairer account of why and where the new method is useful.

3. As indicated by Reviewer #3 (R3), a clear, convincing example of the usefulness of the method on real data is now presented. We show multiple, compelling examples on data where the experimental setup is such that the presence and location of time-frequency packets is induced/evoked experimentally, i.e. one knows reasonably well where in time and/or frequency the packets should be located. We analyze both the local-field potential (LFP) and spiking of neurons, to prove the existence of oscillation bursts at particular time-frequency locations, induced or evoked by stimulation of cortical circuits using light (optogenetics). In addition to a better controlled experimental setting, we also confirm the results with an independent, time-domain technique called “Scaled Correlation” [1]. *We show that superlets can indeed robustly detect high-frequency oscillation packets that are mostly missed or poorly detected by other high-resolution methods, like the MMCE or WVD-based techniques, like Choi-Williams (CW) and Born-Jordan (BJ).*

4. In a gallery of examples in the Supplementary Information (SI) we show that, even on single trials of LFP data, *superlets* can reveal very high-frequency bursts (> 200 Hz) that cannot be resolved with other methods. These results are possibly of great importance to the neuroscience community, because, with the exception of specialized systems, like the auditory cortex for example, it is usually assumed that gamma bursting is localized in a significantly lower band (< 100-120 Hz). *Superlets may reveal a new landscape of very fast oscillation bursts that lies beyond the traditional frequency bands.*

Below, we describe the strategy we used to benchmark the methods on a better controlled experimental setting. We hope R1 will now be convinced of the value of the method and its specific advantage over the other methods – at least on neuroscience data.

Optogenetic control of brain rhythms

We are grateful for the excellent suggestion of R3 to use data with periodic optogenetic stimulation. Luckily, such data was already available in our lab. We stimulated the cortex using rectangular pulses of blue light at various stimulation frequencies (5-105 Hz) – in a setting that is similar to the seminal *Nature* papers of Cardin et al. 2009 and Sohal et al. 2009 [2], [3]. In addition to optogenetics, we applied Scaled-Correlation Analysis (SCA) [1] to confirm the existence of the oscillation packets (bursts) identified by *superlets*. Figure 12 shows the new results. Our strategy was the following:

Initially, we quantified the relative power increase induced by optical stimulation using baselined Welch power spectra for all stimulation frequencies. Out of the 32 recording channels (linear A32 Neuronexus probe), we considered the deepest 23, which showed responses to light. Photoelectric effects were ruled out by postmortem stimulation. We computed a *frequency response characteristic* by considering the frequency of the largest power increase in the 0-150Hz band, for each stimulation frequency (Fig. 12A). The lowest optogenetic stimulation frequency (5 Hz) induced gamma bursts with a dominant power increase around 55 Hz. As stimulation frequency was increased, the cortex gradually locked onto the stimulation frequency, switching from an induced to an evoked (locked in phase) gamma regime. We investigated both the induced and the evoked regime.

We explored the following scenarios:

a) Gamma bursts induced by low-frequency “quasi-tonic” optogenetic drive

At 5 Hz optogenetic stimulation, the rectangular pulse of blue light lasts for 100 ms in each cycle. This “quasi-tonic” drive engages the cortical circuit into beta/gamma oscillations (>20 Hz), via excitatory-inhibitory neuron interactions. The data in Fig. 12 was recorded from a Thy1-ChR2-YFP mouse, where blue light drives mainly principal, excitatory neurons [4]–[7]. The optogenetic drive of this population is compensated by balancing inhibition [8], yielding a push-pull Pyramidal-INterneuron Gamma (PING) oscillation burst [9], [10].

- ⇒ We chose one channel (the deepest) and analyzed first its LFP signal (Fig. 12B). *Superlets* (SLT) reveal multiple gamma oscillation bursts, which are hardly visible using the other high-resolution methods (MMCE or WVD-based). The MMCE parameters (windows and order) were precisely matched to the “windows” and order of the SLT, as required by R1.
- ⇒ We next focused on a clear and strong gamma burst around 55Hz, which matches the dominant frequency observed across all electrodes @5Hz optogenetic stimulation (Fig. 12A).
- ⇒ In this case, the optogenetic stimulation serves as a ground truth for the putative temporal localization of the gamma burst, but it does not tell us anything about the burst itself. Since only the SLT could clearly resolve it, the question arises if this 55Hz burst represents a real phenomenon or not.

- ⇒ To test if a real oscillation burst is present in the data, we applied a time-domain method (SCA) able to compute the autocorrelation of the signal in a way that focuses on a certain timescale [1]. SCA, with a scale segment of 18 ms, revealed a rich structure, with a clear periodic modulation at the location of the burst and matching exactly its period (Fig. 12C). Importantly, SCA can tune in on temporal correlations at different scales, depending on how the scale parameter is chosen (see [1] for more details).
- ⇒ To further verify our hypothesis that this burst was generated by the neural circuit in response to light, we analyzed the spikes (multi-unit activity) recorded on the same electrode. Indeed, optogenetic stimulation at 5Hz induces robust spiking (Fig. 12A), following the light pulses (Fig. 12D, top – see the Peri-Stimulus Time Histogram).
- ⇒ We next focused on the same time period as the SLT-observed gamma burst and, using the binary version of SCA, we found that the autocorrelation structure of the spike train displayed oscillatory modulation with the same period as the SLT-observed burst (Fig. 12D, bottom). Thus, the neural population was truly engaged in the 55Hz gamma rhythm that the SLT was signaling.

Taken together, these results clearly demonstrate the ability of *superlets* to detect and robustly report the presence of induced oscillation packets that cannot be well resolved by the other high-resolution methods we tested.

b) **Gamma bursts evoked by locking to high frequency optogenetic drive**

As stimulation frequency increased, the LFP became increasingly locked to the stimulation, i.e. the dominant frequency measured in the circuit approached the stimulation frequency (Fig. 12A). The advantage of studying the locked regime is that one knows the ground truth both about the time and frequency localization of the response. Ideally, we should have had a transient stimulation of just a few cycles, but a while back, when we designed the experiment, we followed the strategy described in Cardin et al. 2009 [2], where stimulation was applied throughout the trial. Luckily, as stimulation frequency increases, the cortex has “trouble” locking to the light pulses, and more and more frequently locking becomes transient. We thus searched for an example of transient locking as a test case, where the oscillation has both a well-known frequency dimension (due to locking) and a (measurable) temporal localization (due to the transient nature of the locking).

- ⇒ We focused on a single trial, recorded by a more superficial electrode, during 65 Hz stimulation (Fig. 12E). Here, the cortex engaged transiently with the stimulation as revealed by the *superlets* and by the filtered signal in the 60-70 band (Fig. 12F). All methods were able to detect this locking. Due to the matching of the MMCE with the SLT windows, the former had very high frequency resolution but poor temporal resolution in the high frequencies, as also demonstrated in our previous rebuttal. In addition, the MMCE displays excessive fragmentation in the lower frequencies, due to the smallest window, which matches the shortest wavelet at 150 Hz. Similarly to the MMCE, the CW and BJ were mostly unexpressive above 65 Hz.
- ⇒ Importantly, the *superlets* revealed high-frequency bursts at double the stimulation frequency. Posthoc investigation showed that such bursts are ubiquitous in the data we recorded because, at high stimulation frequency, the neural population tends to be entrained by both the rising and falling fronts of the light pulse (i.e., at double the

stimulation frequency) – see also point c), below. This high frequency landscape is hardly recognizable with the other high-resolution methods (Fig. 12E).

- ⇒ The high frequency bursts revealed by *superlets* were indeed obvious in the filtered signal (120-140 Hz; Fig. 12F) and confirmed using SCA (Fig. 12G).
- ⇒ Importantly, we confirmed the transient nature of the coupling at 65Hz by computing the event-related potential, aligned to the light onset of each pulse and during the 3 periods indicated by the SLT (see periods 1, 2, and 3 in Fig. 12E and Fig. 12H). Finally, we confirmed the locking to 2x the stimulation frequency during the high-frequency burst detected by *superlets* (Fig. 12E, first red arrow and Fig. 12I).

Thus, *superlets* were able to robustly detect evoked oscillations and at the same time resolve high-frequency packets, likely emerging from the cortical activations evoked by phasic stimulus changes. The other methods we tested failed to reveal this very interesting behavior – which is documented in more detail below.

c) Very high frequency bursts in single trials

In Fig. S13-S15 of the SI we show examples of analyses in single trials, with various optogenetic stimulation frequencies.

- ⇒ Results indicate that robust high-frequency bursts, sometimes exceeding 200 Hz, occur in LFP data in response to optogenetic stimulation. In some cases, these bursts match the double of the stimulation frequency (Fig. S15), as we have shown above.
- ⇒ Importantly, the ability of *superlets* to reveal these high frequency oscillation packets was not matched by the other high-resolution methods. The MMCE, with windows matched to those of the SLT, showed excessive fragmentation and dilution. Paradoxically, the CW and BJ seemed to provide closer representations to the SLT than the MMCE.

Because *superlets* compute power at scale and scale with frequency, they are able to detect high frequency packets, which are shorter in time and whose power is otherwise diluted by methods based on the Fourier transform and by WVD-based techniques, like the CW and BJ. We believe that the gallery featured in SI provides compelling evidence of the advantage of *superlets* in resolving high-frequency bursts. We hope these examples are convincing enough that *superlets* have the potential to unravel phenomena in neuroscience data that are difficult to observe with other methods. Last but not least, the detection of such phenomena (very high frequency bursting) may change our understanding of the expression, extent, and function of transient bursting in neuroscience data.

Point-by-point replies to reviewers:

Replies to Reviewer #1:

From the outset, we would like to express our appreciation for the reviewer's efforts to help improving the manuscript. Their concerns were of great help to us for understanding the differences between our method and other techniques, to develop analytical proofs, and to figure out the relevant coordinates on which the techniques should be compared. We agree with the reviewer that the manuscript needed to be toned down – our enthusiasm was prompted by

the ability of *superlets* to reveal interesting phenomena in neural data. For example, we were able to identify gamma (30-100 Hz) bursts in EEG data that was recorded 10 years ago in our lab. We were hunting for these bursts for a long time and no other method was helpful. We honestly report that *superlets* are useful for neuroscience data and we hope that the reviewer now agrees with us.

Matching parameters and usage of the “resolution” word with various meanings

Regarding the comparison of “apples and oranges”, we have shown throughout the several revisions of the paper and explained both with examples and analytical arguments why the MMCE windows cannot, in general, be matched with the *superlets*’ wavelets “sizes”. We now explicitly mention this early, when we discuss results from Fig. 7 in the main text. We added a new figure in SI (Fig. S3A), where the MMCE has been matched precisely with the shortest and longest “windows” of the ASLT, as the reviewer insists. The result is the loss of temporal resolution, as expected. Across the 2-150Hz range, the shortest wavelet spans 24ms and the longest 1800ms. With these parameters the MMCE becomes excessively smeared out in time and many important details are lost. The wider the spectral range that is represented, the worse the wavelet-matched MMCE becomes.

MMCE and *superlets* cannot be matched in general. To get a reasonable representation of the MMCE across such wide spectral ranges one must find a trade-off set of window sizes (see Fig. S3). It can be easily shown that with such trade-off parameters, the MMCE has poorer frequency resolution in the low frequency range and poorer time resolution in the high frequency range. In the new analyses (new Fig. 12 and gallery from SI with Figs. S13-S15) we tried to find trade-off parameters but could not find any settings where the MMCE reveals well the known time-frequency packets – therefore, we used the matched MMCE, as the reviewer suggested. This, unfortunately, does serve the method at all.

The *superlets* and MMCE are simply different methods and, on the particular neuroscience applications we tested the methods, the MMCE performs less well (please see also the next section of our reply). We believe we have provided compelling examples and arguments about this and hope the reviewer agrees.

We also hope the reviewer agrees with us that we use the term “resolution” in various contexts and that there are three different meanings: temporal resolution, frequency resolution, and joint time-frequency resolution. When we said, in the previous rebuttal, that we did not claim *superlets* have better resolution than MMCE, we meant better *joint* time-frequency resolution. We apologize if our statement was confusing.

Indeed, the *joint* time-frequency resolution of *superlets* and matched MMCE is similar or close, across the TF plane – as we have shown in the previous rebuttal and as former Fig. 12 (now Fig. 11) clearly shows. When a single frequency is present in the data (e.g., like in the case of Fig. 11), the windows of the MMCE and SLT can be matched at that single frequency and the two methods provide very similar results. We write:

“When a single frequency or a limited range of frequencies is present in the data, the windows of the MMCE can be matched such that the shortest and longest window match the extent of the shortest and longest superlet wavelets, as shown in Fig. 11. In such cases, the time-frequency resolution of the two methods is very similar (Fig. 11C and F). It should be noted however that this window matching is not usually helpful on real data, especially when the signal contains a wide range of frequencies that are explored.”

We now clarified the text further to avoid any confusion about what we mean when we refer to “resolution”. We refer explicitly to either temporal, frequency, or time-frequency resolution when these two methods are compared. We also tried to reduce, as much as we could, the usage of the “resolution” term throughout the text.

Why the results of superlets are important for neuroscience

Why are we excited about blobs appearing in the high frequency range in neuroscience data? We feel we should better explain this here.

Neural oscillations are produced by a variety of mechanisms from interaction of excitatory and inhibitory cells, which create rhythmic excitation followed by inhibition, to synchronization of larger populations of neurons activated rhythmically. What is ubiquitous about neural oscillations is their so-called $1/f$ property, i.e. their amplitude (power) decreases with frequency [11]. This stems from multiple sources. In intracortical recordings “fast” rhythms like gamma (30-80 Hz) have generally lower power because they are usually generated by a local circuit, involving a limited number of cells [12]. In addition, the cortical tissue has low-pass filtering properties, such that fast rhythms have usually lower power and are detected locally, while slow oscillations have larger power and propagate further. On top of this, in EEG, which is recorded outside the skull, the electrical potential has to travel through the bone, which is a very effective low-pass filter, such that gamma bursts are very difficult to find – some labs never even venture to look for fast oscillations in EEG.

More recently, people have realized that in intra-cortical recordings (where the bone does not attenuate high frequencies) the high-frequency oscillations can have large amplitude, but they appear very transiently. A review was just published in September 2020 in *Frontiers* about this issue [13]. The problem is to find these bursts, because with traditional methods, like those based on the spectrogram, the low frequencies overshadow what is going on in the upper bands. This happens because a fixed analysis window is used across all frequencies and the “dilution phenomenon” kicks in, as we have proven mathematically. These fast, transient bursts, even if they have the same peak amplitude as the slower rhythms (and the same number of cycles), would produce less power per analysis window than their slower counterparts, which “fill” a larger proportion of the analysis window. Any Fourier-based technique “suffers” from this issue. We do not mean this is wrong, we just want to underline that such techniques cannot or have a hard time detecting legitimate fast bursts, which have the same peak amplitude as the slow bursts, but yield less total power relative to a fixed-size window. The natural solution to this is to use multi-scale analysis, where the window compresses with scale, and normalization is such that bursts with the same peak amplitude produce the same “intensity” of the representation. All this is hinting to an interesting direction: fractals.

One of the pioneers trying to understand the properties of brain signals from a dynamical systems perspective was Walter Freeman, whose seminal work has emphasized the importance of the $1/f$ property and has outlined the fractal nature of EEG¹. Indeed, brain rhythms seem to obey fractal properties as they emerge from a complex dynamical system that is self-similar across temporal and spatial scales [14]–[16]. In this context, the L^1 norm wavelet is ideally positioned to detect oscillation bursts that are self-similar across scales: Indeed, this representation provides the same “intensity” for packets of different frequency that have the

¹ <https://slideplayer.com/slide/5099493/>

same peak amplitude. More recently, people started to realize the value of this representation for studying natural phenomena, many of which obey power-law scaling and act as fractals. Unsurprisingly, some of the most popular scientific tools have followed this development: Matlab uses the L^1 norm in its wavelet library.

While this L^1 normalization of wavelets is promising (and, indeed, most recent neuroscience papers that employed wavelets used it – some not even being aware of it!), the big issue has always been the inability to properly localize in frequency the fast oscillation bursts. When developing the *superlet* method our intention was to solve this issue by improving on the representation of higher frequencies.

Why are these higher frequencies important for neuroscience? There are many reasons but, primarily, it is thought that fast rhythms (30-80Hz or more extended 30-120Hz) organize and coordinate neural populations because their timescale approaches the relevant timescales of neurons and synapses [17]–[19]. As a consequence, these rhythms may impose precise timing of firing for neurons and shape the coordination between brain areas. The difficulty to estimate the expression of fast rhythms has prompted a (now) three decade-old scientific debate between those who believe that the fast oscillations are meaningful and essential to brain function, and those who claim they are unable to see/find them, or consider them to be an epiphenomenon. Evidence keeps accumulating in favor of the first group, as more and more capable methods are developed, which can reveal the “high frequency blobs”. Our manuscript attempts to add another brick to this effort and, as the extensive testing shows, it is likely that *superlets* will prove useful.

In brain data, estimating slow rhythms (< 30 Hz) is not generally a problem, because these have a lot of power and their temporal modulation is relatively sluggish. On these timescales, the brain acts as an “inertial” system, such that temporal changes in slow oscillations rarely happen abruptly. Slow rhythms have usually slow temporal features because of the nature of the mechanisms that produce them. By contrast, fast rhythms (>30 Hz) pose challenges to estimation methods because these arise from fast neuron-neuron interactions, fast synaptic timescales (on the order of few milliseconds), fast coupling delays (sometimes sub-millisecond), and even direct electrical coupling between neurons via gap junctions (which is extremely fast). As a result, transient, short-lived oscillation bursts whose temporal modulation can be extremely abrupt, are the norm in the high frequency landscape. The ability to fish out these “blobs” is of paramount importance.

The question then arises: why not focus analysis only on a limited high-frequency band to prevent the high-power, slow rhythms from masking out the high frequency range? This was indeed a valid strategy people employed a lot. But there are several big limitations here. First, how does one know where (in what sub-band) to look for fast oscillation bursts? Scanning multiple narrow frequency bands systematically, in tons of ephys or EEG data, is very cumbersome. Second, how does one know that scattered blobs appearing in a narrow frequency band are not simply noise, with practically insignificant amplitude? One should be able to relate, in one view, the amplitude of these blobs with the rest of the spectral landscape, to assign significance to them. Third, in many cases, neuroscientists are looking for coordinated rhythms across frequency bands, such that exploring a large frequency range at once becomes necessary. For all these reasons, methods able to reveal self-similar oscillation bursts, occurring over a wide frequency range, are of great practical value. It is desirable to have representations that can reveal similar peak amplitudes (or powers @ scale) with similar “intensity” and do so over an “as wide as possible” frequency range. On neuroscience data, it is also desirable to have methods whose temporal resolution increases as the oscillation packets compress and

frequency increases, because, from a physiological and mechanistic point of view, the temporal dimension becomes more and more important as frequency increases.

The reason we are excited about *superlets* is that they achieve well these requirements and truly help in analyzing neuroscience data. How and if they will translate to other types of data, remains to be seen. We now struggled to explain these aspects better in the manuscript and we very much hope that the reviewer agrees with us! We thank the reviewer again!

Replies to Reviewer #3:

The reviewer's suggestion to use data recorded with optogenetic stimulation has added a lot of value to the manuscript. Indeed, we are now able to clearly illustrate the usefulness of the method and to demonstrate that *superlets* can detect genuine oscillation bursts that are missed or poorly represented by the other methods we tested. Overall, we are very grateful for all the constructive suggestions and for helping us convey a clearer message. We thank the reviewer once more!

To conclude, we thank the two reviewers for their constructive review of the manuscript and the editor for his patience and willingness to give our manuscript a fair review. We hope the manuscript is now in a sufficiently mature state for publication.

On a funny side note, the method is catching more and more momentum and the attention of the signal processing community. For example, some people consider *superlets* related in some way to Pink Floyd (<http://www.laurent-duval.eu/siva-wits-where-is-the-starlet.html>):

"Futurelet(ter)

Some starlets are shining like crazy diamonds, on the dark side of the moon. Awaiting capture, here they are on the run: <https://www.biorxiv.org/content/biorxiv/early/2019/03/21/583732.full.pdf>

Starlet element number 124 superlet: Superlets: time-frequency super-resolution using wavelet sets".

References

- [1] D. Nikolić, R. C. Mureşan, W. Feng, and W. Singer, "Scaled correlation analysis: a better way to compute a cross-correlogram," *European Journal of Neuroscience*, vol. 35, no. 5, pp. 742–762, 2012, doi: 10.1111/j.1460-9568.2011.07987.x.
- [2] J. A. Cardin *et al.*, "Driving fast-spiking cells induces gamma rhythm and controls sensory responses.," *Nature*, vol. 459, no. 7247, pp. 663–667, Jun. 2009, doi: 10.1038/nature08002.
- [3] V. S. Sohal, F. Zhang, O. Yizhar, and K. Deisseroth, "Parvalbumin neurons and gamma rhythms enhance cortical circuit performance.," *Nature*, vol. 459, no. 7247, pp. 698–702, Jun. 2009, doi: 10.1038/nature07991.
- [4] H. Dana *et al.*, "Thy1 transgenic mice expressing the red fluorescent calcium indicator jRGECO1a for neuronal population imaging in vivo," *PLoS One*, vol. 13, no. 10, Oct. 2018, doi: 10.1371/journal.pone.0205444.

- [5] G. Feng *et al.*, “Imaging Neuronal Subsets in Transgenic Mice Expressing Multiple Spectral Variants of GFP,” *Neuron*, vol. 28, no. 1, pp. 41–51, Oct. 2000, doi: 10.1016/S0896-6273(00)00084-2.
- [6] Q. Chen *et al.*, “Imaging Neural Activity Using Thy1-GCaMP Transgenic Mice,” *Neuron*, vol. 76, no. 2, pp. 297–308, Oct. 2012, doi: 10.1016/j.neuron.2012.07.011.
- [7] D. L. Dobbins, D. C. Klorig, T. Smith, and D. W. Godwin, “Expression of Channelrhodopsin-2 Localized within the Deep CA1 Hippocampal Sublayer in the Thy1 Line 18 Mouse,” *Brain Res*, vol. 1679, pp. 179–184, Jan. 2018, doi: 10.1016/j.brainres.2017.11.025.
- [8] H. He and H. T. Cline, “What Is Excitation/Inhibition and How Is It Regulated? A Case of the Elephant and the Wisemen,” *J Exp Neurosci*, vol. 13, Jun. 2019, doi: 10.1177/1179069519859371.
- [9] V. V. Moca and R. C. Mureşan, “Emergence of beta/gamma oscillations: ING, PING, and what about RING?,” *BMC Neurosci*, vol. 12, no. Suppl 1, p. P230, Jul. 2011, doi: 10.1186/1471-2202-12-S1-P230.
- [10] C. Börgers and N. Kopell, “Synchronization in networks of excitatory and inhibitory neurons with sparse, random connectivity,” *Neural Computation*, vol. 15, no. 3, pp. 509–538, Mar. 2003, doi: 10.1162/089976603321192059.
- [11] G. Buzsáki, *Rhythms of the brain*. Oxford University Press, 2006.
- [12] V. V. Moca, D. Nikolic, W. Singer, and R. C. Mureşan, “Membrane resonance enables stable and robust gamma oscillations,” *Cereb. Cortex*, vol. 24, no. 1, pp. 119–142, Jan. 2014, doi: 10.1093/cercor/bhs293.
- [13] I. Tal, S. Neymotin, S. Bickel, P. Lakatos, and C. E. Schroeder, “Oscillatory Bursting as a Mechanism for Temporal Coupling and Information Coding,” *Front. Comput. Neurosci.*, vol. 14, 2020, doi: 10.3389/fncom.2020.00082.
- [14] G. Werner, “Fractals in the Nervous System: Conceptual Implications for Theoretical Neuroscience,” *Front Physiol*, vol. 1, Jul. 2010, doi: 10.3389/fphys.2010.00015.
- [15] W. Lutzenberger, H. Preissl, and F. Pulvermüller, “Fractal dimension of electroencephalographic time series and underlying brain processes,” *Biol. Cybern.*, vol. 73, no. 5, pp. 477–482, Oct. 1995, doi: 10.1007/BF00201482.
- [16] D. Nikolić, V. V. Moca, W. Singer, and R. C. Mureşan, “Properties of multivariate data investigated by fractal dimensionality,” *Journal of Neuroscience Methods*, vol. 172, no. 1, pp. 27–33, Jul. 2008, doi: 10.1016/j.jneumeth.2008.04.007.
- [17] W. Singer, “Neuronal synchrony: a versatile code for the definition of relations?,” *Neuron*, vol. 24, no. 1, pp. 49–65, 111–25, Sep. 1999.
- [18] P. Fries, “A mechanism for cognitive dynamics: neuronal communication through neuronal coherence.,” *Trends Cogn Sci*, vol. 9, no. 10, pp. 474–480, Oct. 2005, doi: 10.1016/j.tics.2005.08.011.
- [19] P. Fries, D. Nikolić, and W. Singer, “The gamma cycle,” *Trends in Neurosciences*, vol. 30, no. 7, pp. 309–316, Jul. 2007, doi: 10.1016/j.tins.2007.05.005.

REVIEWERS' COMMENTS

Reviewer #3 (Remarks to the Author):

The authors have spent a lot of effort to address the comments of the reviewers. As mentioned before I was happy with the previous version of the manuscript, the added figure makes the paper even more attractive to neuroscientists and the authors have toned down some of their statements.

The new results of optogenetics would however benefit from a couple of references that directly speak to the new results, i.e. on p24 "Interestingly, at low stimulation frequency the cortex engages into induced gamma oscillations at significantly higher frequency than that of the stimulating signal. At 5 Hz drive, we observed ~53-55 Hz and vigorous spiking", please see Tiesinga (2012) European Journal of Neuroscience, as this is one of the findings there. And on p.25, "it is likely that the observed gamma oscillations reflect a push-pull Pyramidal-INterneuron Gamma (PING) interaction", the issue of PING vs ING in reference to these optogenetics manipulations are discussed in Tiesinga & Sejnowski (2009) Neuron.

NCOMMS-19-31444D: Time-frequency super-resolution with superlets

Vasile V. Moca, Harald Bârzan, Adriana Nagy-Dăbâcan, Raul C. Mureşan

Revision #4

Point-by-point reply to reviewers

We have now addressed all the issues raised by Reviewer #3 and the requirements in the author checklist. In summary:

1. As indicated by Reviewer #3, we cited the two papers by Tiesinga (2012) and Tiesinga and Sejnowski (2009) in the relevant sections of the manuscript.
2. We reformatted the figure labels, italics, references, language, as indicated in the author checklist. We followed closely the formatting instructions.
3. We generated figures in vector format, as required.

Reply to Reviewer #3:

“The authors have spent a lot of effort to address the comments of the reviewers. As mentioned before I was happy with the previous version of the manuscript, the added figure makes the paper even more attractive to neuroscientists and the authors have toned down some of their statements.

The new results of optogenetics would however benefit from a couple of references that directly speak to the new results, i.e. on p24 “Interestingly, at low stimulation frequency the cortex engages into induced gamma oscillations at significantly higher frequency than that of the stimulating signal. At 5 Hz drive, we observed ~53-55 Hz and vigorous spiking”, please see Tiesinga (2012) European Journal of Neuroscience, as this is one of the findings there. And on p.25, “it is likely that the observed gamma oscillations reflect a push-pull Pyramidal-INterneuron Gamma (PING) interaction”, the issue of PING vs ING in reference to these optogenetics manipulations are discussed in Tiesinga & Sejnowski (2009) Neuron.”

We thank the reviewer for pointing out the relevant literature! We now write:

“Interestingly, at low stimulation frequency the cortex engages into induced gamma oscillations at significantly higher frequency than that of the stimulating signal. At 5 Hz drive, we observed ~53-55 Hz and vigorous spiking (Fig. 10a, “Frequency” and “Firing rate”), as shown before by Tiesinga⁵⁸.”

Ref. 58: Tiesinga, P. H. E. Motifs in health and disease: the promise of circuit interrogation by optogenetics. European Journal of Neuroscience 36, 2260–2272 (2012).

“Given the tonic optogenetic drive of the excitatory population during the target period, it is likely that the observed gamma oscillations reflect a push-pull Pyramidal-INterneuron Gamma (PING) interaction^{48,60,61} between excitatory and inhibitory cells, whereby inhibition balances out excitation⁶² (see Tiesinga and Sejnowski⁶³).”

Ref. 63: Tiesinga, P. & Sejnowski, T. J. Cortical Enlightenment: Are Attentional Gamma Oscillations Driven by ING or PING? *Neuron* 63, 727–732 (2009).

Finally, we would like to thank Reviewer #3 again for the very constructive and helpful review!